# How to Square Tensor Networks and Circuits Without Squaring Them

**Lorenzo Loconte**[⊥]
l.loconte@sms.ed.ac.uk

**Adrián Javaloy**[⊥]
ajavaloy@ed.ac.uk

**Antonio Vergari**[⊥]
avergari@ed.ac.uk

[⊥]School of Informatics, University of Edinburgh, UK

## Abstract

Squared tensor networks (TNs) and their extension as computational graphs—squared circuits—have been used as expressive distribution estimators, yet supporting closed-form marginalization. However, the squaring operation introduces additional complexity when computing the partition function or marginalizing variables, which hinders their applicability in ML. To solve this issue, canonical forms of TNs are parameterized via unitary matrices to simplify the computation of marginals. However, these canonical forms do not apply to circuits, as they can represent factorizations that do not directly map to a known TN. Inspired by the ideas of orthogonality in canonical forms and determinism in circuits enabling tractable maximization, we show how to parameterize squared circuits to overcome their marginalization overhead. Our parameterizations unlock efficient marginalization even in factorizations different from TNs, but encoded as circuits, whose structure would otherwise make marginalization computationally hard. Finally, our experiments on distribution estimation show how our proposed conditions in squared circuits come with no expressiveness loss, while enabling more efficient learning.

## 1 Introduction

Tensor networks (TNs) are low-rank factorizations of tensors with applications in machine learning (Stoudenmire and Schwab, 2016; Han et al., 2018; Cheng et al., 2019; Novikov et al., 2021; Tomut et al., 2024), quantum physics (Schollwoeck, 2010; Biamonte and Bergholm, 2017) and quantum computing (Markov and Shi, 2008). A TN factorizes a complex function $\psi$ over a set of variables $\mathbf{X} = \{X_i\}_{i=1}^d$ having domain $\mathrm{dom}(\mathbf{X})$, which can be then used to model a probability distribution via modulus squaring, i.e., $p(\mathbf{X}) = Z^{-1}|\psi(\mathbf{X})|^2$, where $Z = \int_{\mathrm{dom}(\mathbf{X})} |\psi(\mathbf{x})|^2 \, \mathrm{d}\mathbf{x}$ is the partition function. Recently, Loconte et al. (2025b;a) have shown that the computations done with a TN can be generalized into computational graphs akin to neural networks, called *circuits* (Darwiche and Marquis, 2002; Choi et al., 2020b; Vergari et al., 2021). This is done by casting contractions between tensors in a TN into a hierarchical composition of sum and product computational units.

The language of circuits offers the opportunity to flexibly build *novel* TN factorizations by stacking layers of sums and products as "Lego blocks" (Loconte et al., 2025a), including different basis input functions, and providing a seamless integration with deep learning architectures (Shao et al., 2022; Gala et al., 2024a;b). Moreover, viewing TNs as circuits allows one to exploit a rich framework of *structural properties*, defined over their computational graph and parameterization, to compose circuits and compute several probabilistic reasoning tasks in closed-form. These include the evaluation of information-theoretic measures and expectations (Vergari et al., 2021), which is crucial for example in reliable neurosymbolic AI (Ahmed et al., 2022; Kurscheidt et al., 2025; Marconato et al., 2024a;b) and causal inference (Choi et al., 2020a; Wang et al., 2024). This is done with *probabilistic circuits* (PCs)—circuits encoding probability distributions, that are traditionally restricted to have positive parameters only, i.e., *monotonic* PCs (Shpilka and Yehudayoff, 2010).

To increase the expressiveness of PCs for representing complicated distributions, one can equip them with real parameters and *square* them (Loconte et al., 2024), similar to TNs. Mixing squared PCs together also provides further expressiveness gains (Loconte et al., 2025b). However, differently from

monotonic PCs that are not squared, squared PCs require additional overhead to be normalized, i.e., to compute $Z$. That is, under particular structural properties, squaring circuits and computing $Z$ has quadratic complexity w.r.t. the circuit size (Vergari et al., 2021). This quadratic complexity overhead carries over the computation of marginals that are simpler than the partition function, i.e., where only a proper subset of variables are integrated out. This overhead limits the application of squared PCs in settings where performing exact yet efficient conditioning is crucial, e.g., as for sampling (Loconte et al., 2024) and in lossless data compression (Yang et al., 2022; Liu et al., 2022).

One possible solution might come from the TN literature, where *canonical forms* are used to simplify the computation of marginals (Schollwoeck, 2010; Bonnevie and Schmidt, 2021). E.g., instead of computing the partition function $Z$ explicitly in a TN, a canonical form ensures $|\psi(\mathbf{X})|^2$ is an already-normalized distribution, i.e., $Z = 1$. In practice, canonical forms are obtained by parameterizing a TN by means of (semi-)unitary matrices. However, different TNs need different canonical forms, and each of them is tailored for specific marginals only, yielding left/right/mixed/upper canonical forms in matrix-product states (MPS) (Orús and Vidal, 2008) and tree TNs (TTNs) (Shi et al., 2006; Cheng et al., 2019). These canonical forms cannot be applied to circuits, as they might not correspond to a known factorization method or TN (Loconte et al., 2025a). Here, we extract the core principle behind canonical forms and reformulate it in terms of new structural properties of circuits. Our conditions revolve around the idea of orthogonality between units of the circuit computational graph, which we find to be surprisingly related with a classical circuit property, called *determinism*, which so far has been mostly exploited in the context of tractable maximization (Darwiche and Marquis, 2002) with PCs and never linked to TNs before.

**Our main contributions** are the following: **(i)** We derive properties based on orthogonality to enable linear-time marginalization in squared PCs, thus improving over its usual quadratic complexity w.r.t. their size (§3). **(ii)** Since PCs often consist of densely-connected layers of sums and products, we relax our orthogonality properties over scalar units in favor of a parameterization over layers, exploiting (semi-)unitary matrices instead. While this parameterization is similar to canonical forms in TTNs, we show it generalizes to a *strictly* larger set of factorizations when represented as circuits (§4). **(iii)** Under this parameterization, we derive an algorithm to marginalize *any variable subset* whose best-case complexity scales linearly w.r.t. the number of layers and their size, thus finding a better complexity bound than a previous known one that squared all layer sizes (§5). **(iv)** Our experiments on distribution estimation show no performance loss under the proposed circuit properties, while enabling more efficient training and the use of previously unavailable circuit architectures (§6).

## 2 FROM TENSOR NETWORKS TO SQUARED PROBABILISTIC CIRCUITS

We introduce the close relationship between TNs and circuits (Loconte et al., 2024; 2025a), and show how they can encode probability distributions via modulus squaring. TNs encode hierarchical factorizations of high dimensional tensors (or functions). Perhaps the most popular TN factorization is the matrix-product state (MPS) (Pérez-García et al., 2007), also called tensor-train (Oseledets, 2011). A rank-$R$ MPS factorization encodes a function $\psi$ over variables $\mathbf{X} = \{X_j\}_{j=1}^d$ as

$$\psi(\mathbf{x}) = \sum\nolimits_{i_1=1}^R \sum\nolimits_{i_2=1}^R \cdots \sum\nolimits_{i_{d-1}=1}^R \psi_1^{i_1}(x_1)\psi_2^{i_1,i_2}(x_2)\cdots\psi_{d-1}^{i_{d-2},i_{d-1}}(x_{d-1})\psi_d^{i_{d-1}}(x_d), \quad (1)$$

where $\psi_1\colon \mathsf{dom}(X_1) \to \mathbb{C}^R$, $\psi_d\colon \mathsf{dom}(X_d) \to \mathbb{C}^R$, and $\psi_k\colon \mathsf{dom}(X_k) \to \mathbb{C}^{R\times R}$ with $1 < k < d$ are the *factors*. The superscript indices in Eq. (1) select scalar entries from the factors. Note that in the case of $\mathbf{X}$ being discrete with finite domain, Eq. (1) can be seen as a factorization of a $d$-dimensional tensor (Kolda, 2006; Kolda and Bader, 2009). Given an assignment $\mathbf{x} = \langle x_1, \ldots, x_d \rangle \in \mathsf{dom}(\mathbf{X})$, computing the value of $\psi(\mathbf{x})$ translates to evaluating the univariate factors, products and sums in Eq. (1), i.e., a complete *contraction* of the TN (Orús, 2013). While the naive way of contracting the TN in Eq. (1) requires time $\mathcal{O}(R^d)$, one can do it in time $\mathcal{O}(R^2d)$ by computing products and sums in a precise left-to-right ordering, i.e., as $d-1$ matrix-vector products. The computational graph of sums and products resulting from the TN contraction in a particular ordering is a *circuit*.

**Definition 1** (Circuit (Choi et al., 2020b; Vergari et al., 2021))**.** A *circuit* $c$ is a parameterized computational graph over variables $\mathbf{X}$ encoding a function $c\colon \mathsf{dom}(\mathbf{X}) \to \mathbb{C}$, and comprising three kinds of units: *input*, *product* and *sum*. Each product or sum unit $n$ receives the outputs of other units as inputs, denoted with the set $\mathsf{in}(n)$. Each unit $n$ encodes a function $c_n$ defined as: (i) $f_n(X)$ if $n$ is an input unit, where $f_n$ is a function over the variable $\mathsf{sc}(n) = \{X\} \subseteq \mathbf{X}$, called its

Figure 1: **Matrix-product states (MPSs) are circuits.** A MPS TN of rank $R = 2$, here in Penrose graphical notation (bottom right), models a function $\psi$ over $\mathbf{X} = \{X_1, X_2, X_3\}$ as $\psi(\mathbf{X}) = \sum_{i_1=1}^{R} \sum_{i_2=1}^{R} \psi_1^{i_1}(X_1) \psi_2^{i_1,i_2}(X_2) \psi_3^{i_2}(X_3)$. Given an assignment $\mathbf{x} = \langle x_1, x_2, x_3 \rangle$, the circuit computes the complete contraction of the MPS, i.e., $\psi(\mathbf{x})$ (above left). The circuit input units (⊘) compute the factors $\psi_1^{i_1}, \psi_2^{i_1,i_2}, \psi_3^{i_2}$ over $X_1, X_2, X_3$, highlighted in their respective colors. The composition of product (⊗) and sum (⊕) units encode the contraction of the factors following a left-to-right ordering, i.e., multiplying and summing the violet ($\psi_1$) and orange ($\psi_2$) factors before the green one ($\psi_3$). Here, sum weights are fixed to 1, but can generally be any complex number.

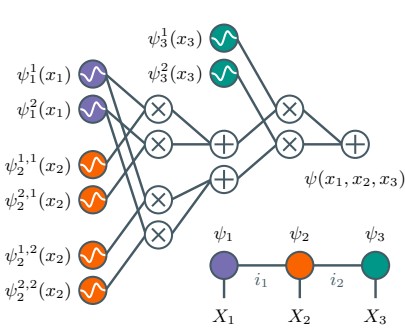

*scope*, (ii) $\prod_{i \in \mathsf{in}(n)} c_i(\mathsf{sc}(i))$ if $n$ is a product, and (iii) $\sum_{i \in \mathsf{in}(n)} w_{n,i} c_i(\mathsf{sc}(i))$ if $n$ is a sum, where $\{w_{n,i} \in \mathbb{C} \setminus \{0\}\}_{i \in \mathsf{in}(n)}$ are the parameters of the sum unit. The scope of a product or sum unit $n$ is the union of the scopes of its inputs, i.e., $\mathsf{sc}(n) = \bigcup_{i \in \mathsf{in}(n)} \mathsf{sc}(i)$.

The circuit size, denoted as $|c|$, is the number of edges between the units. Evaluating a circuit $c$ on an variables assignment $\mathbf{x}$, i.e., computing $c(\mathbf{x})$, is done by evaluating the input functions, products and sums, by following the computational graph, thus requiring time $\mathcal{O}(|c|)$. Fig. 1 shows an example of an MPS approximation of $\psi$ represented in Penrose graphical notation (Penrose, 1971), and the circuit $c$ encoding its left-to-right contraction, i.e., $\psi(\mathbf{x}) = c(\mathbf{x})$ as in Eq. (1). The circuit language allows us to build factorizations by directly connecting sums and products, which in the end might not correspond to any known TN structure or other tensor factorization method (Loconte et al., 2025a).

**Structural properties** specified over a circuit graph structure provide sufficient conditions to guarantee the tractable computation of quantities useful in a number of scenarios (Darwiche and Marquis, 2002; Vergari et al., 2021; Wang et al., 2024). For example, a circuit $c$ supports the exact integration of *any variable subset* in time $\mathcal{O}(|c|)$ if (i) its input functions can be integrated efficiently and (ii) it is *smooth* and *decomposable* (Choi et al., 2020b), as formalized next.

**Definition 2** (Smoothness and decomposability (Darwiche and Marquis, 2002))**.** A circuit is *smooth* if for every sum unit $n$, all its input units depend on the same variables, i.e., $\forall i, j \in \mathsf{in}(n) : \mathsf{sc}(i) = \mathsf{sc}(j)$. A circuit is *decomposable* if the distinct inputs of every product unit $n$ depend on disjoint sets of variables, i.e., $\forall i, j \in \mathsf{in}(n) \; i \neq j : \mathsf{sc}(i) \cap \mathsf{sc}(j) = \varnothing$.

Smoothness and decomposability are related to *multilinearity*, a classical property of tensor factorizations (Kolda and Bader, 2009), as a smooth and decomposable circuit is guaranteed to encode a multilinear function (or polynomial) w.r.t. its input functions (Martens and Medabalimi, 2014; Oliver Broadrick, 2024). We will make use of another property known as determinism, which instead ensures tractable maximum-a-posteriori inference in circuits (Darwiche, 2009; Choi et al., 2020b).

**Definition 3** (Determinism (or support-decomposability) (Darwiche and Marquis, 2002; Choi et al., 2020b))**.** A sum unit $n$ is *deterministic* (or *support-decomposable*) if all its inputs have pairwise disjoint supports, i.e., $\forall i, j \in \mathsf{in}(n), i \neq j : \mathsf{supp}(i) \cap \mathsf{supp}(j) = \varnothing$, where $\mathsf{supp}(n) = \{\mathbf{x} \in \mathsf{dom}(\mathsf{sc}(n)) \mid c_n(\mathbf{x}) \neq 0\}$. A circuit is deterministic if every sum unit in it is deterministic.

Unlike the relationship between smoothness, decomposability and multilinearity, a property similar to determinism has not been explored in tensor factorization techniques. The relationship between determinism and orthogonality will be crucial in §3 to devise canonical forms for circuits.

A **probabilistic circuit** (PC) is a circuit $c$ encoding a non-negative function, thus modeling a possibly unnormalized probability distribution $p(\mathbf{x}) = Z^{-1} c(\mathbf{x})$, where $Z$ is the partition function (Choi et al., 2020b; Vergari et al., 2021). To construct and learn a circuit that is a PC, its parameters and input functions can be enforced to be non-negative, i.e., they are *monotonic* PCs (Shpilka and Yehudayoff, 2010). This in fact ensures the circuit outputs are also non-negative. However, PCs whose parameters can be negative, i.e., *non-monotonic* PCs, have been shown to be strictly more expressive models than monotonic ones (Valiant, 1979). Building and learning non-monotonic PCs flexibly while ensuring

they compute a non-negative function is in general a challenging problem (Dennis, 2016). However, a family of non-monotonic PCs can be constructed via *squaring*, as we detail next.

**Born machines and squared PCs.** As mentioned above, to model a distribution $p(\mathbf{X})$ we can take the modulus square of a complex-valued TN, resulting in a model often called Born machine (Dirac, 1930; Glasser et al., 2018). One can similarly build non-monotonic PCs by *squaring* circuits with real or complex parameters, which comes with theoretically guarantees regarding their increased expressiveness over monotonic ones (Loconte et al., 2024; 2025b). The property-driven framework of circuits precisely tells us how to build a circuit such that its squaring can be marginalized efficiently. This problem is analogous to the one of representing the multiplication of two TNs as yet another TN (Michailidis et al., 2024). Formally, given a circuit $c$, a squared PC $c^2$ encodes a distribution $p(\mathbf{x}) = Z^{-1}|c(\mathbf{x})|^2$, where $Z = \int_{\mathsf{dom}(\mathbf{X})} |c(\mathbf{x})|^2 \, \mathrm{d}\mathbf{x}$. Computing $Z$ or any marginal tractably requires representing $c^2$ as yet another decomposable circuit (Def. 2), which can be obtained by multiplying $c$ with its conjugate $c^*$. While the conjugate circuit $c^*$ can be efficiently obtained from $c$ by simply taking the conjugate of the sum parameters and input functions (Yu et al., 2023), realizing the product of any two decomposable circuits as a decomposable circuit is in general a #P-hard problem (Vergari et al., 2021). However, it can be done efficiently if these circuits are *compatible*.

**Definition 4** (Compatibility (Vergari et al., 2021))**.** Two smooth and decomposable circuits $c_1$, $c_2$ over variables $\mathbf{X}$ are *compatible* if (i) the product of any pair $f_n$, $f_m$ of input functions respectively in $c_1$, $c_2$ over the same variable can be efficiently integrated, and (ii) any pair $n$, $m$ of product units respectively in $c_1$, $c_2$ having the same scope decompose their scope over their inputs in the same way.

We say that a circuit is *structured-decomposable* if it is compatible with itself (Pipatsrisawat and Darwiche, 2008), i.e., all products having the same scope decompose it towards their inputs in the same way. As detailed in App. B, circuits corresponding to MPS and TTN TNs are structured-decomposable. If a circuit $c$ is structured-decomposable, then its products implicitly encode a tree-like partitioning of variables (Kisa et al., 2014), which ensures that the product between $c$ and $c^*$ can be encoded by a decomposable circuit of size $\mathcal{O}(|c|^2)$. This can be done via a circuit multiplication algorithm as shown in Vergari et al. (2021). The quadratic increase in circuit size is why computing the partition function or any marginal ultimately requires time $\mathcal{O}(|c|^2)$. In the next section, we address this quadratic complexity overhead in squared PCs by deriving novel structural properties.

## 3 RELAXING DETERMINISM VIA ORTHOGONALITY

By showing how both TN canonical forms and determinism in circuits bring simplifications when computing marginals, we translate the key idea of orthogonality to the language of circuits. Consider an MPS encoding $\psi$ over variables $\mathbf{X} = \{X_1, X_2\}$, i.e., $\psi(x_1, x_2) = \sum_{i=1}^R \psi_1^i(x_1)\psi_2^i(x_2)$. As an example, consider a left canonical form requiring the factors $\{\psi_1^i\}_{i=1}^R$ over $X_1$ to satisfy the orthonormality condition $\int_{\mathsf{dom}(X_1)} \psi_1^i(x_1)\psi_1^j(x_1)^* \, \mathrm{d}x_1 = \langle \psi_1^i \mid \psi_1^j \rangle = \delta_{ij}$, where $\delta_{ij}$ is the Kronecker delta. Under this condition, we can simplify the marginal $p(x_2) = \int_{\mathsf{dom}(X_1)} |\psi(x_1, x_2)|^2 \, \mathrm{d}x_1$ as

$$\sum_{i=1}^R \sum_{j=1}^R \langle \psi_1^i \mid \psi_1^j \rangle \, \psi_2^i(x_2)\psi_2^j(x_2)^* = \sum_{i=1}^R |\psi_2^i(x_2)|^2, \tag{2}$$

because the inner product $\langle \psi_1^i \mid \psi_1^j \rangle$ is zero whenever $i \neq j$. We observe that the same simplification from $\mathcal{O}(R^2)$ to $\mathcal{O}(R)$ sums would occur also if the factors $\{\psi_1^i\}_{i=1}^R$ were instead defined over non-overlapping supports, i.e., $\forall i, j \in [R], i \neq j$, at least one between $\psi_1^i$ and $\psi_1^j$ is zero. Exploiting factors having non-overlapping supports rather than being orthogonal suggests us we could use determinism in order to simplify marginalization in squared PCs. Formally, given $n$ a deterministic sum unit computing $c_n(x_1, x_2) = \sum_{i \in \mathsf{in}(n)} w_i c_i(x_1, x_2)$, we can write $|c_n(x_1, x_2)|^2$ as

$$\sum_{i \in \mathsf{in}(n)} \sum_{j \in \mathsf{in}(n)} w_i w_j^* \, c_i(x_1, x_2)c_j(x_1, x_2)^* = \sum_{i \in \mathsf{in}(n)} |w_i|^2 \, |c_i(x_1, x_2)|^2, \tag{3}$$

since due to determinism at least one between $c_i$ and $c_j$ is zero whenever $i \neq j$. From Eq. (3) we recover that the number of input connections to a deterministic sum unit does not quadratically increases when taking its modulus square. Therefore, by recursively applying Eq. (3) for all sum units in a deterministic circuit $c$, it turns out that a decomposable squared PC can be obtained from $c$ of the same size. For this reason, the satisfaction of determinism allows us to compute any marginal

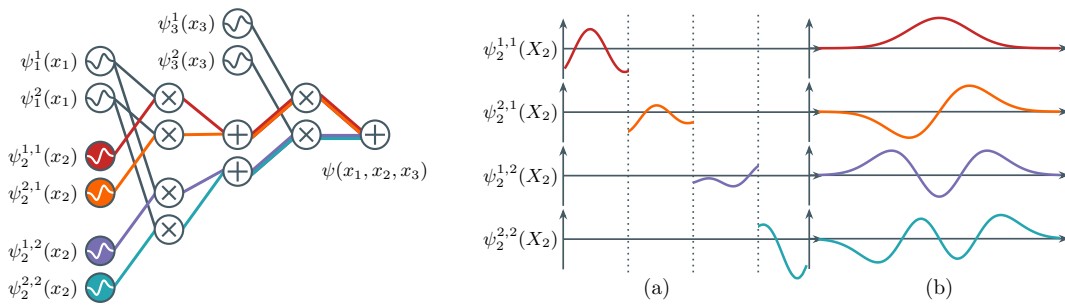

Figure 2: **Deterministic and orthogonal circuits differ by their input functions. (left)** We consider the circuit $c$ representing the MPS shown in Fig. 1, and we color each input function $\psi_2^{i_1,i_2}$ over the variable $X_2$ differently. Each sum unit is basis decomposable, as it partitions the sets of input functions over $X_2$ towards its inputs (see how colored edges are split at sum units). **(right)** If we take input functions over $X_2$ having non-overlapping support (a), we recover determinism in $c$. Instead, if the input functions are orthogonal yet having the same support (b), then $c$ is orthogonal.

in the squared PC in time $\mathcal{O}(|c|)$ rather than $\mathcal{O}(|c|^2)$. However, the caveat is that taking the modulus square of a deterministic circuit can be done by simply replacing each weight and input function with their modulus square, resulting in a PC with non-negative activations only (e.g., see the $|w_i|^2$ in Eq. (3)). As such, real or complex parameters would not bring any expressiveness advantage over monotonic PCs, as also noticed in Loconte et al. (2024, Prop. 4). This begs the question: *How can we parameterize squared PCs to overcome their computational overhead without requiring determinism?* Inspired by the simplification in Eq. (2), we introduce a relaxation of determinism called *orthogonality*, requiring sum units to receive input from units computing orthogonal functions.

**Definition 5** (Orthogonality (or ortho-decomposability)). *A smooth sum unit $n$ with $\mathsf{sc}(n) = \mathbf{Z}$ is orthogonal if all pairs of distinct inputs encode orthogonal functions, i.e., $\forall i, j \in \mathsf{in}(n), i \neq j$: $\int_{\mathsf{dom}(\mathbf{Z})} c_i(\mathbf{z}) c_j(\mathbf{z})^* \, \mathrm{d}\mathbf{z} = 0$. A circuit is orthogonal if all sum units in it are orthogonal.*

Unlike determinism, orthogonality does not necessarily require the inputs to sum units to have disjoint support. As we formalize in App. A.2, orthogonality strictly generalizes determinism in the case of non-monotonic circuits. Similar to our discussion above leveraging determinism, given a circuit $c$ that is orthogonal we have that computing the partition function of its modulus square can be done in time $\mathcal{O}(|c|)$ rather than $\mathcal{O}(|c|^2)$. We formalize this result in the following theorem.

**Theorem 1.** *Let $c$ be a smooth, decomposable and orthogonal circuit over $\mathbf{X}$. Then computing the partition function $Z = \int_{\mathsf{dom}(\mathbf{X})} |c(\mathbf{x})|^2 \, \mathrm{d}\mathbf{x}$ can be done in time $\mathcal{O}(|c|)$.*

We prove it in App. A.1 where we also observe that, unlike determinism, orthogonality allows us to retain real or complex parameters in the modeled distribution representation. To prove Thm. 1 we actually introduce a generalization of orthogonality—called $\mathbf{Z}$-*orthogonality*—that considers sum units having scope overlapping with $\mathbf{Z} \subseteq \mathbf{X}$ and allows us to compute the more general quantity $\int_{\mathsf{dom}(\mathbf{Z})} |c(\mathbf{y}, \mathbf{z})|^2 \, \mathrm{d}\mathbf{z}$ in time $\mathcal{O}(|c|)$, where $\mathbf{y} \in \mathsf{dom}(\mathbf{X} \setminus \mathbf{Z})$. Moreover, as detailed in App. A.4, if a circuit is $\{X\}$-orthogonal for all $X \in \mathbf{X}$, then it is $\mathbf{Z}$-orthogonal for any $\mathbf{Z} \subseteq \mathbf{X}$. Therefore, this other result also shows a condition to marginalize *any* subset $\mathbf{Z}$ of variables in time $\mathcal{O}(|c|)$.

**Unlocking non-structured-decomposable squared PCs.** One aspect of Thm. 1 is that, under orthogonality, computing the partition function requires linear time even for squared PCs that are *not* structured-decomposable. This is perhaps surprising, because integrating the power of a non-deterministic and non-structured-decomposable circuit is in general #P-hard, see Vergari et al. (2021, Thm. 3.3). The key ingredient to overcome marginalization being #P-hard is exploiting cancellations provided by orthogonality to avoid integrating the product of non-compatible circuits, which would be otherwise intractable. To the best of our knowledge, TN structures corresponding to non-structured-decomposable circuits have not been previously investigated. E.g., MPS and TTNs implicitly encode a single hierarchical partitioning of the variables (Grasedyck, 2010), thus being structured-decomposable circuits (see App. B). Furthermore, theoretical results link structured-decomposability to a decrease of expressiveness in circuits and squared PCs (Pipatsrisawat and Darwiche, 2008; 2010; de Colnet and Mengel, 2021; Loconte et al., 2025b). Although our experiments on a particular class

of non-structured-decomposable squared PCs do not show an expressiveness increase (§6), these works motivate future research to develop novel expressive factorizations that are *not* structured-decomposable, yet their modulus squaring enable efficient marginalization via orthogonality.

### 3.1 How to Build Orthogonal Circuits

Peharz et al. (2014) showed one can build a deterministic circuit by (i) choosing the input functions over the same variable such that they have disjoint supports, and (ii) ensuring each sum has inputs that are connected to different input functions in the circuit graph. Each sum unit in a deterministic circuit built in this way—also called *regular selective*—acts like a decision node for the input functions it depends on w.r.t. a variable. This construction can be done recursively (Lowd and Rooshenas, 2013; Shih and Ermon, 2020). To construct circuits that are orthogonal we can use a similar approach, where each sum unit implicitly selects a subset of input functions that are however orthogonal rather than have non-overlapping supports. We start by formalizing the concept of a sum unit acting like a decision node for the input functions it depends on, which we call *basis decomposability*.

**Definition 6** (Basis decomposability). A smooth sum unit $n$ is *basis decomposable* if the inputs to $n$ depend on non-overlapping input functions for a variable, i.e., $\exists X \in \mathsf{sc}(n), \forall i, j \in \mathsf{in}(n), i \neq j : \mathcal{B}_X(i) \cap \mathcal{B}_X(j) = \varnothing$, where $\mathcal{B}_X(i)$ denotes the set of input functions over $X$ in the sub-circuit rooted in the unit $i$. A circuit is basis decomposable if every sum unit in it is basis decomposable.

By requiring basis decomposability and that the input functions over the same variable are orthogonal with each other, we recover the class of *regular orthogonal* circuits that are guaranteed to be *orthogonal*. We formally show this in App. A.3 and define regular orthogonality below.

**Definition 7** (Regular orthogonality). A smooth and decomposable circuit $c$ over $\mathbf{X}$ is *regular orthogonal* if (i) it is basis decomposable, and (ii) if all input units over the same variable $X \in \mathbf{X}$ encode orthogonal functions, i.e., $\forall i, j$ input units over $X, i \neq j$, we have that $\int_{\mathsf{dom}(X)} c_i(x) c_j(x)^* \, \mathrm{d}x = 0$.

Fig. 2 illustrates examples of regular selective and regular orthogonal circuits as basis decomposable circuits having the same computational graph but differing by their input functions. Furthermore, Apps. A.3 and A.4 also generalize regular orthogonality and present sufficient conditions for $\mathbf{Z}$-orthogonality (§3) enabling linear-time marginalization.

However, regular orthogonality is rather restrictive as it requires each input to a sum unit to depend on different input functions. Fig. 3a depicts this, where we assume that differently colored inner units depend on different input functions. Instead, circuits made of densely-connected layers of sums and products can have inputs to a sum that share the same sets of input functions (see Fig. 3b), thus not being basis decomposable. This kind of circuits

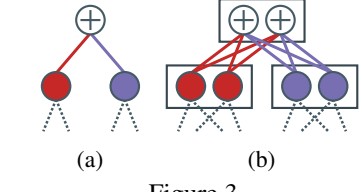

(a)          (b)

Figure 3

include popular TN structures such as TTNs (Shi et al., 2006; Cheng et al., 2019) (see Fig. B.1), as well as circuit architectures that benefit from GPU parallelism (Vergari et al., 2019; Peharz et al., 2020b;a; Liu and Van den Broeck, 2021; Mari et al., 2023; Loconte et al., 2025a; Zhang et al., 2025a). These circuits—called *tensorized circuits*—motivates us in finding different conditions relaxing basis decomposability, yet still useful to simplify the computation of marginals. We show how grouping computational units into layers enables us to define such conditions. Similarly to a canonical form, these conditions are based on orthonormal input functions and (semi-)unitary matrices, and ensure squared PCs encode already-normalized distributions. However, our construction can be applied to circuits, even those that do not map to any known tensor factorization method.

## 4 From Regular Orthogonality to Unitarity

We introduce properties similar to regular orthogonality but instead defined over layers in order to fit tensorized circuit architectures that would otherwise not be basis decomposable due to their densely-connected structure. These properties guarantee that a tensorized circuit is orthogonal and that the squared PC obtained from it encodes an already-normalized distribution, while also providing speed-ups for the computation of any marginal (§5). These perks generalize to tensorized circuit architectures that are *not* structured-decomposable, thus representing a *strictly* larger set of factorizations when compared to TTNs (as noticed for orthogonality in §3). We formalize tensorized circuits below.

**Definition 8** (Tensorized circuit (Loconte et al., 2025a)). A *tensorized circuit* $c$ is a parameterized computational graph encoding a function $c(\mathbf{X})$ and comprising of three kinds of layers: *input*, *product* and *sum*. A layer $\boldsymbol{\ell}$ is a vector-valued function defined over variables $\mathsf{sc}(\boldsymbol{\ell}) \subseteq \mathbf{X}$, called *scope*, and every non-input layer receives the outputs of other layers as input, denoted as $\mathsf{in}(\boldsymbol{\ell})$. The scope of each non-input layer is the union of the scope of its inputs. The three kinds of layers are defined as:

- Each *input layer* $\boldsymbol{\ell}$ has scope $X \in \mathbf{X}$ and computes a collection of $K$ input functions $\{f_i \colon \mathsf{dom}(X) \to \mathbb{C}\}_{i=1}^{K}$, i.e., $\boldsymbol{\ell}$ outputs a $K$-dimensional vector.
- Each *product layer* $\boldsymbol{\ell}$ computes either an element-wise (or Hadamard) or Kronecker product of its $N$ inputs, i.e., $\odot_{i=1}^{N}\boldsymbol{\ell}_i(\mathsf{sc}(\boldsymbol{\ell}_i))$ or $\otimes_{i=1}^{N}\boldsymbol{\ell}_i(\mathsf{sc}(\boldsymbol{\ell}_i))$, respectively.
- A *sum layer* $\boldsymbol{\ell}$ receiving input from $\{\boldsymbol{\ell}_i\}_{i=1}^{N}$ computes the matrix-vector product $\mathbf{W} \cdot [\boldsymbol{\ell}_1(\mathsf{sc}(\boldsymbol{\ell}_1)) \cdots \boldsymbol{\ell}_N(\mathsf{sc}(\boldsymbol{\ell}_N))]$, where $\mathbf{W} \in \mathbb{C}^{K_1 \times K_2}$ and $[\,\cdot\,]$ is the concatenation operation.

As we illustrate in Figs. 4 and B.1, tensorized circuits can be seen as "syntactic sugar" for circuits (Def. 1) having dense connections between sparse groups of units. That is, a sum layer parameterized by $\mathbf{W} \in \mathbb{C}^{K_1 \times K_2}$ consists of $K_1$ sum units each receiving $K_2$ inputs and parameterized by a row in $\mathbf{W}$. Similarly, a product layer consists of scalar product units. Furthermore, we refer to the **size** of a layer $\boldsymbol{\ell}$ as the total number of input connections to the units inside $\boldsymbol{\ell}$. As detailed in App. A.5, there exists a squaring algorithm for tensorized circuits operating on layers and using linear algebra operations. This squaring algorithm extends another one described in Loconte et al. (2024) to support circuits whose sum layers can receive input from more than one layer (as in Def. 8).

Similar to regular orthogonality, we start by requiring that the input units over the same variable encode a collection of *orthonormal* functions, as we formalize below.

**(U1)** Each input layer $\boldsymbol{\ell}$ over a variable $X$ encodes $K$ orthonormal functions, i.e., $\boldsymbol{\ell}(X) = [f_1(X) \cdots f_K(X)]^{\top}$ such that $\forall i, j \in [K] \colon \int_{\mathsf{dom}(X)} f_i(x) f_j(x)^{*}\, \mathrm{d}x = \delta_{ij}$. For any pair of input layers $\boldsymbol{\ell}_i, \boldsymbol{\ell}_j$ over $X$ and with $\boldsymbol{\ell}_i \neq \boldsymbol{\ell}_j$, we have that $\int_{\mathsf{dom}(X)} \boldsymbol{\ell}_i(x) \otimes \boldsymbol{\ell}_j(x)^{*}\, \mathrm{d}x = \mathbf{0}$.

App. C reviews possible choices for flexible orthonormal input functions. To relax basis decomposability, the following property requires that sum layers receive input from layers depending on different input layers for at least one variable. E.g., the sum layer in Fig. 3b receives input from two other layers (with red and violet units), each depending on different input functions. Still, the sums in a sum layer receive input from units that share the input functions, thus not being basis decomposable in general (e.g., in Fig. 3b each sum receives input from *two* red/violet units).

**(U2)** Each sum layer $\boldsymbol{\ell}$ receives inputs from layers $\{\boldsymbol{\ell}_i\}_{i=1}^{N}$ such that $\exists X \in \mathsf{sc}(\boldsymbol{\ell})$, $\forall i, j \in [N]$, $i \neq j$: the sub-circuits rooted in $\boldsymbol{\ell}_i$ and $\boldsymbol{\ell}_j$ dot not share input layers over the variable $X$.

In the particular case of a circuit where each layer consists of exactly one unit, (U2) is equivalent to basis decomposability. Finally, each sum layer has to be parameterized by a (semi-)unitary matrix.

**(U3)** Each sum layer is parameterized by a (semi-)unitary matrix $\mathbf{W} \in \mathbb{C}^{K_1 \times K_2}$, $K_1 \leq K_2$, i.e., $\mathbf{W}\mathbf{W}^{\dagger} = \mathbf{I}_{K_1}$ or, equivalently, the rows of $\mathbf{W}$ are orthonormal.

If a tensorized circuit satisfies **(U1-3)** above, we say it is **unitary**. As formalized below and as we prove in App. A.6, by exploiting cancellations provided by the orthonormality of input functions and weights, the modulus squaring of a unitarity circuit encodes a normalized distribution.

**Theorem 2.** Let $c$ be smooth and decomposable circuit over variables $\mathbf{X}$. If $c$ is unitary, i.e., if it satisfies conditions **(U1-3)**, then we have that $c$ is orthogonal and $Z = \int_{\mathsf{dom}(\mathbf{X})} |c(\mathbf{x})|^2\, \mathrm{d}\mathbf{x} = 1$.

This is similar in spirit to Born machines obtained as the modulus square of a TTN in a convenient canonical form—also called *upper-canonical form* (Cheng et al., 2019)—ensuring normalization by exploiting (semi-)unitary matrices. As detailed in App. B.1, our unitarity conditions strictly generalize such canonical form to more general tensorized circuits. This is because we can build unitary circuits that are *not* structured-decomposable, i.e., whose structure encodes multiple hierarchical variables partitionings (see Fig. 4), yet their squaring encode a normalized distribution as for Thm. 2.

**Expressiveness analysis.** Since we introduced new families of circuits based on orthogonality and unitarity properties, in Apps. A.8 and A.9 we provide a preliminary analysis regarding their *expressive efficiency*, i.e., the ability of a circuit class to encode a function in a polysize computational graph. This contributes to a number of works investigating the expressiveness of circuits and squared PCs (Darwiche and Marquis, 2002; Martens and Medabalimi, 2014; de Colnet and Mengel, 2021; Glasser

Figure 4: **Tensorized circuits can encode custom hierarchical factorizations** with no corresponding TTN. The shown circuit encodes a factorization over $\mathbf{X}$ using a mix of Hadamard and Kronecker product layers and two input layers per variable in $\mathbf{X} = \{X_1, X_2, X_3\}$, Unlike the TTN in Fig. B.1, this circuit is *not* structured-decomposable since there are product units that factorize their scope $\mathbf{X}$ differently (pointed by arrows): $\{\{X_3\}, \{X_1, X_2\}\}$ and $\{\{X_1, X_3\}, \{X_2\}\}$, as indicated with the color stripes, each corresponding to a dependency w.r.t. a particular variable. Remarkably, this circuit does satisfy properties (U2) and (U4), as the pointed product layers that are input to the root sum layer do not share input layers.

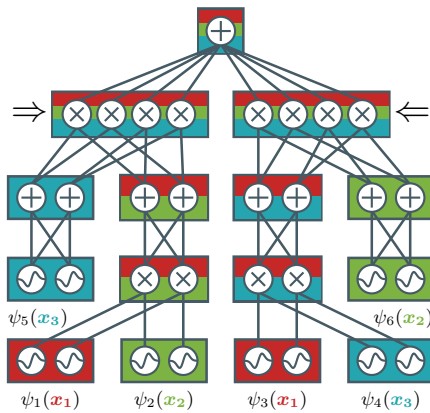

et al., 2019; Loconte et al., 2025b). In App. A.8 we firstly show that enforcing orthogonality in a smooth and decomposable circuit is #P-hard, thus suggesting future work looking at whether some functions encoded by smooth and decomposable circuits cannot be encoded by polysize orthogonal ones. We conjecture this to hold similarly to the case of deterministic circuits (Bova et al., 2016). Instead, in App. A.9 we show that enforcing (semi-)unitary weights in sum layers can be done in polytime, thus guaranteeing no loss in terms of expressive efficiency. Our experiments in §6 confirm this, showing one can learn unitary squared PCs that perform similarly to non-unitary ones, while App. D.1 shows that Fourier input functions are competitive w.r.t. Gaussians for density estimation.

## 5 A Tighter Complexity for Variable Marginalization

As shown in Loconte et al. (2024) and discussed in App. A.5, computing any marginal probability in a PC obtained by taking the modulus square of a structured-decomposable and tensorized circuit $c$ requires time $\mathcal{O}(L^2 S_{\max}^2)$, where $L$ is the number of layers and $S_{\max}$ is an upper-bound to the size of each layer in $c$. In fact, the marginalization algorithm would (i) represent the modulus squaring of $c$ as another decomposable circuit, thus quadratically increasing its size $\mathcal{O}(L S_{\max})$, and (ii) compute a marginal with a single forward-pass over the squared PC. Here, we not only present an algorithm to compute any marginal that can be much more efficient, but we also show it generalizes to non-structured-decomposable squared PCs as well. We do so by exploiting the unitarity conditions (U1-3).

Our idea is that, when computing marginal probabilities we do *not* need to evaluate the layers whose scope depends on only the variables being integrated out, because they simplify to identity matrices (see proof of Thm. 2). We also observe that we do not need to square the whole tensorized circuit $c$, but only a fraction of the layers depending on *both* the marginalized variables and the ones being left over. By doing so, a part of the complexity will ultimately depend on $S_{\max}$ rather than $S_{\max}^2$. However, in order to be able to marginalize *any variable subset*, we need to specialize (U2) in unitarity by requiring that the layers that are input to a sum layer have disjoint input functions dependencies w.r.t. *all* variables. As formalized below, we simply need to change "$\exists X$" to "$\forall X$" in (U2).

**(U4)** Each sum layer $\ell$ receives inputs from layers $\{\ell_i\}_{i=1}^N$ such that $\forall X \in \mathsf{sc}(\ell)$, $\forall i, j \in [N]$, $i \neq j$: the sub-circuits rooted in $\ell_i$ and $\ell_j$ dot not share input layers over the variable $X$.

For instance, the non-structured-decomposable circuit in Fig. 4 satisfies (U4). Thanks to (U4), we can ignore the integration of all pairwise products between multiple inputs to a sum layer, as they annihilate thanks to orthogonality. This makes our complexity ultimately depend on $\mathcal{O}(L)$ rather than $\mathcal{O}(L^2)$. Alg. A.3 presents our marginalization algorithm and, as we show in App. A.7, the following theorem guarantees it is correct also in the case of non-structured tensorized circuits.

**Theorem 3.** Let $c$ be a smooth and decomposable circuit over $\mathbf{X}$ that satisfies **(U1-4)**, and let $\mathbf{Z} \subseteq \mathbf{X}$, $\mathbf{Y} = \mathbf{X} \setminus \mathbf{Z}$. Computing the marginal $p(\mathbf{y}) = \int_{\mathsf{dom}(\mathbf{Z})} |c(\mathbf{y}, \mathbf{z})|^2 \, \mathrm{d}\mathbf{z}$ requires time $\mathcal{O}(|\phi_\mathbf{Y} \setminus \phi_\mathbf{Z}| S_{\max} + |\phi_\mathbf{Y} \cap \phi_\mathbf{Z}| S_{\max}^2)$, where $\phi_\star$ is the set of layers whose scope depends on at least one variable in $\star$.

Moreover, App. A.7 details why our algorithm has complexity $\mathcal{O}(|\phi_\mathbf{Y} \setminus \phi_\mathbf{Z}| S_{\max})$ in the best case. **The relationship with TN canonical forms.** To speed-up the computation of a certain marginal with a Born machine, we firstly need to adapt the TN parameters into a canonical form that is specific for

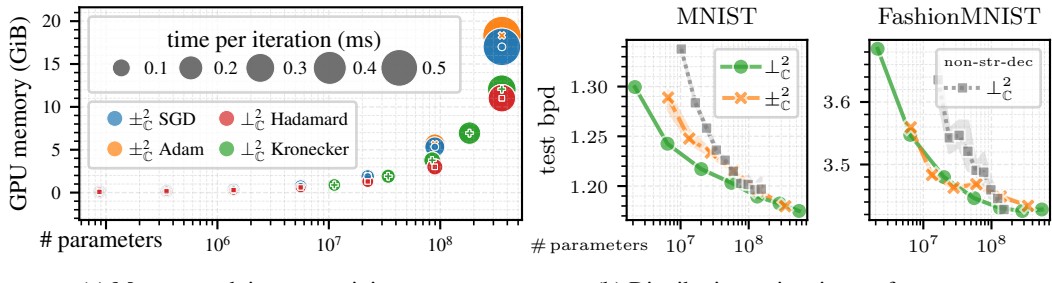

(a) Memory and time per training step.  (b) Distribution estimation performances.

Figure 5: **Squared unitary PCs scale better than squared PCs while retaining performance.** By virtue of not materializing their squares as yet another circuit to compute the partition function $Z$, squared unitary PCs result in more efficient learning, even when using Kronecker product layers **(a)**. This is, in practice, without any sacrifice in model performance, as we observe on the bits-per-dimension (bpd, lower is better) on image datasets **(b)**. Remarkably, our parametrization allows efficiently training squared non-structured-decomposable PCs (gray lines).

the chosen marginal (Vidal, 2003; Bonnevie and Schmidt, 2021). This comes with additional overhead depending on the number of variables and the TN shape. Instead, our marginalization algorithm does not require us to change the parameters depending on the marginal being computed, yet it provides a speed-up when compared to the naive approach that materializes a squared PC as a decomposable one. Moreover, our algorithm generalizes over non-structured-decomposable factorizations when represented as unitary circuits, which otherwise would *not* support tractable marginalization (§2).

## 6   EMPIRICAL EVALUATION

We now assess the practical benefits of using unitary squared PCs. We remark that our aim in this section is not to achieve state-of-the-art results, but rather to answer the following research questions comparing squared PCs with and without our unitary parameterization. That is, we investigate whether unitary circuits result in faster and lighter squared PCs (**RQ1**); if we can learn unitary squared PCs without sacrificing model performance in distribution estimation tasks (**RQ2**); and whether we can, for the first time, efficiently learn squared PCs that are *not* structured-decomposable (**RQ3**). All the details, hyperparameters and additional results can be found in App. D.

**Experimental setting.** Given a training dataset $\mathcal{D}$ on variables $\mathbf{X}$, we aim at finding the parameters of a given squared PC that maximize the data likelihood. We follow Loconte et al. (2024) and, given a batch $\mathcal{B} \subset \mathcal{D}$, write its negative log-likelihood as $\mathcal{L} := |\mathcal{B}| \log Z - \sum_{\mathbf{x} \in \mathcal{B}} 2 \log |c(\mathbf{x})|$, such that we just need to materialize the squared PC once per batch to compute the log-partition function, significantly speeding up computations. For every circuit we use complex-valued parameters, identical architectures and batch sizes, and report results on a test dataset of the model with best validation performance during training. Similar to Loconte et al. (2024), we employ Hadamard product layers for the baseline squared PCs, denoted as $\pm_{\mathbb{C}}^2$, while we use Kronecker product layers for the squared unitary PCs, $\perp_{\mathbb{C}}^2$, as we found them to perform significantly better in practice (see Apps. D.1 and D.2).

**RQ1: Improved throughput.** First, we measure to which extent squared unitary PCs improve the time and memory overhead of squared PCs that instead require computing $Z$ explicitly during training. To this end, we build circuits with increasing number of units per layer and measure the time and memory required to perform one optimization step. Fig. 5a and Tab. D.3 show that squared unitary PCs are consistently faster to evaluate and use less memory. This becomes especially clear at large scales where a squared unitary PC with Kronecker product layers and $357\,\mathrm{M}$ parameters takes $12\,\mathrm{GiB}$ of GPU memory and $0.29\,\mathrm{ms}$ per iteration, against the $18\,\mathrm{GiB}$ and $0.52\,\mathrm{ms}$ of its counterpart with Hadamard layers. Hence, *unitary circuits enable learning squared PCs with Kronecker layers at scale*, which is remarkable given that materializing the squared PCs as a decomposable one to compute $Z$ would further increase the Kronecker layer sizes dramatically (details in App. A.5).

**RQ2: Learning unitary circuits.** Next, we learn squared PCs for distribution estimation on MNIST (LeCun et al., 2010) and FashionMNIST (Xiao et al., 2017) images. We choose these benchmarks on distribution estimation over images because it is arguably the most common evaluation

setting appearing in the probabilistic circuits community (Liu et al., 2023; Gala et al., 2024b; Garg et al., 2025). In addition, due to the high-dimensionality of the data these benchmarks allow us to scale PCs up to 1B of parameters, hence making optimizing over the Stiefel manifold particularly challenging. Nevertheless, Fig. 5b shows the bits-per-dimension of squared PCs of increasing sizes on both datasets, where we can observe that *squared unitary PCs gracefully scale, matching the performance of their baseline counterparts*. To this end, we adapted the LandingSGD optimizer (Ablin and Peyré, 2022; Ablin et al., 2024) to our setting, which we describe in App. E and can be of independent interest to the community. We stress that this is remarkable, since orthogonally-constrained optimization is notoriously challenging and in *active development* (Ablin et al., 2024; Kochurov et al., 2020; Lezcano Casado, 2019) and therefore, further advancements in the field can only benefit squared unitary PCs.

**RQ3: Non-structured-decomposable squared unitary PCs.** Lastly, we reuse the previous setup and learn a squared unitary PC whose architecture is *not* structured-decomposable. and hence materializing its square as a decomposable circuit to compute $Z$ explicitly is generally not tractable (Vergari et al., 2021; Zhang et al., 2025b). We detail our construction of a non-structured-decomposable circuit in App. D.2. As discussed in §3, this circuit architecture differs substantially from MPS and TTNs factorizations, as they correspond to structured-decomposable circuits. Fig. 5b shows that such PCs can be competitive with their structured-decomposable counterparts, especially at large scales, but might be harder to learn Thus, our theory and preliminary experiments open future interesting venues to design and learn better non-structured-decomposable PCs and TNs, which can be exponentially more expressive than structured ones (Pipatsrisawat and Darwiche, 2008; 2010; de Colnet and Mengel, 2021).

## 7    CONCLUSION AND FUTURE WORK

Inspired by determinism in circuits and canonical forms in TNs, we introduced novel conditions described in the circuit framework to simplify the computation of marginals in squared PCs and ensure they encode already-normalized distributions. As for the close connection between circuits and TNs, our conditions motivate research aimed at exploring new factorization structures that can be more expressive, yet enabling exact and efficient marginalization and sampling. Recently, determinism has been generalized as a property between two circuits in Wang et al. (2024) to bring complexity simplifications for exact causal inference and weighted model counting (Chavira and Darwiche, 2008). We believe one can extend our orthogonality (§3) as a property between two circuits similarly, thus possibly simplifying the computation of compositional operations while being less restrictive than determinism. Finally, as we detail in App. C, there are a number of directions aimed at understanding the relative expressiveness of the proposed circuit families w.r.t. other classes of PCs, e.g., the recent *positive unital circuits* generalizing squared PCs shown in Zuidberg Dos Martires (2025).

## REPRODUCIBILITY STATEMENT

To ensure reproducibility, we release our code and necessary scripts here. The repository contains the models implementation, scripts for the experiments and results, and instructions for setting up the environment. Furthermore, a detailed description of all experimental settings is included in App. D.

## ACKNOWLEDGMENTS

We acknowledge Raul Garcia-Patron Sanchez for meaningful discussions about tensor networks and quantum circuits. AV and AJ are supported by the "UNREAL: Unified Reasoning Layer for Trustworthy ML" project (EP/Y023838/1) selected by the ERC and funded by UKRI EPSRC.

## CONTRIBUTIONS

LL and AV conceived the initial idea of the paper. LL is responsible for all theoretical contributions, circuit illustrations, and algorithms related to the proofs in App. A. LL and AJ are responsible for the implementation. AJ developed the LandingPC algorithm, run the experiments and plotted the results. LL led the writing, with the exception of the experimental section led by AJ, and all authors provided feedback about the paper. AV supervised all phases of the project.

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

# Appendix

## Table of Contents

## A  PROOFS

**Assumptions.** Below we implicitly make the following mild assumptions. We require each inner unit to compute a Lebesgue-integrable function over its support. We also assume that the functions computed by input units can be evaluated and integrated over their support efficiently. With a slight abuse of notation, we use integrals to actually denote summations if taken w.r.t. discrete variables.

### A.1  LINEAR-TIME PARTITION FUNCTION AND MARGINALS COMPUTATION VIA ORTHOGONALITY

In order to prove that computing the partition function of a squared PC, obtained by taking the modulus square of an orthogonal circuit $c$ (Def. 5), requires time $\mathcal{O}(|c|)$ (i.e., our Thm. 1), here we firstly introduce a generalization of orthogonality. This other condition, which we call $\mathbf{Z}$-orthogonality, considers only sum units having scope overlapping with $\mathbf{Z}$ and requires the inputs to sum units to encode orthogonal functions when variables that are *not* in the variables set $\mathbf{Z}$ are kept fixed. We formalize $\mathbf{Z}$-orthogonality below.

**Definition A.1** ($\mathbf{Z}$-orthogonality). A smooth sum unit $n$ is $\mathbf{Z}$-*orthogonal*, with $\hat{\mathbf{Z}} = \mathsf{sc}(n) \cap \mathbf{Z} \neq \varnothing$, if all pairs of its inputs encode orthogonal functions when fixing the variables in $\mathsf{sc}(n) \setminus \hat{\mathbf{Z}}$, i.e., $\forall i, j \in \mathsf{in}(n), i \neq j\colon \int_{\mathsf{dom}(\hat{\mathbf{Z}})} c_i(\mathbf{y}, \hat{\mathbf{z}}) c_j(\mathbf{y}, \hat{\mathbf{z}})^* \, \mathrm{d}\hat{\mathbf{z}} = 0$, for any $\mathbf{y} \in \mathsf{dom}(\mathsf{sc}(n) \setminus \hat{\mathbf{Z}})$. Moreover, we say a circuit over variables $\mathbf{X}$ is $\mathbf{Z}$-orthogonal, with $\mathbf{Z} \subseteq \mathbf{X}$, if all sum units having scope overlapping with $\mathbf{Z}$ are $\mathbf{Z}$-orthogonal.

Under the satisfaction of $\mathbf{Z}$-orthogonality in $c$, the following lemma shows that computing the quantity $\int_{\mathsf{dom}(\mathbf{Z})} |c(\mathbf{y}, \mathbf{z})|^2 \, \mathrm{d}\mathbf{z}$ can be done in time $\mathcal{O}(|c|)$, where $\mathbf{y} \in \mathsf{dom}(\mathbf{X} \setminus \mathbf{Z})$. In particular, we show the correctness and complexity of our Alg. A.1 to compute this quantity. We will later show that, by setting $\mathbf{Z} = \mathbf{X}$, one recovers Thm. 1, i.e., computing the partition function $Z = \int_{\mathsf{dom}(\mathbf{X})} |c(\mathbf{x})|^2 \, \mathrm{d}\mathbf{x}$ can be done in time $\mathcal{O}(|c|)$.

**Lemma A.1.** Let $c$ be a smooth, decomposable and $\mathbf{Z}$-orthogonal circuit over variables $\mathbf{X}$. Then computing $\int_{\mathsf{dom}(\mathbf{Z})} |c(\mathbf{y}, \mathbf{z})|^2 \, \mathrm{d}\mathbf{z}$ can be done in time $\mathcal{O}(|c|)$, where $\mathbf{y} \in \mathsf{dom}(\mathbf{X} \setminus \mathbf{Z})$.

*Proof.* We prove it by showing the correctness of Alg. A.1 via induction on the structure of $c$.

**Units having scope not overlapping with Z.** Let $n$ be a unit in $c$. In the case of $\mathsf{sc}(n) \cap \mathbf{Z} = \varnothing$, we have that we can compute $|c_n(\hat{\mathbf{y}})|^2$, for some assignments $\hat{\mathbf{y}}$ obtained from $\mathbf{y}$ by restriction over variables in $\mathsf{sc}(n)$, in time $\mathcal{O}(|c|)$. This is because we can evaluate $c_n$ by doing a feed-forward evaluation of the sub-circuit rooted in $c$, and then take the modulus square of the result. This case as formalized in L1-3 in Alg. A.1.

**Sum units.** Let $n$ be a sum unit in $c$ such that $\hat{\mathbf{Z}} = \mathsf{sc}(n) \cap \mathbf{Z} \neq \varnothing$, i.e., the variables scope of $n$ overlaps with $\mathbf{Z}$. Thus, assume that $n$ computes $c_n(\mathbf{y}, \hat{\mathbf{z}}) = \sum_{i \in \mathsf{in}(n)} w_{n,i} c_i(\mathbf{y}, \hat{\mathbf{z}})$, where $\hat{\mathbf{z}} \in \mathsf{dom}(\hat{\mathbf{Z}})$ and $\mathbf{y} \in \mathsf{dom}(\mathbf{Y})$ with $\mathbf{Y} = \mathsf{sc}(n) \setminus \hat{\mathbf{Z}}$. By hypothesis $n$ is $\mathbf{Z}$-orthogonal, and therefore we have that $\forall i, j \in \mathsf{in}(n), i \neq j\colon \int_{\mathsf{dom}(\hat{\mathbf{Z}})} c_i(\mathbf{y}, \hat{\mathbf{z}}) c_j(\mathbf{y}, \hat{\mathbf{z}})^* \, \mathrm{d}\hat{\mathbf{z}} = 0$. For this reason, we can write

$$\int_{\mathsf{dom}(\hat{\mathbf{Z}})} |c_n(\mathbf{y}, \hat{\mathbf{z}})|^2 \, \mathrm{d}\hat{\mathbf{z}} = \sum_{i \in \mathsf{in}(n)} \sum_{j \in \mathsf{in}(n)} w_{n,i} w_{n,j}^* \int_{\mathsf{dom}(\hat{\mathbf{Z}})} c_i(\mathbf{y}, \hat{\mathbf{z}}) c_j(\mathbf{y}, \hat{\mathbf{z}})^* \, \mathrm{d}\hat{\mathbf{z}}$$

$$= \sum_{i \in \mathsf{in}(n)} |w_{n,i}|^2 \int_{\mathsf{dom}(\hat{\mathbf{Z}})} |c_i(\mathbf{y}, \hat{\mathbf{z}})|^2 \, \mathrm{d}\hat{\mathbf{z}}.$$

Thus, we can compute the integral of the modulus squaring of $n$ by firstly evaluating the integral of the modulus squaring of its inputs and then computing a weighted summation. This is L4-7 in Alg. A.1. By inductive hypothesis, computing the $\int_{\mathsf{dom}(\hat{\mathbf{Z}})} |c_i(\mathbf{y}, \hat{\mathbf{z}})|^2 \, \mathrm{d}\hat{\mathbf{z}}$ in our algorithm requires time $\mathcal{O}(|c|)$ and therefore evaluating $\int_{\mathsf{dom}(\hat{\mathbf{Z}})} |c_n(\mathbf{y}, \hat{\mathbf{z}})|^2 \, \mathrm{d}\hat{\mathbf{z}}$ also requires time $\mathcal{O}(|c|)$. Furthermore, we observe that if the sub-circuits respectively rooted in $i$ and $j$, with $i, j \in \mathsf{in}(n), i \neq j$ are *not* compatible (Def. 4), then computing the integral $\int_{\mathsf{dom}(\hat{\mathbf{Z}})} c_i(\mathbf{y}, \hat{\mathbf{z}}) c_j(\mathbf{y}, \hat{\mathbf{z}})^* \, \mathrm{d}\hat{\mathbf{z}}$ would be in general a #P-hard problem (Vergari et al., 2021). In particular, $c$ would not be a structured-decomposable circuit.

However, due to cancellations arising from $\mathbf{Z}$-orthogonality, our algorithm avoids the computation of these integrals as they cancels out, thus allowing us to efficiently marginalize variables in the case of non-structured-decomposable squared PCs.

**Product units.** Let $n$ be a product unit in $c$ such that $\hat{\mathbf{Z}} = \mathsf{sc}(n) \cap \mathbf{Z} \neq \varnothing$ and computing $c_n(\mathbf{y}, \hat{\mathbf{z}}) = \prod_{i \in \mathsf{in}(n)} c_i(\mathbf{y}_i, \hat{\mathbf{z}}_i)$. Since $c$ is decomposable we have that $(\hat{\mathbf{Z}}_i)_{i \in \mathsf{in}(n)}$ forms a partitioning of variables $\hat{\mathbf{Z}}$ with $\hat{\mathbf{z}}_i \in \mathsf{dom}(\hat{\mathbf{Z}}_i)$, and the assignment $\mathbf{y}$ to $\mathbf{Y}$ is partitioned into assignments $(\mathbf{y}_i)_{i \in \mathsf{in}(n)}$. Thus, we can write

$$\int_{\mathsf{dom}(\hat{\mathbf{Z}})} |c_n(\mathbf{y}, \hat{\mathbf{z}})|^2 \, \mathrm{d}\hat{\mathbf{z}} = \int_{\times_{i \in \mathsf{in}(n)} \mathsf{dom}(\hat{\mathbf{Z}}_i)} \left( \prod_{i \in \mathsf{in}(n)} |c_i(\mathbf{y}_i, \hat{\mathbf{z}}_i)|^2 \right) \mathrm{d}\hat{\mathbf{z}}_1 \cdots \mathrm{d}\hat{\mathbf{z}}_{|\mathsf{in}(n)|}$$

$$= \prod_{i \in \mathsf{in}(n)} \int_{\mathsf{dom}(\hat{\mathbf{Z}}_i)} |c_i(\mathbf{y}_i, \hat{\mathbf{z}}_i)|^2 \, \mathrm{d}\hat{\mathbf{z}}_i.$$

Thus, similar to the case of $n$ being a sum unit above, we have that computing the integral of the modulus squaring of $n$ translates to multiplying the integrals of the modulus squaring of their inputs. Note that with a slight abuse of notation we allow $\hat{\mathbf{Z}}_i$ to be possibly empty for some $i \in \mathsf{in}(n)$. This allows us to recursively call our Alg. A.1 to the inputs of $n$, thus yielding L8-12 in it. Again, by inductive hypothesis we have that this case requires time $\mathcal{O}(|c|)$.

Consider the base case where $n$ is an input unit over $X \in \mathbf{Z}$ and computing $f(X)$, i.e., $c_n(X) = f(X)$. By assuming that the modulus squaring of $f$ can be integrated efficiently, we have that computing $\int_{\mathsf{dom}(X)} |c_n(x)|^2 \, \mathrm{d}x$ is efficient. Therefore, since all the cases considered above take time $\mathcal{O}(|c|)$, we have that computing $\int_{\mathsf{dom}(\mathbf{Z})} |c(\mathbf{y}, \mathbf{z})|^2 \, \mathrm{d}\mathbf{z}$ with Alg. A.1 requires time $\mathcal{O}(|c|)$. $\square$

---

**Algorithm A.1** MAR-ORTHO-DEC$(c, \mathbf{y}, \mathbf{Z})$

**Input:** A circuit $c$ over variables $\mathbf{X}$ that is $\mathbf{Z}$-orthogonal for some $\mathbf{Z} \subseteq \mathbf{X}$, and an assignment $\mathbf{y}$ to variables $\mathbf{Y} = \mathbf{X} \setminus \mathbf{Z}$. We denote as $n$ the output unit of $c$. **Output:** The value of $\int_{\mathsf{dom}(\mathbf{Z})} |c(\mathbf{y}, \mathbf{z})|^2 \, \mathrm{d}\mathbf{z}$.

1: **if** $\mathsf{sc}(n) \cap \mathbf{Z} = \varnothing$ **then**                        ▷ $n$ does not depend on the variables to marginalize
2:     **let** $\hat{\mathbf{y}}$ be the restriction of assignments $\mathbf{y}$ to variables in $\mathsf{sc}(n)$.
3:     $r \leftarrow$ EVAL-FEED-FORWARD$(n, \hat{\mathbf{y}})$
4:     **return** $|r|^2$
5: **else if** $n$ is a sum unit **then**
6:     **let** $n$ receive input from units $\mathsf{in}(n)$ and parameterized by $\{w_{n,i}\}_{i \in \mathsf{in}(n)}$
7:     **let** $r_i \leftarrow$ MAR-ORTHO-DEC$(i, \mathbf{y}, \mathbf{Z} \cap \mathsf{sc}(n))$, $\forall i \in \mathsf{in}(n)$
8:     **return** $\sum_{i \in \mathsf{in}(n)} |w_{n,i}|^2 r_i$
9: **else if** $n$ is a product unit **then**
10:     **let** $n$ receive input from units $\mathsf{in}(n)$
11:     $r_i \leftarrow$ MAR-ORTHO-DEC$(i, \mathbf{y}, \mathbf{Z} \cap \mathsf{sc}(i))$, $\forall i \in \mathsf{in}(n)$
12:     **return** $\prod_{i \in \mathsf{in}(n)} r_i$
13: **else**
14:     **let** $n$ be an input unit over a variable $X \in \mathbf{Z}$
15:     **return** $\int_{\mathsf{dom}(X)} |c_n(x)|^2 \, \mathrm{d}x$                        ▷ Assuming it can be computed efficiently

---

From Lem. A.1 we are now able to prove Thm. 1, as formalized below.

**Theorem 1.** Let $c$ be a smooth, decomposable and orthogonal circuit over $\mathbf{X}$. Then computing the partition function $Z = \int_{\mathsf{dom}(\mathbf{X})} |c(\mathbf{x})|^2 \, \mathrm{d}\mathbf{x}$ can be done in time $\mathcal{O}(|c|)$.

*Proof.* Since $c$ is orthogonal, then $c$ is also $\mathbf{Z}$-orthogonal with $\mathbf{Z} = \mathbf{X}$. This can be seen by noticing that $\mathsf{sc}(n) \cap \mathbf{Z} = \mathsf{sc}(n)$ for any unit $n$ and with $\mathbf{Z} = \mathbf{X}$. Therefore, from Lem. A.1 we have that computing $Z$ requires time $\mathcal{O}(|c|)$ by using Alg. A.1. $\square$

**Unlike determinism, orthogonality preserves complex parameters.** Assume that $n$ is a deterministic (Def. 3) smooth sum unit computing $c_n(\mathbf{X}) = \sum_{i \in \mathsf{in}(n)} w_{n,i} c_i(\mathbf{X})$. Then, for any $i, j \in [n]$, $i \neq j$, we have that $c_i(\mathbf{X}) c_j(\mathbf{X}) = 0$ as $i$ and $j$ have disjoint supports. Thus, we can write

$|c_n(\mathbf{X})|^2 = \sum_{i \in \mathsf{in}(n)} |w_{n,i}|^2 |c_i(\mathbf{X})|^2$. For this reason, the modulus squaring of a deterministic circuit with possibly complex parameters turns out to be equivalent to another deterministic and monotonic circuit. However, under orthogonality of $n$ instead, we cannot rewrite $|c_n(\mathbf{X})|^2$ in the same way, because cancellations only occur when integrating variables out, and the inputs to $n$ can have overlapping support (e.g., see Fig. 2). For this reason, unlike determinism we observe that orthogonality retains the possibly real or complex parameters in the distribution representation modeled as $p(\mathbf{X}) \propto |c(\mathbf{X})|^2$, which is a crucial feature aiding the expressiveness of squared PCs over monotonic ones (Loconte et al., 2024; 2025b).

## A.2 ORTHOGONALITY STRICTLY GENERALIZES DETERMINISM

In the following, we show that determinism is a sufficient but not a necessary condition for orthogonality in the case of circuits whose unit outputs can be negative or complex valued. In other words, orthogonality is a strict generalization of determinism in the case of non-monotonic circuits.

**Proposition A.1.** If a circuit $c$ is deterministic, then it is orthogonal. Under mild assumptions, the converse implication holds if $c$ is also monotonic.

*Proof.* ( $\implies$ ) By determinism of $c$ we have that, for any sum unit $n$ having scope $\mathsf{sc}(n) = \mathbf{Z}$, $\forall i, j \in \mathsf{in}(n), i \neq j \colon \mathsf{supp}(i) \cap \mathsf{supp}(j) = \varnothing$. Therefore, we have that either $c_i(\mathbf{z}) = 0$ or $c_j(\mathbf{z}) = 0$ for any $i \neq j$ and $\mathbf{z} \in \mathsf{dom}(\mathbf{Z})$. Now, if $c_i(\mathbf{z}) = 0$ then $c_i(\mathbf{z})c_j(\mathbf{z})^* = 0$, and if $c_j(\mathbf{z}) = c_j(\mathbf{z})^* = 0$ then $c_i(\mathbf{z})c_j(\mathbf{z})^* = 0$. Therefore, we have that $\forall i, j \in \mathsf{in}(n), i \neq j \colon \int_{\mathsf{dom}(\mathbf{Z})} c_i(\mathbf{z})c_j(\mathbf{z})^* \, \mathrm{d}\mathbf{z} = 0$, i.e., $n$ is orthogonal. Therefore, we conclude that $c$ is an orthogonal circuit.

( $\impliedby$ , if $c$ is also monotonic, under mild assumptions) To show the converse direction, we start by assuming that $c$ is both orthogonal and monotonic, i.e., all input functions and sum unit weights in $c$ are positive. This means that every computational unit in $c$ computes a positive function over its support. Now, by orthogonality of $c$ we have that $\forall i, j \in \mathsf{in}(n), i \neq j \colon \int_{\mathsf{dom}(\mathbf{Z})} c_i(\mathbf{z})c_j(\mathbf{z})^* \, \mathrm{d}\mathbf{z} = 0$. However, due to monotonicity we have that $c_j(\mathbf{z})^* = c_j(\mathbf{z})$ and, for orthogonality to hold, we recover that the inputs $i$ and $j$ to $n$, with $i \neq j$, must have disjoint supports, i.e., $n$ is deterministic. In other words, under monotonicity, if the supports of $i$ and $j$ were overlapping, then $c_i$ and $c_j$ would not be in general orthogonal functions as their inner product would be non-zero. Therefore, the circuit $c$ is deterministic. However, in the case of continuous variables $\mathbf{X}$, for this to hold we require an additional mild assumption over the supports of $i$ and $j$. That is, we also need that the set $\mathsf{supp}(i) \cap \mathsf{supp}(j)$ has non-zero measure. Otherwise, $\mathsf{supp}(i) \cap \mathsf{supp}(j)$ being of zero measure but non-empty (e.g., a finite set) would imply orthogonality of $c_i$ and $c_j$ yet they have overlapping supports (as the integral taken over a zero measure set is zero).

( $\not\impliedby$ , if $c$ is non-monotonic) To prove that the converse direction does *not* hold in the more general case of non-monotonic circuits, we need to find a single non-monotonic circuit that is orthogonal yet non-deterministic. This circuit can be built as a single sum unit that receives input from two real-valued orthogonal functions both having $\mathbb{R}$ as support, e.g., Hermite functions (Roman and Rota, 1978).

In conclusion, orthogonality is a strict generalization of determinism in the case of non-monotonic circuits and, under mild assumptions regarding the measure of supports intersections, determinism and orthogonality become equivalent properties in the case of monotonic circuits. $\qquad\square$

## A.3 REGULAR ORTHOGONALITY IS SUFFICIENT FOR ORTHOGONALITY

In this section, we prove that regular orthogonality (Def. 7) is sufficient for orthogonality to hold (Def. 5). For this purpose, here we firstly introduce a generalization of regular orthogonality, called $\mathbf{Z}$-regular orthogonality, and then show it is sufficient for $\mathbf{Z}$-orthogonality as defined in Def. A.1. By doing so, we present sufficient conditions based on the structure of parameterization of a circuit to marginalize a subset $\mathbf{Z}$ of variables in linear time w.r.t. the circuit size. We start by introducing the $\mathbf{Z}$-basis decomposability property, which specializes basis decomposability (Def. 6) to only sum units having scope overlapping with $\mathbf{Z}$. We start by formally introducing the concept of *basis scope*.

**Definition A.2** (Basis scope). The *basis scope* of a unit $n$ for a variable $X \in \mathsf{sc}(n)$, denoted as $\mathcal{B}_X(n)$, is the set of input unit functions over $X$ that the unit $n$ depends on, i.e., found in the sub-circuit rooted in $n$.

**Definition A.3** (**Z**-basis decomposability). A smooth sum unit $n$ is **Z**-*basis decomposable*, with $\mathsf{sc}(n) \cap \mathbf{Z} \neq \varnothing$, if the inputs to $n$ depend on non-overlapping basis scopes for a variable in **Z**, i.e., $\exists X \in \mathsf{sc}(n) \cap \mathbf{Z}, \forall i, j \in \mathsf{in}(n), i \neq j \colon \mathcal{B}_X(i) \cap \mathcal{B}_X(j) = \varnothing$. A circuit is **Z**-basis decomposable if every sum unit is **Z**-basis decomposable.

In the following, we use **Z**-basis decomposability to define the **Z**-regular orthogonal property.

**Definition A.4** (**Z**-regular orthogonality). A smooth and decomposable circuit $c$ over **X** is **Z**-*regular orthogonal*, with $\mathbf{Z} \subseteq \mathbf{X}$, if (i) it is **Z**-basis decomposable, and (ii) if all input units over the same variable $X \in \mathbf{Z}$ encode orthogonal functions, i.e., $\forall i, j$ input units over $X$, $i \neq j$, we have that $\int_{\mathsf{dom}(X)} c_i(x) c_j(x)^* \, \mathrm{d}x = 0$.

For instance, the circuit shown in Fig. 2 is $\{X_2\}$-orthogonal, since the input functions over the same variable $X_2$ are orthogonal and each sum unit having $X_2$ in their variables scope is $\{X_2\}$-basis decomposable, i.e., it splits the input functions over $X_2$ it depends on towards its inputs. Note that, given a circuit $c$ over variables **X**, we observe that **Z**-basis decomposability in $c$ coincides with basis decomposability (Def. 6) in the special case of $\mathbf{Z} = \mathbf{X}$. Therefore, **Z**-regular orthogonality of $c$ in the special case $\mathbf{Z} = \mathbf{X}$ is equivalent to regular orthogonality as defined in Def. 7.

We aim at showing that **Z**-regular orthogonality is sufficient for **Z**-orthogonality (Lem. A.3), thus implying regular orthogonal is sufficient for orthogonality in the special case $\mathbf{X} = \mathbf{Z}$ (Thm. A.1). In order to show this result, we firstly prove the following lemma, saying that the integral over variables $\mathbf{Z} \subseteq \mathbf{X}$ of the product of two circuits $c_1, c_2$ defined over **X** annihilates (i.e., it is zero), whenever the input functions in $c_1$ and $c_2$ over the same variable $X \in \mathbf{Z}$ are orthogonal with each other.

**Lemma A.2.** Let $c_1, c_2$ be smooth and decomposable circuits over variables **X**, having $n_1, n_2$ as output units, respectively. Assume that, for some variable $X \in \mathbf{X}$, the input functions over $X$ in $c_1$ and $c_2$ are orthogonal, i.e., $\exists X \in \mathbf{X} \colon \forall f \in \mathcal{B}_X(n_1), \forall g \in \mathcal{B}_X(n_2) \colon \int_{\mathsf{dom}(X)} f(x) g(x)^* \, \mathrm{d}x = 0$. Then, for any $\mathbf{Z} \subseteq \mathbf{X}$ such that $X \in \mathbf{Z}$, we have that $\int_{\mathsf{dom}(\mathbf{Z})} c_1(\mathbf{y}, \mathbf{z}) c_2(\mathbf{y}, \mathbf{z})^* \, \mathrm{d}\mathbf{z} = 0$, where $\mathbf{y} \in \mathsf{dom}(\mathbf{X} \setminus \mathbf{Z})$.

*Proof.* For the proof we will write down the polynomial encoded by a circuit, also called the *circuit polynomial* (Choi et al., 2020b), whose construction relies on the idea of *induced sub-circuit* of a unit.

**Definition A.5** (Induced sub-circuit (Choi et al., 2020b)). Let $c$ be a circuit over variables **X**. An *induced sub-circuit* $\zeta$ is a circuit constructed from $c$ as follows. The output unit $n$ in $c$ is also the output unit of $\zeta$. If $n$ is a product unit in $\zeta$ then every unit $i \in \mathsf{in}(n)$, i.e., with a connection from $i$ to $n$, is in $\zeta$. If $n$ is a sum unit in $\zeta$, then exactly one of its input unit $i \in \mathsf{in}(n)$ is in $\zeta$.

Note that each input unit in an induced sub-circuit $\zeta$ of a circuit $c$ is also an input unit in $c$. By "unrolling" the representation of $c$ as the sum of the collection of all its induced sub-circuits, we have that the function computed by $c$ can be written as the *circuit polynomial* below (Shpilka and Yehudayoff, 2010; Choi et al., 2020b):

$$c(\mathbf{X}) = \sum_{\zeta \in \mathcal{H}(c)} \left( \prod_{w \in \Theta(\zeta)} \right) \prod_{n \in \mathcal{I}(\zeta)} c_n(\mathsf{sc}(n))^{\kappa(n, \zeta)}, \tag{4}$$

where $\mathcal{H}(c)$ is the set of all induced sub-circuits of $c$, $\Theta(\zeta)$ is the set of all sum unit weights covered by the induced sub-circuit $\zeta$, $\mathcal{I}(\zeta)$ is the set of all input units in $\zeta$ and $\kappa(n, \zeta)$ is a positive integer denoting how many times the input unit $n \in \mathcal{I}(\zeta)$ is reachable in $\zeta$ from the output unit of $\zeta$. For brevity, we will denote the coefficients of the polynomial in Eq. (4) as $\omega(\zeta) = \prod_{w \in \Theta(\zeta)} w$.

In the particular case of $c$ being smooth and decomposable, we observe that also $\zeta$ must be from the construction of an induced sub-circuit. Therefore, under smoothness and decomposability, we have that each input unit in $\mathcal{I}(\zeta)$ can be reached from the output unit in $\zeta$ exactly one time, i.e., $\kappa(n, \zeta) = 1$ for any $n \in \mathcal{I}(\zeta)$ and for any $\zeta \in \mathcal{H}(c)$. Thanks to smoothness and decomposability, we also recover

that each input unit in $\mathcal{I}(\zeta)$ is defined over a different variable in $\mathbf{X}$. With these observations, we can rewrite Eq. (4) as follows

$$c(\mathbf{X}) = \sum_{\zeta \in \mathcal{H}(c)} \omega(\zeta) \prod_{X \in \mathbf{X}} f_{\zeta,X}(X), \tag{5}$$

where each $f_{\zeta,X}$ is the function computed by the only input unit in $\zeta$ over the variable $X \in \mathbf{X}$, i.e., $\exists n \in \mathcal{I}(\zeta), \mathsf{sc}(n) = \{X\} \colon c_n(X) = f_{\zeta,X}(X)$.

Now, given $c_1$, $c_2$ circuits as per hypothesis, from Eq. (5) we can write their circuit polynomials as

$$c_1(\mathbf{X}) = \sum_{\zeta_1 \in \mathcal{H}(c_1)} \omega(\zeta_1) \prod_{X \in \mathbf{X}} f_{\zeta_1,X}(X) \qquad \text{and} \qquad c_2(\mathbf{X}) = \sum_{\zeta_2 \in \mathcal{H}(c_2)} \omega(\zeta_2) \prod_{X \in \mathbf{X}} g_{\zeta_2,X}(X). \tag{6}$$

By hypothesis, we have that $\exists X \in \mathbf{X}$ such that $\forall f \in \mathcal{B}_X(n_1)$, $\forall g \in \mathcal{B}_X(n_2)$, $\int_{\mathsf{dom}(X)} f(x)g(x)^* \, \mathrm{d}x = 0$, where $n_1$, $n_2$ respectively denote the output units of $c_1$, $c_2$. Since the input functions of any induced sub-circuit of a circuit $c$ are always also input units of $c$, i.e., $f_{\zeta_1,X} \in \mathcal{B}_X(n_1)$, $g_{\zeta_2,X} \in \mathcal{B}_X(n_2)$ for any $X \in \mathbf{X}$ and for any $\zeta_1 \in \mathcal{H}(c_1)$, $\zeta_2 \in \mathcal{H}(c_2)$, we have that the following statement holds.

$$\exists X \in \mathbf{X}, \forall \zeta_1 \in \mathcal{H}(c_1), \forall \zeta_2 \in \mathcal{H}(c_2) \colon \int_{\mathsf{dom}(X)} f_{\zeta_1,X}(x) g_{\zeta_2,X}(x)^* \, \mathrm{d}x = 0 \tag{7}$$

In other words, for at least one variable $X \in \mathbf{X}$, we have that the functions encoded by input units over $X$ in any pair $\zeta_1, \zeta_2$ of induced sub-circuits are orthogonal with each other. In the following, we exploit this observation to annihilate the integral over the product of $c_1$ and the conjugate of $c_2$ over any variables $\mathbf{Z} \subseteq \mathbf{X}$ such that $X \in \mathbf{Z}$, thus yielding the wanted result.

That is, by fixing $\mathbf{y} \in \mathbf{Y} = \mathsf{dom}(\mathbf{X} \setminus \mathbf{Z})$ and from the circuit polynomials in Eq. (6), we write down the integral of the product of $c_1$ and $c_2$ w.r.t. to the variables in $\mathbf{Z}$ as follows

$$\int_{\mathsf{dom}(\mathbf{Z})} c_1(\mathbf{y}, \mathbf{z}) c_2(\mathbf{y}, \mathbf{z})^* \, \mathrm{d}\mathbf{z} = \sum_{\zeta_1 \in \mathcal{H}(c_1)} \sum_{\zeta_2 \in \mathcal{H}(c_2)} \omega(\zeta_1)\omega(\zeta_2)^* \overbrace{\left( \prod_{V \in \mathbf{Y}} f_{\zeta_1,V}(\mathbf{y}_V) g_{\zeta_2,V}(\mathbf{y}_V)^* \right)}^{\text{products of functions not depending on } \mathbf{Z} \text{ or } X}$$

$$\cdot \underbrace{\left( \prod_{V \in \mathbf{Z} \setminus \{X\}} \int_{\mathsf{dom}(V)} f_{\zeta_1,V}(\mathbf{z}_V) g_{\zeta_2,V}(\mathbf{z}_V)^* \, \mathrm{d}\mathbf{z}_V \right)}_{\text{integrals of products of functions depending on } \mathbf{Z} \setminus \{X\}} \underbrace{\left( \int_{\mathsf{dom}(X)} f_{\zeta_1,X}(\mathbf{z}_X) g_{\zeta_2,X}(\mathbf{z}_X)^* \, \mathrm{d}\mathbf{z}_X \right)}_{= \, 0 \text{ because of Eq. (7)}}$$

where we generally denote as $\mathbf{x}_V$ the assignment to the variable $V$ found in the assignments $\mathbf{x}$. By plugging Eq. (7) into the formula above, we recover that the products annihilate, since the inner products of functions over $X$ in $\zeta_1$ and in $\zeta_2$ are orthogonal. Therefore, this shows that $\int_{\mathsf{dom}(\mathbf{Z})} c_1(\mathbf{y}, \mathbf{z}) c_2(\mathbf{y}, \mathbf{z})^* \, \mathrm{d}\mathbf{z} = 0$. $\qquad\square$

By applying Lem. A.2, we prove in the following theorem that $\mathbf{Z}$-regular orthogonality is a sufficient condition for $\mathbf{Z}$-orthogonality. Then, in Thm. A.1 we prove that, when $\mathbf{Z} = \mathbf{X}$, our Lem. A.3 (Def. 7) implies the sufficiency of regular orthogonality for orthogonality (Def. 5).

**Lemma A.3** ($\mathbf{Z}$-regular orthogonality $\implies$ $\mathbf{Z}$-orthogonality). *Let $c$ be a $\mathbf{Z}$-regular orthogonal circuit over variables $\mathbf{X}$, with $\mathbf{Z} \subseteq \mathbf{X}$. Then, $c$ is $\mathbf{Z}$-orthogonal.*

*Proof.* By $\mathbf{Z}$-regular orthogonality of $c$, we have that all input units in $c$ over the same variable $X \in \mathbf{Z}$ encode orthogonal functions. In addition, let $n$ be a sum unit in $c$ having scope overlapping with $\mathbf{Z}$, i.e., $\hat{\mathbf{Z}} = \mathsf{sc}(n) \cap \mathbf{Z} \neq \varnothing$, and let $i, j \in \mathsf{in}(n)$ be any pair of inputs to $n$ such that $i \neq j$. From $\mathbf{Z}$-regular orthogonality of $c$, we have that $\exists X \in \hat{\mathbf{Z}} \colon \mathcal{B}_X(i) \cap \mathcal{B}_X(j) = \varnothing$. By combining orthogonality of input functions over $X \in \mathbf{Z}$ and $\mathbf{Z}$-basis decomposability of $n$, we recover that $\exists X \in \hat{\mathbf{Z}}, \forall f \in \mathcal{B}_X(i), \forall g \in \mathcal{B}_X(j) \colon \int_{\mathsf{dom}(X)} f(x)g(x)^* \, \mathrm{d}x = 0$. Now, under these results we can apply Lem. A.2 and obtain that $c_i$ and $c_j$ are orthogonal when fixing the variables not in $\hat{\mathbf{Z}}$, i.e., $\int_{\mathsf{dom}(\hat{\mathbf{z}})} c_i(\mathbf{y}, \hat{\mathbf{z}}) c_j(\mathbf{y}, \hat{\mathbf{z}})^* \, \mathrm{d}\hat{\mathbf{z}} = 0$, where $\mathbf{y} \in \mathsf{dom}(\mathsf{sc}(n) \setminus \hat{\mathbf{Z}})$. Thus, we recovered the wanted result, i.e., $\forall i, j \in \mathsf{in}(n), i \neq j, \int_{\mathsf{dom}(\hat{\mathbf{z}})} c_i(\mathbf{y}, \hat{\mathbf{z}}) c_j(\mathbf{y}, \hat{\mathbf{z}})^* \, \mathrm{d}\hat{\mathbf{z}} = 0$. That is, every sum unit in $c$ having scope overlapping with $\mathbf{Z}$ is $\mathbf{Z}$-orthogonal, i.e., $c$ is a $\mathbf{Z}$-orthogonal circuit. $\qquad\square$

**Theorem A.1.** Let $c$ be a regular orthogonal circuit over variables $\mathbf{X}$. Then, $c$ is orthogonal.

*Proof.* Regular orthogonality in $c$ translates to $\mathbf{Z}$-regular orthogonality in $c$ with $\mathbf{Z} = \mathbf{X}$. Therefore, from Lem. A.3 we have that $c$ is $\mathbf{X}$-orthogonal and thus orthogonal. □

### A.4 MARGINALIZING ANY VARIABLES SUBSET IN LINEAR TIME

Under $\mathbf{Z}$-orthogonality, Lem. A.1 ensures that computing the particular marginal quantity $\int_{\mathsf{dom}(\mathbf{Z})} |c(\mathbf{y}, \mathbf{z})|^2 \, d\mathbf{z}$ requires time $\mathcal{O}(|c|)$. As we formalize in the following lemma, if a circuit is $\{X\}$-orthogonal for all variables $X \in \mathbf{X}$, then it is $\mathbf{Z}$-orthogonal for all $\mathbf{Z} \subseteq \mathbf{X}$, thus allowing us to compute *any* marginal in time $\mathcal{O}(|c|)$ by using our Alg. A.1. We will use this lemma later in Thm. A.2 to formalize sufficient conditions based on regular orthogonality (see App. A.3), i.e., based on the circuit structure and parameterization, to marginalize any variables subset in linear time.

**Lemma A.4.** Let $c$ be a circuit over variables $\mathbf{X}$ that is $\{X\}$-orthogonal for all $X \in \mathbf{X}$. Then $c$ is $\mathbf{Z}$-orthogonal for all $\mathbf{Z} \subseteq \mathbf{X}$.

*Proof.* To prove this, we need to show that if every sum unit $n$ in $c$ is $\{X\}$-orthogonal for all $X \in \mathsf{sc}(n)$, then $n$ is $\mathbf{Z}$-orthogonal for all $\mathbf{Z} \subseteq \mathsf{sc}(n)$. Let $n$ be a sum unit in $c$ having scope $\mathsf{sc}(n)$. By $\{X\}$-orthogonality of $c$ for all $X \in \mathbf{X}$, we have that the inputs to $n$ encode orthogonal functions whenever we fix the variables in $\mathsf{sc}(n) \setminus \{X\}$ for all $X \in \mathsf{sc}(n)$. Formally, we have that $\forall X \in \mathsf{sc}(n), \forall i, j \in \mathsf{in}(n), i \neq j \colon \int_{\mathsf{dom}(X)} c_i(\mathbf{y}, x) c_j(\mathbf{y}, x)^* \, dx = 0$, for any $\mathbf{y} \in \mathsf{dom}(\mathsf{sc}(n) \setminus X)$. Now, consider a subset $\mathbf{Z} \subseteq \mathsf{sc}(n)$. Therefore, given any $X \in \mathbf{Z}, \hat{\mathbf{Z}} = \mathbf{Z} \setminus \{X\}$, for all $i, j \in \mathsf{in}(n)$ with $i \neq j$ we can write

$$\int_{\mathsf{dom}(\mathbf{Z})} c_i(\mathbf{y}, \mathbf{z}) c_j(\mathbf{y}, \mathbf{z})^* \, d\mathbf{z} = \int_{\mathsf{dom}(\hat{\mathbf{Z}})} \left( \int_{\mathsf{dom}(X)} c_i(\mathbf{y}, \hat{\mathbf{z}}, x) c_j(\mathbf{y}, \hat{\mathbf{z}}, x)^* \, dx \right) d\hat{\mathbf{z}} = 0,$$

since the inner integral over $X$ is equal to zero for any variables assignments $\mathbf{y}$ and $\hat{\mathbf{z}}$, as $n$ is $\{X\}$-orthogonal. Therefore, we recover that $n$ is $\mathbf{Z}$-orthogonal for all $\mathbf{Z} \subseteq \mathsf{sc}(n)$. □

We then use the above lemma to formalize the result saying that if a circuit is $\{X\}$-regular orthogonal w.r.t. all variables $X$ (see Def. A.4), then it enables the computation of any marginal in linear time.

**Theorem A.2.** Let $c$ be a circuit over variables $\mathbf{X}$ that is $\{X\}$-regular orthogonal for all variables $X \in \mathbf{X}$. That is, for any sum unit $n$ in $c$ we have that its inputs depend on non-overlapping basis scopes for all variables, i.e., $\forall X \in \mathsf{sc}(n), \forall i, j \in \mathsf{in}(n), i \neq j \colon \mathcal{B}_X(i) \cap \mathcal{B}_X(j) = \varnothing$; and all input units over the same variable encode orthogonal functions. Then computing $\int_{\mathsf{dom}(\mathbf{Z})} |c(\mathbf{y}, \mathbf{z})|^2 \, d\mathbf{z}$ for any $\mathbf{Z} \subseteq \mathbf{X}$, with $\mathbf{y} \in \mathsf{dom}(\mathbf{X} \setminus \mathbf{Z})$ can be done in time $\mathcal{O}(|c|)$.

*Proof.* Since $c$ is $\{X\}$-regular orthogonal for all $X \in \mathbf{X}$, from Lem. A.3 we recover that $c$ is $\{X\}$-orthogonal for all $X \in \mathbf{X}$. Thus, we can apply Lem. A.4 to say that $c$ is also $\mathbf{Z}$-orthogonal for all variables subsets $\mathbf{Z}$ of $\mathbf{X}$. Therefore, by applying Lem. A.1 we conclude that computing any marginal can be done in time $\mathcal{O}(|c|)$ by using Alg. A.1. □

### A.5 TENSORIZED CIRCUIT MULTIPLICATION ALGORITHM

In the case of tensorized circuits whose sum layers can only receive input from exactly one other layer, Loconte et al. (2024) already proposed a circuit squaring algorithm operating on layers that only requires simple linear algebra operations. This assumption over sum layers is particularly convenient, as it ensures the that tensorized circuit is structured-decomposable by construction (Loconte et al., 2024), and therefore representing their squaring as yet another decomposable circuit is tractable (Vergari et al., 2021). In this section, we extend this squaring algorithm to circuits whose sum layers can receive input from more than one layer, as required by the tensorized circuit definition by Loconte et al. (2025a) and that we report in Def. 8. This particular difference between Def. 8 and the circuit representation used in Loconte et al. (2024) allows us to build tensorized squared PCs that are not necessarily structured-decomposable. Nevertheless, in §5 we show conditions to enable

---

**Algorithm A.2** MULTIPLY$(c_1, c_2)$

**Input:** Tensorized and compatible circuits $c_1$, $c_2$ over variables $\mathbf{X}_1$, $\mathbf{X}_2$ respectively, and having $\boldsymbol{\ell}_1$, $\boldsymbol{\ell}_2$ as output layers, respectively. **Output:** The output layer $\boldsymbol{\ell}$ of a circuit over variables $\mathbf{X}_1 \cup \mathbf{X}_2$ such that $\boldsymbol{\ell}(\mathbf{X}_1 \cup \mathbf{X}_2) = \boldsymbol{\ell}_1(\mathbf{X}_1) \otimes \boldsymbol{\ell}_2(\mathbf{X}_2)$.

1: **if** $\mathsf{sc}(\boldsymbol{\ell}_1) \cap \mathsf{sc}(\boldsymbol{\ell}_2) = \varnothing$ **then return** $\boldsymbol{\ell}_1 \otimes \boldsymbol{\ell}_2$

2: **if** $\boldsymbol{\ell}_1, \boldsymbol{\ell}_2$ are input layers **then**
3:      Assume $\boldsymbol{\ell}_1$ (resp. $\boldsymbol{\ell}_2$) computes $K_1$ (resp. $K_2$) functions over a variable $X$
4:      **return** An input layer $\boldsymbol{\ell}$ computing $K_1 K_2$ functions as $\boldsymbol{\ell}_1 \otimes \boldsymbol{\ell}_2$

5: **if** $\boldsymbol{\ell}_1$ and $\boldsymbol{\ell}_2$ are sum layers **then**
6:      let $\boldsymbol{\ell}_1 = \mathbf{W}^{(1)}[\boldsymbol{\ell}_{11} \cdots \boldsymbol{\ell}_{1N_1}]$ and $\boldsymbol{\ell}_2 = \mathbf{W}^{(2)}[\boldsymbol{\ell}_{21} \cdots \boldsymbol{\ell}_{2N_2}]$
7:      Assume $\mathbf{W}_1 = [\mathbf{W}_{11} \cdots \mathbf{W}_{1N_1}]$ and $\mathbf{W}_2 = [\mathbf{W}_{21} \cdots \mathbf{W}_{2N_2}]$
8:      $\boldsymbol{\ell}'_{ij} \leftarrow$ MULTIPLY$(\boldsymbol{\ell}_{1i}, \boldsymbol{\ell}_{2j}), \forall i \in [N_1] \, \forall j \in [N_2]$
9:      **return** $(\mathbf{W}_1 \boxtimes \mathbf{W}_2)[\boldsymbol{\ell}'_{11} \cdots \boldsymbol{\ell}'_{ij} \cdots \boldsymbol{\ell}'_{N_1 N_2}]$
10:          where $\boxtimes$ denotes the Tracy-Singh product (see proof of Prop. A.2)

11: **if** $\boldsymbol{\ell}_1$ and $\boldsymbol{\ell}_2$ are Hadamard product layers **then**
12:      Assume $\boldsymbol{\ell}_1 = \boldsymbol{\ell}_{11} \odot \boldsymbol{\ell}_{12}$ and $\boldsymbol{\ell}_2 = \boldsymbol{\ell}_{21} \odot \boldsymbol{\ell}_{22}$
13:          where the circuit in $\boldsymbol{\ell}_{11}$ (resp. $\boldsymbol{\ell}_{12}$) is compatible with the circuit $\boldsymbol{\ell}_{21}$ (resp. $\boldsymbol{\ell}_{22}$).
14:      $\boldsymbol{\ell}'_1 \leftarrow$ MULTIPLY$(\boldsymbol{\ell}_{11}, \boldsymbol{\ell}_{21})$
15:      $\boldsymbol{\ell}'_2 \leftarrow$ MULTIPLY$(\boldsymbol{\ell}_{12}, \boldsymbol{\ell}_{22})$
16:      **return** $\boldsymbol{\ell}'_1 \odot \boldsymbol{\ell}'_2$

17: **if** $\boldsymbol{\ell}_1$ and $\boldsymbol{\ell}_2$ are Kronecker product layers **then**
18:      Assume $\boldsymbol{\ell}_1 = \boldsymbol{\ell}_{11} \otimes \boldsymbol{\ell}_{12}$ and $\boldsymbol{\ell}_2 = \boldsymbol{\ell}_{21} \otimes \boldsymbol{\ell}_{22}$
19:          where the circuit in $\boldsymbol{\ell}_{11}$ (resp. $\boldsymbol{\ell}_{12}$) is compatible with the circuit $\boldsymbol{\ell}_{21}$ (resp. $\boldsymbol{\ell}_{22}$).
20:      $\boldsymbol{\ell}'_1 \leftarrow$ MULTIPLY$(\boldsymbol{\ell}_{11}, \boldsymbol{\ell}_{21})$
21:      $\boldsymbol{\ell}'_2 \leftarrow$ MULTIPLY$(\boldsymbol{\ell}_{12}, \boldsymbol{\ell}_{22})$
22:      **return** $\mathbf{P}\,(\boldsymbol{\ell}'_1 \otimes \boldsymbol{\ell}'_2)$ where $\mathbf{P}$ is a permutation matrix (see proof of Prop. A.2).

---

tractable marginalization of any variables subset. Furthermore, here we trivially extend such squaring algorithm to tensorized circuits with complex parameters.

**Preliminaries.** Given a structured-decomposable and tensorized circuit $c$, we represent its modulus squaring as yet another decomposable and circuit. This can be done by multiplying $c$ with its conjugate $c^*$ (§2). Note that the conjugate $c^*$ can be efficiently obtained from $c$ by taking the conjugate of the functions computed by the input layers and the conjugate of the sum layer weights (Yu et al., 2023). This procedure preserves the structural properties of $c$, thus $c^*$ is also structured-decomposable and compatible with $c$. Thus, we can represent the product between $c$ and $c^*$ as yet another decomposable circuit in polytime (Vergari et al., 2021). In the following, we present an algorithm, namely MULTIPLY (Alg. A.2), in order to multiply two tensorized circuits that are compatible as another decomposable tensorized circuit. Therefore, the modulus squaring of a structured-decomposable circuit $c$ is the result of MULTIPLY$(c, c^*)$.

**Notation.** To simplify the notation, note that from now on we typically remove the scopes of the layers from the argument of the layer evaluations, e.g., we use $\boldsymbol{\ell} = \boldsymbol{\ell}_1 \otimes \boldsymbol{\ell}_2$ to mean $\boldsymbol{\ell}(\mathsf{sc}(\boldsymbol{\ell})) = \boldsymbol{\ell}_1(\mathsf{sc}(\boldsymbol{\ell}_1)) \otimes \boldsymbol{\ell}_2(\mathsf{sc}(\boldsymbol{\ell}_2))$. Furthermore, for simplicity we will assume that the product layers in the tensorized circuits $c_1, c_2$ to be multiplied are either Kronecker or Hadamard (i.e., no product between circuits with mixed Kronecker and Hadamard layers). This is without loss of generality, as one can always rewrite an Hadamard product as a Kronecker product followed by a sum layer encoding a linear transformation via a selection matrix that filters out the cross products (e.g., see Liu and Trenkler (2008, Lem. 1)).

**Proposition A.2.** Let $c_1, c_2$ be tensorized and compatible circuits over variables $\mathbf{X}_1$, $\mathbf{X}_2$ respectively. Then, there exists an algorithm constructing a smooth and decomposable circuit $c$ over $\mathbf{X}_1 \cup \mathbf{X}_2$ such that $c(\mathbf{x}) = c_1(\mathbf{x})\,c_2(\mathbf{x})$ for any $\mathbf{x} \in \mathsf{dom}(\mathbf{X}_1 \cup \mathbf{X}_2)$. Moreover, the algorithm runs in time $\mathcal{O}(L_1 L_2 S_{1,\max} S_{2,\max})$, where $L_1$ (resp. $L_2$) denotes the number of layers in $c_1$ (resp. $c_1$), and $S_{1,\max}$ (resp. $S_{2,\max}$) denotes the maximum layer size in $c_1$ (resp. $c_2$).

*Proof.* We prove the correctness of Alg. A.2 by structural induction. That is, given $\boldsymbol{\ell}_1$ and $\boldsymbol{\ell}_2$ the output layers of two compatible circuits over variables $\mathbf{X}_1$, $\mathbf{X}_2$, respectively, we show that Alg. A.2 returns the output layer $\boldsymbol{\ell}$ of another tensorized circuit over variables $\mathbf{X}_1 \cup \mathbf{X}_2$ such that $\boldsymbol{\ell} = \boldsymbol{\ell}_1 \otimes \boldsymbol{\ell}_2$.

To begin with, consider the case where $\mathbf{X}_1 \cap \mathbf{X}_2 = \varnothing$. Then, we construct $\ell$ as a Kronecker product layer taking $\ell_1, \ell_2$ as inputs. Moreover, if $\ell_1, \ell_2$ are both input layers over the same variable $\mathbf{X}_1 = \mathbf{X}_2 = \{X\}$, we construct another input layer over $X$ such that it computes all the pairwise products of the function computed by $\ell_1$ and $\ell_2$. That is, consider $\ell_1(X) = [f_1(X) \cdots f_{K_1}(X)]^\top$ and $\ell_2(X) = [g_1(X) \cdots g_{K_2}(X)]^\top$, then $\ell(X) = \ell_1(X) \otimes \ell_2(X) = [f_1(X)g_1(X) \cdots f_i(X)g_j(X) \cdots f_{K_1}(X)g_{K_2}(X)]^\top$. Next, we consider the cases where $\ell_1, \ell_2$ have overlapping scope and are either sum or product layers.

We continue the proof by first reviewing the definitions of *commutation matrix* and *Tracy-Singh product* and their properties, as they will be used to perform linear algebra transformations needed to show the correctness of Alg. A.2.

**Definition A.6** (Commutation matrix (Magnus and Neudecker, 1979))**.** A *commutation matrix* $\mathbf{K}^{(m,n)}$ is a $nm \times nm$ permutation matrix for which, for any $m \times n$ matrix $\mathbf{A}$, we have that $\mathbf{K}^{(m,n)}\mathsf{vec}(\mathbf{A}^\top) = \mathsf{vec}(\mathbf{A})$, where vec denotes the flattening (or vectorization) operation.

**Proposition A.3.** Let $\mathbf{P}_{rs}^{mn}$ denote the permutation matrix $\mathbf{I}_n \otimes \mathbf{K}^{(s,m)} \otimes \mathbf{I}_r$. The following properties hold (Neudecker and Wansbeek, 1983; Tracy and Jinadasa, 1989; Tracy, 1990).

(C1) Given $\mathbf{v} \in \mathbb{C}^m$, $\mathbf{w} \in \mathbb{C}^n$, then $\mathbf{K}^{(m,n)}(\mathbf{v} \otimes \mathbf{w}) = \mathbf{w} \otimes \mathbf{v}$.

(C2) $(\mathbf{K}^{(s,m)})^\top = (\mathbf{K}^{(s,m)})^{-1} = \mathbf{K}^{(m,s)}$ and $(\mathbf{P}_{rs}^{mn})^\top = (\mathbf{P}_{rs}^{mn})^{-1} = \mathbf{P}_{rm}^{sn} = \mathbf{I}_n \otimes \mathbf{K}^{(m,s)} \otimes \mathbf{I}_r$.

(C3) Given $\mathbf{A} \in \mathbb{C}^{m \times n}$, $\mathbf{B} \in \mathbb{C}^{r \times s}$, then $\mathsf{vec}(\mathbf{A} \otimes \mathbf{B}) = \mathbf{P}_{rm}^{sn}(\mathsf{vec}(\mathbf{A}) \otimes \mathsf{vec}(\mathbf{B}))$.

(C4) Given $\mathbf{a} \in \mathbb{C}^m$, $\mathbf{b} \in \mathbb{C}^n$, $\mathbf{c} \in \mathbb{C}^r$, $\mathbf{d} \in \mathbb{C}^s$, then $\mathbf{P}_{sn}^{rm}(\mathbf{a} \otimes \mathbf{b} \otimes \mathbf{c} \otimes \mathbf{d}) = \mathbf{a} \otimes \mathbf{c} \otimes \mathbf{b} \otimes \mathbf{d}$.

**Definition A.7** (Tracy-Singh product (Tracy and Singh, 1972))**.** Let $\mathbf{A} \in \mathbb{C}^{m \times n}$ be a block matrix where each block $\mathbf{A}^{(i,j)}$ is a $m_i \times n_j$ matrix, and similarly let $\mathbf{B} \in \mathbb{C}^{r \times s}$ be a block matrix where each block $\mathbf{B}^{(i,j)}$ is a $r_i \times s_j$ matrix. Here, $m = \sum_i m_i$, $n = \sum_j n_j$, $r = \sum_i r_i$, $s = \sum_j s_j$. The Tracy-Singh product between $\mathbf{A}$ and $\mathbf{B}$, with such blocks and denoted as $\mathbf{A} \boxtimes \mathbf{B}$ is defined as the block matrix $\mathbf{C} \in \mathbb{C}^{mr \times ns}$ where each block $\mathbf{C}^{((i,k),(j,l))}$ is computed as $\mathbf{A}^{(i,j)} \otimes \mathbf{B}^{(k,l)}$.

**Proposition A.4.** The following properties hold (Tracy and Jinadasa, 1989; Tracy, 1990).

(T1) For non-block matrices $\mathbf{A}, \mathbf{B}$, we have that $\mathbf{A} \boxtimes \mathbf{B} = \mathbf{A} \otimes \mathbf{B}$.

(T2) For block matrices $\mathbf{A}, \mathbf{B}$, $(\mathbf{A} \boxtimes \mathbf{B})^\top = \mathbf{A}^\top \boxtimes \mathbf{B}^\top$.

(T3) For block matrices $\mathbf{A}, \mathbf{B}$, we have that $\mathbf{A} \boxtimes \mathbf{B} = \mathbf{S}_m^r \mathbf{G}_m^r (\mathbf{A} \otimes \mathbf{B}) \mathbf{H}_n^s \mathbf{S}_s^n$, where each of the matrices $\mathbf{S}_\beta^\alpha$, $\mathbf{G}_\beta^\alpha$ and $\mathbf{H}_\beta^\alpha$ is a special kind of a $\alpha\beta \times \alpha\beta$ permutation matrix that depends on the number of row and column blocks in $\mathbf{A}, \mathbf{B}$.

(T4) For block matrices $\mathbf{A}, \mathbf{B}$ consisting of a single row of blocks, i.e., $m = m_1$ and $r = r_1$, we have that $\mathbf{A} \boxtimes \mathbf{B} = (\mathbf{A} \otimes \mathbf{B}) \mathbf{H}_n^s \mathbf{S}_s^n$.

(T5) For block matrices $\mathbf{A}, \mathbf{B}$, we have that $\mathbf{A} \otimes \mathbf{B} = \mathbf{H}_m^r \mathbf{S}_r^m (\mathbf{A} \boxtimes \mathbf{B}) \mathbf{S}_n^s \mathbf{G}_n^s$.

For a precise formalization of the permutation matrices $\mathbf{S}_\beta^\alpha$, $\mathbf{G}_\beta^\alpha$, $\mathbf{H}_\beta^\alpha$, and for the proofs of these statements, refer to Tracy and Jinadasa (1989, §2) and Tracy and Jinadasa (1989, Thm. 7).

We now consider the product and sum layers case by case.

**Case (i): Hadamard layers.** We assume without loss of generality that each product layer receives input from exactly two other layers and, due to compatibility, two pairs of product layers with overlapping scope will factorize their scope towards their layer inputs in the same way. Let $\ell_1$, $\ell_2$ be Hadamard product layers receiving inputs from $\mathsf{in}(\ell_1) = \{\ell_{11}, \ell_{12}\}, \mathsf{in}(\ell_2) = \{\ell_{21}, \ell_{22}\}$, respectively. From decomposability and compatibility of $\ell_1$ and $\ell_2$, we have that $\ell_{11}$ is compatible with $\ell_{21}$ and $\ell_{12}$ is compatible with $\ell_{22}$. As such, let $\ell_1'$ (resp. $\ell_2'$) be the output layer of the tensorized circuit obtained by recursively calling MULTIPLY$(\ell_{11}, \ell_{21})$ (resp. MULTIPLY$(\ell_{12}, \ell_{22})$). In other words, $\ell_1'$ and $\ell_2'$ compute $\ell_{11} \otimes \ell_{21}$ and $\ell_{12} \otimes \ell_{22}$, respectively. Then, we encode $\ell_1 \otimes \ell_2$ as yet another Hadamard layer $\ell$:

$$\ell = \ell_1 \otimes \ell_2 = (\ell_{11} \odot \ell_{12}) \otimes (\ell_{21} \odot \ell_{22}) = (\ell_{11} \otimes \ell_{21}) \odot (\ell_{12} \otimes \ell_{22}) = \ell_1' \odot \ell_2', \qquad (8)$$

where we used the mixed-product property of the Kronecker, with respect to the Hadamard product. Thus, Alg. A.2 returns another Hadamard layer $\ell$ that receive inputs from $\ell'_1, \ell'_2$, respectively.

**Case (ii): Kronecker layers.** We proceed similarly to the case of Hadamard layers above. Let $\ell_1, \ell_2$ be Kronecker product layers receiving inputs from $\mathrm{in}(\ell_1) = \{\ell_{11}, \ell_{12}\}, \mathrm{in}(\ell_2) = \{\ell_{21}, \ell_{22}\}$, respectively. From decomposability and compatibility of $\ell_1$ and $\ell_2$, we have that $\ell_{11}$ is compatible with $\ell_{21}$ and $\ell_{12}$ is compatible with $\ell_{22}$. As such, let $\ell'_1$ (resp. $\ell'_2$) be the output layer of the tensorized circuit obtained by recursively calling MULTIPLY$(\ell_{11}, \ell_{21})$ (resp. MULTIPLY$(\ell_{12}, \ell_{22})$). Then, we encode $\ell_1 \otimes \ell_2$ as yet another Hadamard layer $\ell$:

$$\ell = \ell_1 \otimes \ell_2 = (\ell_{11} \otimes \ell_{12}) \otimes (\ell_{21} \otimes \ell_{22}) = \mathbf{P}^{K_3 K_1}_{K_4 K_2}((\ell_{11} \otimes \ell_{21}) \otimes (\ell_{12} \otimes \ell_{22})) = \mathbf{P}^{K_3 K_1}_{K_4 K_2}(\ell'_1 \otimes \ell'_2),$$

where $\mathbf{P}^{K_3 K_1}_{K_4 K_2}$ is a permutation matrix used to re-arrange the Kronecker products as defined and shown in Prop. A.3, and $K_1, K_2, K_3, K_4$ respectively denote the output size of layers $\ell_{11}, \ell_{21}, \ell_{12}, \ell_{22}$. Therefore, Alg. A.2 returns a composition of a sum and a Kronecker layer, where the sum layer applies the permutation matrix $\mathbf{P}^{K_3 K_1}_{K_4 K_2}$ and the Kronecker layer receive inputs from $\ell'_1, \ell'_2$, respectively.

**Case (iii): sum layers.** Let $\ell_1, \ell_2$ be sum layers receiving inputs from layers $\mathrm{in}(\ell_1) = \{\ell_{11}, \ldots, \ell_{1N_1}\}$ and $\mathrm{in}(\ell_2) = \{\ell_{21}, \ldots, \ell_{2N_2}\}$, respectively. Moreover, let $\mathbf{W}^{(1)}, \mathbf{W}^{(2)}$ denote the parameter matrices of $\ell_1, \ell_2$, respectively. We will firstly consider the case $N_1 = N_2 = 1$, and generalize it for any $N_1, N_2$ later. If $N_1 = N_2 = 1$, then due to smoothness we have that $\ell_{11}$ and $\ell_{21}$ are compatible. As such let $\ell'_1$ denote the result from MULTIPLY$(\ell_{11}, \ell_{21})$. From induction hypothesis, we have that $\ell'_1$ computes $\ell_{11} \otimes \ell_{21}$. Now, we instantiate a layer $\ell$ over variables $\mathbf{X}_1 \cup \mathbf{X}_2$ such that

$$\ell = \ell_{11} \otimes \ell_{21} = (\mathbf{W}^{(1)} \ell_{11}) \otimes (\mathbf{W}^{(2)} \ell_{21}) = (\mathbf{W}^{(1)} \otimes \mathbf{W}^{(2)})(\ell_{11} \otimes \ell_{21}),$$

where we used the mixed-product property of the Kronecker operation. Therefore, we realize $\ell$ as another sum layer having $\mathbf{W}^{(1)} \otimes \mathbf{W}^{(2)}$ as parameters and receiving input from $\ell'_1$. Next, we consider the case $N_1, N_2 > 1$. In this case we rewrite $\mathbf{W}^{(1)} \in \mathbb{C}^{K_1 \times K_2}, \mathbf{W}^{(2)} \in \mathbb{C}^{J_1 \times J_2}$ as blockwise matrices as follows

$$\mathbf{W}^{(1)} = [\mathbf{W}^{(1,1)} \cdots \mathbf{W}^{(1,N_1)}] \qquad \mathbf{W}^{(1)} = [\mathbf{W}^{(2,1)} \cdots \mathbf{W}^{(2,N_2)}].$$

Now, by smoothness of $\ell_1$ and $\ell_2$, we have that $\forall i \in [N_1], \forall j \in [N_2], \ell_{1i}$ is compatible with $\ell_{2j}$. As such, by induction hypothesis we will denote as $\ell'_{ij}$ the layer obtained by calling MULTIPLY$(\ell_{1i}, \ell_{2j})$, i.e., $\ell'_{ij}$ is the output layer of smooth and decomposable circuit that computes the Kronecker product $\ell'_{ij} = \ell_{1i} \otimes \ell_{2j}$. Now, let $\mathbf{L}_1, \mathbf{L}_2$ denote placeholders for the layer concatenations $[\ell_{11} \cdots \ell_{1N_1}]$ and $[\ell_{21} \cdots \ell_{2N_2}]$, respectively. To retrieve another layer $\ell$ such that it computes $\ell_1 \otimes \ell_2$, we rewrite

$$\ell = \ell_1 \otimes \ell_2 = (\mathbf{W}^{(1)} \otimes \mathbf{W}^{(2)})(\mathbf{L}_1 \otimes \mathbf{L}_2) = (\mathbf{W}^{(1)} \otimes \mathbf{W}^{(2)})\mathbf{H}^{J_2}_{K_2}\mathbf{S}^{K_2}_{J_2}(\mathbf{L}_1 \boxtimes \mathbf{L}_2)$$

$$= (\mathbf{W}^{(1)} \boxtimes \mathbf{W}^{(2)})(\mathbf{L}_1 \boxtimes \mathbf{L}_2) = (\mathbf{W}^{(1)} \boxtimes \mathbf{W}^{(2)})[\ell'_{11} \cdots \ell'_{ij} \cdots \ell'_{N_1 N_2}],$$

where we used the following properties: (i) Kronecker mixed-product property; (ii) the transformation from Kronecker to Tracy-Singh product by using the permutation matrix $\mathbf{H}^{J_2}_{K_2}\mathbf{S}^{K_2}_{J_2}$ as written in the property (T5) in Prop. A.4; and (iii) the similar transformation using the permutation matrix $\mathbf{H}^{J_2}_{K_2}\mathbf{S}^{K_2}_{J_2}$ as written in property (T4) in Prop. A.4, which is a specialization of (T3) as the number of block rows in $\mathbf{W}^{(1)}$ and $\mathbf{W}^{(2)}$ is one. Therefore, we obtained that for sum layers $\ell_1, \ell_2$ having arity $N_1 \geq 1$, $N_2 \geq 1$ in general, our Alg. A.2 returns another sum layer $\ell$ receiving inputs $\mathrm{in}(\ell) = \{\ell'_{ij}\}^{N_1, N_2}_{i=1, j=1}$ and parameterized by the weight matrix $\mathbf{W}^{(1)} \boxtimes \mathbf{W}^{(2)}$.

Finally, we consider the complexity of our Alg. A.2. Due to Kronecker products, we firstly observe that the size of each layer in the resulting product circuit must be $\mathcal{O}(S_{1,\max} S_{2,\max})$ where $S_{1,\max}$ (resp. $S_{2,\max}$) denotes the maximum layer size in $c_1$ (resp. $c_2$). Furthermore, multiplying sum layers $\ell_1$, $\ell_2$ receiving inputs from $N_1$ layers and $N_2$ layers respectively, is done calling Alg. A.2 recursively on all the $N_1 N_2$ pairings of inputs to $\ell_1$ and $\ell_2$. Thus, the number of layers of the resulting product circuit is $\mathcal{O}(L_1 L_2)$ in the worst case, where $L_1$ (resp. $L_2$) denotes the number of layers in $c_1$ (resp. $c_2$). Therefore, Alg. A.2 must run in worst case time $\mathcal{O}(L_1 L_2 S_{1,\max} S_{2,\max})$. $\qquad\square$

A.6    ALREADY-NORMALIZED TENSORIZED SQUARED CIRCUITS VIA UNITARITY

In the following we prove that the modulus squaring of a tensorized circuit that is unitarity, i.e., it satisfies (U1) to (U3), is orthogonal (Def. 5) and encodes an already-normalized distribution (Thm. 2). Below we start by clarifying some notation.

**Notation.** In this section and in App. A.7, we require integrating layers $\boldsymbol{\ell}$ that output vectors, e.g., in $\mathbb{C}^K$. That is, given a layer $\boldsymbol{\ell}$ having scope $\mathsf{sc}(\boldsymbol{\ell}) = \mathbf{Y} \cup \mathbf{Z}$ and encoding a function $\boldsymbol{\ell} \colon \mathsf{dom}(\mathbf{Y} \cup \mathbf{Z}) \to \mathbb{C}^K$, we write $\int_{\mathsf{dom}(\mathbf{Z})} \boldsymbol{\ell}(\mathbf{y}, \mathbf{z}) \, \mathrm{d}\mathbf{z}$ to refer to the $K$-dimensional vector obtained by integrating the $K$ univariate function components encoded by the scalar computational units in $\boldsymbol{\ell}$:

$$\int_{\mathsf{dom}(\mathbf{Z})} \boldsymbol{\ell}(\mathbf{y}, \mathbf{z}) \, \mathrm{d}\mathbf{z} = \begin{bmatrix} \int_{\mathsf{dom}(\mathbf{Z})} \boldsymbol{\ell}(\mathbf{y}, \mathbf{z})_1 \, \mathrm{d}\mathbf{z} & \int_{\mathsf{dom}(\mathbf{Z})} \boldsymbol{\ell}(\mathbf{y}, \mathbf{z})_2 \, \mathrm{d}\mathbf{z} & \cdots & \int_{\mathsf{dom}(\mathbf{Z})} \boldsymbol{\ell}(\mathbf{y}, \mathbf{z})_K \, \mathrm{d}\mathbf{z} \end{bmatrix} \in \mathbb{C}^K.$$

Moreover, due to the linearity of the matrix-vector product, we can write

$$\int_{\mathsf{dom}(\mathbf{Z})} \mathbf{W} \boldsymbol{\ell}(\mathbf{y}, \mathbf{z}) \, \mathrm{d}\mathbf{z} = \mathbf{W} \begin{bmatrix} \int_{\mathsf{dom}(\mathbf{Z})} \boldsymbol{\ell}(\mathbf{y}, \mathbf{z})_1 \, \mathrm{d}\mathbf{z} & \cdots & \int_{\mathsf{dom}(\mathbf{Z})} \boldsymbol{\ell}(\mathbf{y}, \mathbf{z})_K \, \mathrm{d}\mathbf{z} \end{bmatrix} = \mathbf{W} \int_{\mathsf{dom}(\mathbf{Z})} \boldsymbol{\ell}(\mathbf{y}, \mathbf{z}) \, \mathrm{d}\mathbf{z}.$$

Furthermore, for Hadamard products of layers having disjoint scopes, we can write

$$\int_{\mathsf{dom}(\mathbf{Z}_1) \times \mathsf{dom}(\mathbf{Z}_2)} \boldsymbol{\ell}_1(\mathbf{y}_1, \mathbf{z}_1) \odot \boldsymbol{\ell}_2(\mathbf{y}_2, \mathbf{z}_2) \, \mathrm{d}\mathbf{z}_1 \, \mathrm{d}\mathbf{z}_2 = \int_{\mathsf{dom}(\mathbf{Z}_1)} \boldsymbol{\ell}_1(\mathbf{y}_1, \mathbf{z}_1) \, \mathrm{d}\mathbf{z}_1 \odot \int_{\mathsf{dom}(\mathbf{Z}_2)} \boldsymbol{\ell}_2(\mathbf{y}_1, \mathbf{z}_2) \, \mathrm{d}\mathbf{z}_2,$$

where $(\mathbf{Y}_1, \mathbf{Y}_2)$ is a partitioning of $\mathbf{Y}$ and $(\mathbf{Z}_1, \mathbf{Z}_2)$ is a partitioning of $\mathbf{Z}$. The above equality still holds if we replace the Hadamard product ($\odot$) with the Kronecker product ($\otimes$).

**Theorem 2.** Let $c$ be smooth and decomposable circuit over variables $\mathbf{X}$. If $c$ is unitary, i.e., it satisfies conditions **(U1-3)**, then we have that $c$ is orthogonal and $Z = \int_{\mathsf{dom}(\mathbf{X})} |c(\mathbf{x})|^2 \, \mathrm{d}\mathbf{x} = 1$.

*Proof.* We prove it bottom-up by showing that, since $c$ is unitary, then every layer $\boldsymbol{\ell}$ over variables $\mathbf{Z} \subseteq \mathbf{X}$ in $c$ satisfies $\int_{\mathsf{dom}(\mathbf{Z})} \boldsymbol{\ell}(\mathbf{z}) \otimes \boldsymbol{\ell}(\mathbf{z})^* \, \mathrm{d}\mathbf{z} = \mathsf{vec}(\mathbf{I}_K)$, where $\mathsf{vec}$ denotes the vectorization (or flattening) operation and $K$ denotes the size of the output of $\boldsymbol{\ell}$. Thus, for the last layer in $c$, i.e., having number of units $K = 1$ and computing the output of $c$, we have that $Z = 1$. In other words, we will inductively prove that each layer consisting of $K$ units encodes a vector of $K$ orthonormal functions. This will not only give us $Z = 1$ for the last layer in $c$, but also that $c$ is is orthogonal by observing orthonormality of the inputs to a sum layer. Below we proceed by cases.

**Case (i): input layer.** Let $\boldsymbol{\ell}$ be an input layer in $c$ over the variable $X$. By unitarity of $c$ and in particular from (U1), we have that $\boldsymbol{\ell}$ computes a vector of $K$ orthonormal functions $\boldsymbol{\ell}(X) = [f_1(X) \cdots f_K(X)]^\top$. Therefore, we have that $\int_{\mathsf{dom}(X)} \boldsymbol{\ell}(x) \otimes \boldsymbol{\ell}(x)^* \, \mathrm{d}x = \mathsf{vec}(\mathbf{I}_K)$.

**Case (ii): Hadamard product layer.** Let $\boldsymbol{\ell}$ be a Hadamard product layer in $c$ receiving inputs from layers $\mathsf{in}(\boldsymbol{\ell}) = \{\boldsymbol{\ell}_1, \boldsymbol{\ell}_2\}$. By decomposability, we have that $\mathsf{sc}(\boldsymbol{\ell}_1) = \mathbf{Z}_1$, $\mathsf{sc}(\boldsymbol{\ell}_2) = \mathbf{Z}_2$, with $\mathbf{Z}_1 \cap \mathbf{Z}_2 = \varnothing$ and $\mathsf{sc}(\boldsymbol{\ell}) = \mathbf{Z} = \mathbf{Z}_1 \cup \mathbf{Z}_2$. Assume by induction hypothesis that $\int_{\mathsf{dom}(\mathbf{Z}_1)} \boldsymbol{\ell}_1(\mathbf{z}_1) \otimes \boldsymbol{\ell}_1(\mathbf{z}_1)^* \, \mathrm{d}\mathbf{z}_1 = \mathsf{vec}(\mathbf{I}_K)$ and $\int_{\mathsf{dom}(\mathbf{Z}_2)} \boldsymbol{\ell}_2(\mathbf{z}_2) \otimes \boldsymbol{\ell}_2(\mathbf{z}_2)^* \, \mathrm{d}\mathbf{z}_2 = \mathsf{vec}(\mathbf{I}_K)$. Then, we have that

$$\int_{\mathsf{dom}(\mathbf{Z})} \boldsymbol{\ell}(\mathbf{z}) \otimes \boldsymbol{\ell}(\mathbf{z})^* \, \mathrm{d}\mathbf{z} = \int_{\mathsf{dom}(\mathbf{Z}_1) \times \mathsf{dom}(\mathbf{Z}_2)} (\boldsymbol{\ell}_1(\mathbf{z}_1) \odot \boldsymbol{\ell}_2(\mathbf{z}_2)) \otimes (\boldsymbol{\ell}_1(\mathbf{z}_1)^* \odot \boldsymbol{\ell}_2(\mathbf{z}_2)^*) \, \mathrm{d}\mathbf{z}_1 \, \mathrm{d}\mathbf{z}_2$$

$$= \left( \int_{\mathsf{dom}(\mathbf{Z}_1)} \boldsymbol{\ell}_1(\mathbf{z}_1) \otimes \boldsymbol{\ell}_1(\mathbf{z}_1)^* \, \mathrm{d}\mathbf{z}_1 \right) \odot \left( \int_{\mathsf{dom}(\mathbf{Z}_2)} \boldsymbol{\ell}_2(\mathbf{z}_2) \otimes \boldsymbol{\ell}_2(\mathbf{z}_2)^* \, \mathrm{d}\mathbf{z}_2 \right) = \mathsf{vec}(\mathbf{I}_K) \odot \mathsf{vec}(\mathbf{I}_K) = \mathsf{vec}(\mathbf{I}_K),$$

where we used the Kronecker mixed-product property with respect to the Hadamard product, and decomposed the integral into lower dimensional ones by using the fact that $\mathbf{Z}_1 \cap \mathbf{Z}_2 = \varnothing$.

**Case (iii): Kronecker product layer.** Let $\boldsymbol{\ell}$ be a Kronecker product layer in $c$ receiving inputs from layers $\mathsf{in}(\boldsymbol{\ell}) = \{\boldsymbol{\ell}_1, \boldsymbol{\ell}_2\}$. By decomposability, we have that $\mathsf{sc}(\boldsymbol{\ell}_1) = \mathbf{Z}_1$, $\mathsf{sc}(\boldsymbol{\ell}_2) = \mathbf{Z}_2$, with $\mathbf{Z}_1 \cap \mathbf{Z}_2 = \varnothing$ and $\mathsf{sc}(\boldsymbol{\ell}) = \mathbf{Z} = \mathbf{Z}_1 \cup \mathbf{Z}_2$. Assume by induction hypothesis that $\int_{\mathsf{dom}(\mathbf{Z}_1)} \boldsymbol{\ell}_1(\mathbf{z}_1) \otimes$

$\ell_1(\mathbf{z}_1)^* \, d\mathbf{z}_1 = \text{vec}(\mathbf{I}_{K_1})$ and $\int_{\text{dom}(\mathbf{Z}_2)} \ell_2(\mathbf{z}_2) \otimes \ell_2(\mathbf{z}_2)^* \, d\mathbf{z}_2 = \text{vec}(\mathbf{I}_{K_2})$. Then, we have that

$$
\begin{aligned}
\int_{\text{dom}(\mathbf{Z})} \ell(\mathbf{z}) \otimes \ell(\mathbf{z})^* \, d\mathbf{z} &= \int_{\text{dom}(\mathbf{Z}_1) \times \text{dom}(\mathbf{Z}_2)} (\ell_1(\mathbf{z}_1) \otimes \ell_2(\mathbf{z}_2)) \otimes (\ell_1(\mathbf{z}_1)^* \otimes \ell_2(\mathbf{z}_2)^*) \, d\mathbf{z}_1 \, d\mathbf{z}_2 \\
&= \int_{\text{dom}(\mathbf{Z}_1) \times \text{dom}(\mathbf{Z}_2)} \mathbf{P}_{K_2 K_1}^{K_2 K_1} [(\ell_1(\mathbf{z}_1) \otimes \ell_1(\mathbf{z}_1)^*) \otimes (\ell_2(\mathbf{z}_2) \otimes \ell_2(\mathbf{z}_2)^*)] \, d\mathbf{z}_1 \, d\mathbf{z}_2 \\
&= \mathbf{P}_{K_2 K_1}^{K_2 K_1} \left[ \left( \int_{\text{dom}(\mathbf{Z}_1)} \ell_1(\mathbf{z}_1) \otimes \ell_1(\mathbf{z}_1)^* \, d\mathbf{z}_1 \right) \otimes \left( \int_{\text{dom}(\mathbf{Z}_2)} \ell_2(\mathbf{z}_2) \otimes \ell_2(\mathbf{z}_2)^* \, d\mathbf{z}_2 \right) \right] \\
&= \mathbf{P}_{K_2 K_1}^{K_2 K_1} (\text{vec}(\mathbf{I}_{K_1}) \otimes \text{vec}(\mathbf{I}_{K_2})) = \text{vec}(\mathbf{I}_{K_1} \otimes \mathbf{I}_{K_2}) = \text{vec}(\mathbf{I}_{K_1 K_2}),
\end{aligned}
$$

where $\mathbf{P}_{K_2 K_1}^{K_2 K_1}$ is a permutation matrix as defined as in Prop. A.3. In particular, in the above we apply the property (C4) in Prop. A.3, then we decompose the integral into lower dimensional ones by using the fact that $\mathbf{Z}_1 \cap \mathbf{Z}_2 = \varnothing$, and finally use the property (C3) in Prop. A.3.

**Case (iv): sum layer.** Let $\ell$ be a sum layer in $c$ receiving inputs from layers $\text{in}(\ell) = \{\ell_i\}_{i=1}^N$, and $\ell$ is parameterized by a (semi-)unitary matrix $\mathbf{W} \in \mathbb{C}^{K_1 \times K_2}$ with $K_1 \leq K_2$ by unitarity of $c$, i.e., $\mathbf{W}\mathbf{W}^\dagger = \mathbf{I}_{K_1}$ as for (U3). From smoothness, we recover that $\text{sc}(\ell_i) = \text{sc}(\ell) = \mathbf{Z}$, for any $i \in [N]$. We firstly assume that $N = 1$, i.e., $\text{in}(\ell) = \{\ell_1\}$, and then handle the case $N > 1$ later. For $N = 1$ by induction hypothesis we have that $\int_{\text{dom}(\mathbf{Z})} \ell_1(\mathbf{z}) \otimes \ell_1(\mathbf{z})^* \, d\mathbf{z} = \text{vec}(\mathbf{I}_{K_2})$. Then, we have that

$$
\begin{aligned}
\int_{\text{dom}(\mathbf{Z})} \ell(\mathbf{z}) \otimes \ell(\mathbf{z})^* \, d\mathbf{z} &= \int_{\text{dom}(\mathbf{Z})} (\mathbf{W}\ell_1(\mathbf{z})) \otimes (\mathbf{W}^*\ell_1(\mathbf{z})^*) \, d\mathbf{z} = (\mathbf{W} \otimes \mathbf{W}^*) \int_{\text{dom}(\mathbf{Z})} \ell_1(\mathbf{z}) \otimes \ell_1(\mathbf{z})^* \, d\mathbf{z} \\
&= (\mathbf{W} \otimes \mathbf{W}^*)\text{vec}(\mathbf{I}_{K_2}) = \text{vec}(\mathbf{W}^*\mathbf{I}_{K_2}\mathbf{W}^\top) = \text{vec}((\mathbf{W}\mathbf{W}^\dagger)^*) = \text{vec}(\mathbf{I}_{K_1}),
\end{aligned}
$$

where we used the Kronecker mixed-product property, and the following property of the Kronecker: $(\mathbf{A} \otimes \mathbf{B})\text{vec}(\mathbf{V}) = \text{vec}(\mathbf{B}\mathbf{V}\mathbf{A}^\top)$ for matrices $\mathbf{A}, \mathbf{B}, \mathbf{V}$. Consider now the case $N > 1$. For all $i \in [N]$ by induction hypothesis we have that $\int_{\text{dom}(\mathbf{Z})} \ell_i(\mathbf{z}) \otimes \ell_i(\mathbf{z})^* \, d\mathbf{z} = \text{vec}(\mathbf{I}_{J_i})$, where $J_i$ denotes the size of the output of $\ell_i$, i.e., $K_2 = \sum_{i=1}^N J_i$. With a slight abuse of notation, we overload the basis scope definition (Def. A.2) to layers rather than units, i.e., we denote as $\mathcal{B}_X(\ell)$ the union of all basis scopes w.r.t. $X$ of the units within $\ell$. Then, from (U2) of unitarity by hypothesis, we can apply the following lemma.

**Lemma A.5.** Let $\ell_1, \ell_2$ be output layers of smooth and decomposable tensorized circuits $c_1, c_2$ over variables $\mathbf{X}$. Assume that, for some $X \in \mathbf{X}$, the input functions computed by the input layers over $X$ in $c_1$ and $c_2$ are orthogonal with each other, i.e., $\exists X \in \mathbf{X} \colon \forall f \in \mathcal{B}_X(\ell_1), \forall g \in \mathcal{B}_X(\ell_2) \colon \int_{\text{dom}(X)} f(x)g(x)^* \, dx = 0$. Then, for any $\mathbf{Z} \subseteq \mathbf{X}$ such that $X \in \mathbf{Z}$, we have that $\int_{\text{dom}(\mathbf{Z})} \ell_1(\mathbf{y}, \mathbf{z}) \otimes \ell_2(\mathbf{y}, \mathbf{z})^* \, d\mathbf{z} = \mathbf{0}$, where $\mathbf{y} \in \text{dom}(\mathbf{X} \setminus \mathbf{Z})$.

*Proof.* By hypothesis for some $X \in \mathbf{Z} \subseteq \mathbf{X}$ the basis scopes of $\ell_1$ and $\ell_2$ w.r.t. $X$ consists of orthogonal functions over $X$. As such, for any pair of units $n$ and $m$ respectively in $\ell_1$ and $\ell_2$, the input functions over $X$ in the sub-circuits rooted in $n$ and $m$, respectively, are orthogonal. Therefore, we can apply Lem. A.2 to recover the wanted result: all pairs of units $n$ and $m$ respectively in $\ell_1$ and $\ell_2$ encode orthogonal functions when fixing variables not in $\mathbf{Z}$, i.e., $\int_{\text{dom}(\mathbf{Z})} \ell_1(\mathbf{y}, \mathbf{z}) \otimes \ell_2(\mathbf{y}, \mathbf{z})^* \, d\mathbf{z} = \mathbf{0}$. $\square$

Therefore, for all $i \in [N], j \in [N], i \neq j$, by leveraging (U1) and (U2) of unitarity we can apply Lem. A.5 and recover that $\int_{\text{dom}(\mathbf{Z})} \ell_i(\mathbf{z}) \otimes \ell_j(\mathbf{z})^* \, d\mathbf{z} = \mathbf{0}$, i.e., a zero vector of size $J_i J_j$. From these equalities and by rewriting Kronecker products in terms of outer products (denoted as $\circ$), we also obtain that $\int_{\text{dom}(\mathbf{Z})} \ell_i(\mathbf{z})^* \circ \ell_i(\mathbf{z}) \, d\mathbf{z} = \mathbf{I}_{J_i}$, and $\int_{\text{dom}(\mathbf{Z})} \ell_i(\mathbf{z})^* \circ \ell_j(\mathbf{z}) \, d\mathbf{z} = \mathbf{0}$ for any $i, j \in [N]$, $i \neq j$, where $\circ$ denotes the outer product. Therefore, we can write the following:

$$
\begin{aligned}
\int_{\text{dom}(\mathbf{Z})} \ell(\mathbf{z}) \otimes \ell(\mathbf{z})^* \, d\mathbf{z} &= \int_{\text{dom}(\mathbf{Z})} (\mathbf{W}[\ell_1(\mathbf{z}) \cdots \ell_N(\mathbf{z})]) \otimes (\mathbf{W}^*[\ell_1(\mathbf{z})^* \cdots \ell_N(\mathbf{z})^*]) \, d\mathbf{z} \\
&= (\mathbf{W} \otimes \mathbf{W}^*) \int_{\text{dom}(\mathbf{Z})} \text{vec} ([\ell_1(\mathbf{z})^* \cdots \ell_N(\mathbf{z})^*] \circ [\ell_1(\mathbf{z}) \cdots \ell_N(\mathbf{z})] \, d\mathbf{z})
\end{aligned}
$$

$$
\begin{aligned}
&= (\mathbf{W} \otimes \mathbf{W}^*)\mathsf{vec}\left(\begin{bmatrix} \int_{\mathsf{dom}(\mathbf{Z})} \boldsymbol{\ell}_1(\mathbf{z})^* \circ \boldsymbol{\ell}_1(\mathbf{z})\,\mathrm{d}\mathbf{z} & \cdots & \int_{\mathsf{dom}(\mathbf{Z})} \boldsymbol{\ell}_1(\mathbf{z})^* \circ \boldsymbol{\ell}_N(\mathbf{z})\,\mathrm{d}\mathbf{z} \\ \vdots & \ddots & \vdots \\ \int_{\mathsf{dom}(\mathbf{Z})} \boldsymbol{\ell}_N(\mathbf{z})^* \circ \boldsymbol{\ell}_1(\mathbf{z})\,\mathrm{d}\mathbf{z} & \cdots & \int_{\mathsf{dom}(\mathbf{Z})} \boldsymbol{\ell}_N(\mathbf{z})^* \circ \boldsymbol{\ell}_N(\mathbf{z})\,\mathrm{d}\mathbf{z} \end{bmatrix}\right) \\
&= (\mathbf{W} \otimes \mathbf{W}^*)\mathsf{vec}\left(\begin{bmatrix} \mathbf{I}_{J_1} & & \mathbf{0} \\ & \ddots & \\ \mathbf{0} & & \mathbf{I}_{J_N} \end{bmatrix}\right) \\
&= (\mathbf{W} \otimes \mathbf{W}^*)\mathsf{vec}(\mathbf{I}_{K_2}) = \mathsf{vec}(\mathbf{W}^*\mathbf{I}_{K_2}\mathbf{W}^\top) = \mathsf{vec}((\mathbf{W}\mathbf{W}^\dagger)^*) = \mathsf{vec}(\mathbf{I}_{K_1}),
\end{aligned}
$$

where we applied the same properties used for the case $N = 1$ shown above. From the above we recover that each sum unit in the sum layer $\boldsymbol{\ell}$ receives input from units encoding orthogonal functions, as integrating all pairwise products yields an identity matrix. Therefore, it turns out that $c$ is orthogonal. By recursively applying the above cases, if $\boldsymbol{\ell}$ is the output layer of the tensorized circuit $c$, then $K_1 = 1$, and we have that $Z = \int_{\mathsf{dom}(\mathbf{X})} |c(\mathbf{x})|^2\,\mathrm{d}\mathbf{x} = \int_{\mathsf{dom}(\mathbf{X})} \boldsymbol{\ell}(\mathbf{x})\boldsymbol{\ell}(\mathbf{x})^*\,\mathrm{d}\mathbf{x} = \mathsf{vec}(\mathbf{I}_1) = 1.$ $\square$

---

**Algorithm A.3** MAR-SQUARED-UNITARY$(c, \mathbf{y}, \mathbf{Z})$

**Input:** A tensorized circuit $c$ over variables $\mathbf{X}$ satisfying conditions (U1) to (U4), where $\boldsymbol{\ell}$ is the output layer in $c$; a set of variables $\mathbf{Z} \subseteq \mathbf{X}$ to marginalize, and an assignment $\mathbf{y}$ to variables $\mathbf{Y} = \mathbf{X} \setminus \mathbf{Z}$.
**Output:** The vector $\int_{\mathsf{dom}(\mathbf{Z})} \boldsymbol{\ell}(\mathbf{y}, \mathbf{z}) \otimes \boldsymbol{\ell}(\mathbf{y}, \mathbf{z})^*\,\mathrm{d}\mathbf{z}$. If $\boldsymbol{\ell}$ is the last layer of $c$, then it consists of exactly one unit, and thus the algorithm returns the marginal likelihood $p(\mathbf{y}) = \int_{\mathsf{dom}(\mathbf{Z})} |c(\mathbf{y}, \mathbf{z})|^2\,\mathrm{d}\mathbf{z}$.

1: **if** $\mathsf{sc}(\boldsymbol{\ell}) \setminus \mathbf{Z} = \varnothing$ **then**   ▷ $\boldsymbol{\ell}$ depends on only the variables being marginalized
2:    **return** $\mathsf{vec}(\mathbf{I}_K)$
3: **else if** $\mathsf{sc}(\boldsymbol{\ell}) \cap \mathbf{Z} = \varnothing$ **then**   ▷ $\boldsymbol{\ell}$ does not depend on the variables to marginalize
4:    $\mathbf{r} \leftarrow$ EVAL-FEED-FORWARD$(\boldsymbol{\ell}, \mathbf{y})$
5:    **return** $\mathbf{r} \otimes \mathbf{r}^*$
6: **else if** $\boldsymbol{\ell}$ is a sum layer **then**   ▷ $\boldsymbol{\ell}$ depend on *both* the variables to marginalize and the ones left over
7:    let $\boldsymbol{\ell}$ receive inputs from $\{\boldsymbol{\ell}_1, \ldots, \boldsymbol{\ell}_N\}$ and parameterized by $\mathbf{W} \in \mathbb{C}^{K_1 \times K_2}$
8:    Assume $\mathbf{W}$ is a block matrix $\mathbf{W} = [\mathbf{W}^{(1)} \cdots \mathbf{W}^{(N)}]$
9:    $\mathbf{r}_i \leftarrow$ MAR-SQUARED-UNITARY$(\boldsymbol{\ell}_i, \mathbf{y}, \mathbf{Z} \cap \mathsf{sc}(\boldsymbol{\ell}_i)), \forall i \in [N]$.
10:    let $\mathbf{R}_{ii}$ be the reshaping of $\mathbf{r}_i$ as a $J_i \times J_i$ matrix, $\forall i \in [N]$
11:    **return** $\mathsf{vec}\left(\sum_{i=1}^N \mathbf{W}^{(i)}\mathbf{R}_{ii}\mathbf{W}^{(i)\dagger}\right)^*$
12: **else if** $\boldsymbol{\ell}$ is a Hadamard product layer **then**
13:    let $\boldsymbol{\ell} = \boldsymbol{\ell}_1 \odot \boldsymbol{\ell}_2$
14:    $\mathbf{r}_1 \leftarrow$ MAR-SQUARED-UNITARY$(\boldsymbol{\ell}_1, \mathbf{y}, \mathbf{Z} \cap \mathsf{sc}(\boldsymbol{\ell}_1))$
15:    $\mathbf{r}_2 \leftarrow$ MAR-SQUARED-UNITARY$(\boldsymbol{\ell}_2, \mathbf{y}, \mathbf{Z} \cap \mathsf{sc}(\boldsymbol{\ell}_2))$
16:    **return** $\mathbf{r}_1 \odot \mathbf{r}_2$
17: **else**   ▷ $\boldsymbol{\ell}$ is a Kronecker product layer
18:    let $\boldsymbol{\ell} = \boldsymbol{\ell}_1 \otimes \boldsymbol{\ell}_2$
19:    $\mathbf{r}_1 \leftarrow$ MAR-SQUARED-UNITARY$(\boldsymbol{\ell}_1, \mathbf{y}, \mathbf{Z} \cap \mathsf{sc}(\boldsymbol{\ell}_1))$
20:    $\mathbf{r}_2 \leftarrow$ MAR-SQUARED-UNITARY$(\boldsymbol{\ell}_2, \mathbf{y}, \mathbf{Z} \cap \mathsf{sc}(\boldsymbol{\ell}_2))$
21:    **return** $\mathbf{P}(\mathbf{r}_1 \otimes \mathbf{r}_2)$, where $\mathbf{P}$ is a permutation matrix

---

### A.7 A TIGHTER MARGINALIZATION COMPLEXITY

**Theorem 3.** Let $c$ be a tensorized circuit over variables $\mathbf{X}$ that satisfies (U1-4), and let $\mathbf{Z} \subseteq \mathbf{X}$, $\mathbf{Y} = \mathbf{X} \setminus \mathbf{Z}$. Computing the marginal $p(\mathbf{y}) = \int_{\mathsf{dom}(\mathbf{Z})} |c(\mathbf{y}, \mathbf{z})|^2\,\mathrm{d}\mathbf{z}$ requires time $\mathcal{O}(|\phi_\mathbf{Y} \setminus \phi_\mathbf{Z}|S_{\max} + |\phi_\mathbf{Y} \cap \phi_\mathbf{Z}|S_{\max}^2)$, where $\phi_\star$ is the set of layers whose scope depends on at least one variable in $\star$.

*Proof.* We prove it by constructing Alg. A.3, i.e., the algorithm computing the marginal likelihood given by hypothesis. Alg. A.3 is based on two ideas. First, integrating sub-circuits whose layer depend only on the variables being integrated over (i.e., $\mathbf{Z}$) will yield identity matrices, so there is no need to evaluate these sub-circuits. Second, the sub-circuits whose layers depend on the variables that are *not* integrated over (i.e., $\mathbf{Y}$) do not need to be squared and can be evaluated bringing a linear rather than quadratic complexity w.r.t. the circuit size. Below, we consider different cases of layers based on the variables they depend on, and we later discuss the overall complexity.

**Case (i): layers depending on variables Z only.** Consider a layer $\ell$ in $c$ such that $\mathsf{sc}(\ell) \subseteq \mathbf{Z}$, i.e., $\ell \in \phi_{\mathbf{Z}} \setminus \phi_{\mathbf{Y}}$ by hypothesis. Since the tensorized circuit $c$ is unitary by hypothesis, the tensorized sub-circuit rooted in $\ell$ is also unitary. Therefore, from our proof for Thm. 2 we recover that integrating the Kronecker product of $\ell$ and its conjugate yields the flattening of an identity matrix. Formally, we have that $\int_{\mathsf{dom}(\mathsf{sc}(\ell) \cap \mathbf{Z})} \ell(\mathbf{z}) \otimes \ell(\mathbf{z})^* \, \mathrm{d}\mathbf{z} = \mathsf{vec}(\mathbf{I}_K)$, where $K$ denote the size of the output of $\ell$. Therefore, layers in $\phi_{\mathbf{Z}} \setminus \phi_{\mathbf{Y}}$ do not need be evaluated, and this is reflected in L1-2 of our Alg. A.3.

**Case (ii): layers depending on variables Y only.** Consider a layer $\ell$ in $c$ such that $\mathsf{sc}(\ell) \cap \mathbf{Z} = \varnothing$, i.e., $\ell \in \phi_{\mathbf{Y}} \setminus \phi_{\mathbf{Z}}$ by hypothesis. Since $\ell$ does not depend on the variables to be marginalized out, we can compute $\ell(\mathbf{y}) \otimes \ell(\mathbf{y})^*$ with $\mathbf{y} \in \mathsf{dom}(\mathsf{sc}(\ell) \cap \mathbf{Y})$ by evaluating $\ell$ on $\mathbf{y}$ and then computing the conjugation and Kronecker product. This case is captured by L3-5 in Alg. A.3. Note that the complexity of evaluating $\ell$ on $\mathbf{y}$ is $\mathcal{O}(|\phi_{\mathbf{Y}} \setminus \phi_{\mathbf{Z}}| S_{\max})$. Moreover, we observe that L3-5 are executed only on layers $\ell$ that are input to other layers having scope overlapping with *both* $\mathbf{Y}$ and $\mathbf{Z}$, i.e., they are in $\phi_{\mathbf{Y}} \cap \phi_{\mathbf{Z}}$ If that were not the case, then either **Case (i)** would have been executed, or L3-5 would have been executed on the layer receiving input from $\ell$ instead. Since each product layer in $\phi_{\mathbf{Y}} \cap \phi_{\mathbf{Z}}$ receives input from exactly two other layers $\ell_1, \ell_2$ and at most one between $\ell_1$ and $\ell_2$ can depend on variables $\mathbf{Y}$ only (i.e, it is in $\phi_{\mathbf{Y}} \setminus \phi_{\mathbf{Z}}$), we have that L3-5 are executed a number of times that is in $\mathcal{O}(|\phi_{\mathbf{Y}} \cap \phi_{\mathbf{Z}}|)$. This would be true even if a product layer receives input from more than two layers, as it can be casted into multiple product layers receiving input from exactly two other layers. Furthermore, since the size of a layer $\ell$ (i.e., the number of scalar input connections) is bounded by below by the number of units $K$ in $\ell$, we have that the Kronecker products (L5) will account for just a $\mathcal{O}(|\phi_{\mathbf{Y}} \cap \phi_{\mathbf{Z}}|K^2) \subseteq \mathcal{O}(|\phi_{\mathbf{Y}} \cap \phi_{\mathbf{Z}}|S_{\max}^2)$ factor to the overall time complexity of Alg. A.3.

**Case (iii): layers depending on variables both in Y and Z.** Consider a layer $\ell$ in $c$ such that $\mathsf{sc}(\ell) \cap \mathbf{Y} \neq \varnothing$ and $\mathsf{sc}(\ell) \cap \mathbf{Z} \neq \varnothing$, i.e., $\ell \in \phi_{\mathbf{Y}} \cap \phi_{\mathbf{Z}}$ by hypothesis. Since we assume that input layers can only compute univariate functions (Def. 8), we have that $\ell$ must be either a sum or product layer.

**Case (iii-a): product layers.** Now, assume $\ell$ is an Hadamard product layer in $c$ receiving input from $\ell_1, \ell_2$. Let $\hat{\mathbf{Z}} = \mathsf{sc}(\ell) \cap \hat{\mathbf{Z}}, \hat{\mathbf{Z}}_1 = \mathsf{sc}(\ell_1) \cap \hat{\mathbf{Z}}, \hat{\mathbf{Z}}_2 = \mathsf{sc}(\ell_2) \cap \hat{\mathbf{Z}}$. From decomposability, we recover that $(\hat{\mathbf{Z}}_1, \hat{\mathbf{Z}}_2)$ is a partitioning of $\hat{\mathbf{Z}}$. We denote as $\hat{\mathbf{y}}$ the variables assignment obtained from $\mathbf{y}$ by restriction to variables in $\mathsf{sc}(\ell) \setminus \hat{\mathbf{Z}}$, and similarly let $\hat{\mathbf{y}}_1, \hat{\mathbf{y}}_2$ denote the variables assignments obtained from $\mathbf{y}$ by restriction to $\mathsf{sc}(\ell_1) \setminus \hat{\mathbf{Z}}_1, \mathsf{sc}(\ell_2) \setminus \hat{\mathbf{Z}}_2$, respectively. Therefore, we can write

$$\int_{\mathsf{dom}(\hat{\mathbf{Z}})} \ell(\hat{\mathbf{y}}, \hat{\mathbf{z}}) \otimes \ell(\hat{\mathbf{y}}, \hat{\mathbf{z}})^* \, \mathrm{d}\hat{\mathbf{z}} = \left( \int_{\mathsf{dom}(\hat{\mathbf{Z}}_1)} \ell_1(\hat{\mathbf{y}}_1, \hat{\mathbf{z}}_1) \otimes \ell_1(\hat{\mathbf{y}}_1, \hat{\mathbf{z}}_1)^* \, \mathrm{d}\hat{\mathbf{z}}_1 \right) \odot \left( \int_{\mathsf{dom}(\hat{\mathbf{Z}}_2)} \ell_2(\hat{\mathbf{y}}_2, \hat{\mathbf{z}}_2) \otimes \ell_2(\hat{\mathbf{y}}_2, \hat{\mathbf{z}}_2)^* \, \mathrm{d}\hat{\mathbf{z}}_2 \right)$$

where we used the Kronecker mixed-product property w.r.t. the Hadamard product, and split the integral into lower dimensional ones. Therefore, L12-16 in our Alg. A.3 use recursion to compute the integrals w.r.t. the layers $\ell_1, \ell_2$ and then aggregate the results with an Hadamard product. In the case of $\ell$ being a Kronecker product layer in $c$ instead, a similar approach can be used, resulting in L18-21 in Alg. A.3. In the case of Kronecker product layers, a permutation matrix described as in Thm. 2 is used.

**Case (iii-b): sum layers.** Let $\ell$ be a sum layer receiving inputs from layers $\mathsf{in}(\ell) = \{\ell_1, \ldots, \ell_N\}$ and parameterized by $\mathbf{W} \in \mathbb{C}^{K_1 \times K_2}$. Assume that $\mathbf{W}$ is a block matrix $\mathbf{W} = [\mathbf{W}^{(1)} \cdots \mathbf{W}^{(N)}]$. Moreover, let $\hat{\mathbf{Z}} = \mathsf{sc}(\ell) \cap \mathbf{Z}, \hat{\mathbf{Y}} = \mathsf{sc}(\ell) \setminus \mathbf{Z}$ and let $\hat{\mathbf{y}}$ denote the variables assignment obtained from $\mathbf{y}$ by restriction to variables in $\mathsf{sc}(\ell) \setminus \hat{\mathbf{Z}}$. For any $i, j \in [N]$, we will denote as $\mathbf{R}_{ij}$ the matrix $\mathbf{R}_{ij} = \int_{\mathsf{dom}(\hat{\mathbf{z}})} \ell_i(\hat{\mathbf{y}}, \hat{\mathbf{z}}) \circ \ell_j(\hat{\mathbf{y}}, \hat{\mathbf{z}})^* \, \mathrm{d}\hat{\mathbf{z}}$. By the satisfaction of unitary and the property (U4), and by applying Lem. A.5 we have that $\forall \ell_i, \ell_j \in \mathsf{in}(\ell), \ell_i \neq \ell_j, \mathbf{R}_{ij} = \mathbf{0}$. Therefore, similarly to our proof for Thm. 2, we recover that

$$\int_{\mathsf{dom}(\hat{\mathbf{Z}})} \ell(\hat{\mathbf{y}}, \hat{\mathbf{z}}) \otimes \ell(\hat{\mathbf{y}}, \hat{\mathbf{z}})^* \, \mathrm{d}\hat{\mathbf{z}} = \int_{\mathsf{dom}(\mathbf{Z})} (\mathbf{W}[\ell_1(\hat{\mathbf{y}}, \hat{\mathbf{z}}) \cdots \ell_N(\hat{\mathbf{y}}, \hat{\mathbf{z}})]) \otimes (\mathbf{W}^*[\ell_1(\hat{\mathbf{y}}, \hat{\mathbf{z}})^* \cdots \ell_N(\hat{\mathbf{y}}, \hat{\mathbf{z}})^*]) \, \mathrm{d}\hat{\mathbf{z}}$$

$$= (\mathbf{W} \otimes \mathbf{W}^*) \int_{\mathsf{dom}(\hat{\mathbf{Z}})} \mathsf{vec}([\ell_1(\hat{\mathbf{y}}, \hat{\mathbf{z}})^* \cdots \ell_N(\hat{\mathbf{y}}, \hat{\mathbf{z}})^*] \circ [\ell_1(\hat{\mathbf{y}}, \hat{\mathbf{z}}) \cdots \ell_N(\hat{\mathbf{y}}, \hat{\mathbf{z}})] \, \mathrm{d}\hat{\mathbf{z}})$$

$$= (\mathbf{W} \otimes \mathbf{W}^*)\mathsf{vec}\left(\begin{bmatrix} \mathbf{R}_{11}^* & & \mathbf{0} \\ & \ddots & \\ \mathbf{0} & & \mathbf{R}_{NN}^* \end{bmatrix}\right) = \mathsf{vec}\left(\mathbf{W}^*\mathsf{diag}(\mathbf{R}_{11}^*, \cdots, \mathbf{R}_{NN}^*)\mathbf{W}^\top\right)$$

$$= \mathsf{vec}\left(\sum_{i=1}^N \mathbf{W}^{(i)*}\mathbf{R}_{ii}^*\mathbf{W}^{(i)\top}\right) = \mathsf{vec}\left(\sum_{i=1}^N \mathbf{W}^{(i)}\mathbf{R}_{ii}\mathbf{W}^{(i)\dagger}\right)^*.$$

L6-11 in our Alg. A.3 recursively marginalize the Kronecker product of $\boldsymbol{\ell}_i$ by its conjugate, for all $i \in [N]$, resulting in matrices $\{\mathbf{R}_{ii}\}_{i=1}^N$ and then performing matrix multiplications and summations.

**Computational complexity.** We recover the overall time complexity stated in the theorem. First, **Case (ii)** has an overall complexity of $\mathcal{O}(|\phi_{\mathbf{Y}} \setminus \phi_{\mathbf{Z}}|S_{\max} + |\phi_{\mathbf{Y}} \cap \phi_{\mathbf{Z}}|S_{\max}^2)$. Second, for **Cases (iii-a)** and **(iii-b)** above, we have that we need to evaluate the integral of a "squared" layer, i.e., the integral $\int_{\mathsf{dom}(\hat{\mathbf{z}})} \boldsymbol{\ell}(\hat{\mathbf{y}}, \hat{\mathbf{z}}) \otimes \boldsymbol{\ell}(\hat{\mathbf{y}}, \hat{\mathbf{z}})^* \, d\hat{\mathbf{z}}$. Thus, these cases account for a quadratic complexity w.r.t. the layer size, i.e., $\mathcal{O}(S_{\max}^2)$ as highlighted in the proof of Prop. A.2, and they occur a number of times that is $|\phi_{\mathbf{Y}} \cap \phi_{\mathbf{Z}}|$. Therefore, we conclude that the overall complexity of our Alg. A.3 is $\mathcal{O}(|\phi_{\mathbf{Y}} \setminus \phi_{\mathbf{Z}}|S_{\max} + |\phi_{\mathbf{Y}} \cap \phi_{\mathbf{Z}}|S_{\max}^2)$. Furthermore, note that we have $|\phi_{\mathbf{Y}} \setminus \phi_{\mathbf{Z}}| + |\phi_{\mathbf{Y}} \cap \phi_{\mathbf{Z}}| = |\phi_{\mathbf{Y}}| \le L$. In other words, the complexity is independent on the number of layers whose scope is a subset of $\mathbf{Z}$, i.e., $|\phi_{\mathbf{Z}} \setminus \phi_{\mathbf{Y}}|$. Although the complexity depends on the particular marginal being computed and the circuit structure chosen, our Alg. A.3 can be much more efficient than $\mathcal{O}(L^2 S_{\max}^2)$. To see this, we consider the following example. The structure for a circuit defined over pixel variables can be built by recursively splitting an image into patches obtained by alternating vertical and horizontal even cuts (Mari et al., 2023; Loconte et al., 2025a). If $\mathbf{Z}$ consists of only the pixel variables in the left-hand side of an image (i.e., we are computing the marginal of the right-hand side $\mathbf{Y}$), then $|\phi_{\mathbf{Y}} \cap \phi_{\mathbf{Z}}|$ is constant w.r.t. $L$ since only a few layers near the circuit output layer will depend on variables both in $\mathbf{Y}$ and $\mathbf{Z}$. The rest of the layers will entirely depend either on $\mathbf{Y}$ or on $\mathbf{Z}$. Therefore, the best-case complexity considers $|\phi_{\mathbf{Y}} \cap \phi_{\mathbf{Z}}|$ being independent of the total number of layers $L$, i.e., it is $\mathcal{O}(|\phi_{\mathbf{Y}} \setminus \phi_{\mathbf{Z}}|S_{\max})$. $\qquad\square$

## A.8 Enforcing Orthogonality is #P-hard

We presented new families of circuits through the introduction of novel circuit properties, namely orthogonality (§3) and unitarity (§4). In general, each family of circuits with a particular parameterization and a set of structural properties they satisfy exhibit a different *expressive efficiency*, which refers to the ability of encoding a function or distribution with a circuit computational graph having polynomial size w.r.t the number of variables (Martens and Medabalimi, 2014). As such many works focused on the formulation of hierarchies that compare different circuit clases in terms of their expressive efficiency (Darwiche and Marquis, 2002; de Colnet and Mengel, 2021; Loconte et al., 2025b). Here, we provide a preliminary expressiveness analysis by investigating which of the presented properties can be enforced in polytime, as this would immediately guarantee no loss expressive efficiency. We start with a negative result, which tells us that enforcing orthogonality is #P-hard.

**Theorem A.3.** Let $c$ be a smooth and decomposable circuit over variables $\mathbf{X}$. Then, constructing a circuit $c'$ from $c$ such that $c'(\mathbf{X}) = c(\mathbf{X})$ and $c'$ is an orthogonal circuit is #P-hard.

*Proof.* The idea is to construct a reduction from the problem of making a smooth and decomposable circuit also orthogonal to #3SAT, which is known to be a #P-hard problem. In particular, we leverage the same technique used to prove that representing any power of a non-structured-decomposable circuit as another decomposable circuit is in general #P-hard (Vergari et al., 2021, Thm. 3.3).

We start by defining the #3SAT problem. Let $\mathbf{X} = \{X_i\}_{i=1}^n$ be a set of Boolean variables, and let $\Phi$ be a CNF formula that contains $m$ clauses $\Gamma = \{c_j\}_{j=1}^m$, where each clause contains exactly 3 literals. The #3SAT problem consists of counting the number of assignments to the variables $\mathbf{X}$ that satisfy $\Phi$, i.e., the quantity $\sum_{\mathbf{x} \in \mathsf{dom}(\mathbf{X})} \Phi(\mathbf{x})$, where $\Phi(\mathbf{x})$ is 1 if $\mathbf{x}$ satisfies $\Phi$ and 0 otherwise. For every variable $X_i$ and for every clause $c_j$, we introduce an auxiliary variable $X_{ij}$. We denote as $\hat{\mathbf{X}}$ the set of all such auxiliary variables, i.e., $\hat{\mathbf{X}} = \{X_{ij} \mid X_i \in \mathbf{X}, c_j \in \Gamma\}$. For every variable $X_i$ we set all auxiliary variables associated to it to share the same Boolean value of $X_i$, which can be described by the logic formula $\beta = \wedge_{X_i \in \mathbf{X}}(X_{i1} \Leftrightarrow X_{i2} \Leftrightarrow \cdots \Leftrightarrow X_{im})$. In order to encode $\Phi$ using an equivalent logic formula defined over the auxiliary variables instead, we introduce the logic formula

$\gamma = \wedge_{c_j \in \Gamma} \vee_{X_i \in \varphi(c_j)} l(X_{ij})$, where we denote as $\varphi(c_j)$ the variables scope of the clause $c_j$ and $l(X_{ij})$ is the literal of $X_i$ found in $c_j$. We can see that $\Phi$ is equivalent to $\beta \wedge \gamma$. As detailed in Khosravi et al. (2019) and Vergari et al. (2021, §A.3), the logic formulae $\beta$ and $\gamma$ can be respectively encoded by structured-decomposable and deterministic circuits $c_\beta$ and $c_\gamma$ having polynomial size. This is done by casting conjunction and disjunctions into products and deterministic sums, respectively. Therefore, $\Phi(\mathbf{x})$ can be computed as the product of the outputs of $c_\beta$ and $c_\gamma$ when evaluated on $\hat{\mathbf{x}}$, which is obtained from the assignments $\mathbf{x}$ as for the sharing of Boolean values between the auxiliary variables of each $X_i$. Crucially, we have that $c_\beta$ and $c_\gamma$ are not compatible circuits by construction.

**Reducing finding an orthogonal circuit to solving #3SAT.** Consider now the circuit $c_\alpha$ computing $c_\alpha(\hat{\mathbf{x}}) = c_\beta(\hat{\mathbf{x}}) + c_\gamma(\hat{\mathbf{x}})$ for any $\hat{\mathbf{x}} \in \mathsf{dom}(\hat{\mathbf{X}})$. Since $c_\beta, c_\gamma$ are structured-decomposable, non-compatible, and with overlapping support for a generic satisfiable logic formula $\Phi$, we have that $c_\alpha$ is a smooth and decomposable circuit that is non-deterministic and non-structured decomposable. Moreover, from the sizes of $c_\alpha$ and $c_\beta$ we recover that $c_\alpha$ has also polynomial size. From now on, we will focus on the problem of computing the quantity $\sum_{\hat{\mathbf{x}} \in \mathsf{dom}(\hat{\mathbf{X}})} c_\alpha(\hat{\mathbf{x}})^2$, called POW2PC in Vergari et al. (2021), and reduce it to solving #3SAT. Formally, by the construction of $c_\alpha$ we have that the POW2PC quantity can be written as

$$\sum_{\hat{\mathbf{x}} \in \mathsf{dom}(\hat{\mathbf{X}})} c_\alpha(\hat{\mathbf{x}})^2 = \sum_{\hat{\mathbf{x}} \in \mathsf{dom}(\hat{\mathbf{X}})} c_\beta(\hat{\mathbf{x}})^2 + \sum_{\hat{\mathbf{x}} \in \mathsf{dom}(\hat{\mathbf{X}})} c_\gamma(\hat{\mathbf{x}})^2 + 2 \sum_{\hat{\mathbf{x}} \in \mathsf{dom}(\hat{\mathbf{X}})} c_\beta(\hat{\mathbf{x}}) c_\gamma(\hat{\mathbf{x}}).$$

Now, since $c_\beta$ and $c_\gamma$ are both deterministic, we observe that $\sum_{\hat{\mathbf{x}} \in \mathsf{dom}(\hat{\mathbf{X}})} c_\beta(\hat{\mathbf{x}})^2$ and $\sum_{\hat{\mathbf{x}} \in \mathsf{dom}(\hat{\mathbf{X}})} c_\gamma(\hat{\mathbf{x}})^2$ can both be computed in polytime (Vergari et al., 2021). Assume by absurdum that there exists a polytime algorithm taking a smooth and decomposable circuit as input, e.g., $c_\alpha$, and that returns an orthogonal circuit computing the same function. Then, from Thm. 1 we have that we would be able to compute $\sum_{\hat{\mathbf{x}} \in \mathsf{dom}(\hat{\mathbf{X}})} c_\alpha(\hat{\mathbf{x}})^2$ in polytime. As a consequence, from the definition of POW2PC, we would be able to compute the remaining quantity $\sum_{\hat{\mathbf{x}} \in \mathsf{dom}(\hat{\mathbf{X}})} c_\beta(\hat{\mathbf{x}}) c_\gamma(\hat{\mathbf{x}})$ in polytime. However, computing this last quantity in polytime would imply solving #3SAT in polytime, since the conjunction of $\beta$ and $\gamma$ is equivalent to the logic formula $\Phi$ as described in the preliminaries above. Therefore, an algorithm receiving a smooth and decomposable circuit as input and converting it to an orthogonal circuit cannot run in polytime. In particular, the problem of representing a smooth and decomposable circuit as an orthogonal one must be at least #P-hard. □

From Thm. A.3 it turns out that enforcing regular orthogonality and unitarity must also be hard, since they both imply orthogonality (see App. A.3 and Thm. 2). However, Thm. A.3 does not necessarily imply an expressiveness *separation* (Martens and Medabalimi, 2014) between orthogonal and smooth and decomposable circuits, i.e., the existence of a family of functions that *cannot* be encoded by any orthogonal and polysize circuit, while it can by a smooth and decomposable circuit. The reason is that Thm. A.3 does not say anything about the minimum circuit sizes required by an orthogonal circuit. While investigating such separation deserves a separate work, here we conjecture it to hold similarly to a known separation between deterministic and non-deterministic circuits (Bova et al., 2016).

In the following, we instead investigate whether the choice of orthonormal input functions and (semi-)unitary weights in tensorized circuits (i.e., only the conditions (U1) and (U3) in unitarity) can restrict their expressiveness when compared to squared PCs that do not satisfy such conditions. First, as we further detail in App. C, there are many choices of orthonormal basis functions that come with guarantees about the families of functions they can arbitrarily approximate. Second, the following theorem guarantees that there is no loss in terms of expressive efficiency from restricting the sum layer parameters to be (semi-)unitary matrices (i.e., (U3)), as it can be enforced in polytime.

### A.9 Enforcing (Semi-)Unitary Parameters is Efficient

**Theorem A.4.** Let $c$ be a smooth and decomposable tensorized circuit over $\mathbf{X}$. There exists an algorithm running in polynomial time returning a circuit $c'$ with (semi-)unitary matrices as sum layer weights, where $c'$ is equivalent to $c$ up to a multiplicative constant, i.e., $c'(\mathbf{X}) = \beta c(\mathbf{X})$, $\beta \geq 0$.

*Proof.* We prove the correctness of our Alg. A.4 to *"unitarize"* the weights of a tensorized circuit, whose idea is to recursively make the circuit parameters (semi-)unitary via QR decompositions. More

---

**Algorithm A.4** UNITARIZE($\boldsymbol{\ell}$)

---

**Input:** A tensorized circuit $c$ over variables $\mathbf{X}$, where $\boldsymbol{\ell}$ is the output layer in $c$.
**Output:** The output layer $\boldsymbol{\ell}'$ of a tensorized circuit $c'$ over $\mathbf{X}$ such that each sum layer in $c'$ has a (semi-)unitary matrix as weight; and a matrix $\mathbf{R} \in \mathbb{C}^{K_1 \times K_2}$, $K_1 \leq K_2$, such that $\boldsymbol{\ell}$ equivalently computes $\mathbf{R}\boldsymbol{\ell}'$ where $K_1, K_2$ are the width of layers $\boldsymbol{\ell}, \boldsymbol{\ell}'$ respectively (i.e., the number of units in the layers $\boldsymbol{\ell}, \boldsymbol{\ell}'$).

1: **if** $\boldsymbol{\ell}$ is an input layer **then**
2:     Assume $\boldsymbol{\ell}$ computes $K$ orthonormal functions
3:     **return** $(\boldsymbol{\ell}, \mathbf{I}_K)$
4: **if** $\boldsymbol{\ell}$ is a sum layer receiving inputs from $\mathsf{in}(\boldsymbol{\ell}) = \{\boldsymbol{\ell}_1, \ldots, \boldsymbol{\ell}_N\}$ and parameterized by $\mathbf{W} \in \mathbb{C}^{K_1 \times K_2}$ **then**
5:     $(\boldsymbol{\ell}_i', \mathbf{R}_i) \leftarrow$ UNITARIZE$(\boldsymbol{\ell}_i), \forall i \in [N]$
6:        where $\mathbf{R}_i \in \mathbb{C}^{J_i \times H_i}$
7:     **let** $\mathbf{W} = [\mathbf{W}^{(1)} \cdots \mathbf{W}^{(N)}] \in \mathbb{C}^{K_1 \times K_2}$
8:        where $\forall i \in [N] : \mathbf{W}^{(i)} \in \mathbb{C}^{K_1 \times J_i}$ and $K_2 = \sum_{i=1}^{N} J_i$
9:     **let** $\mathbf{V} = [\mathbf{W}^{(1)}\mathbf{R}^{(1)} \cdots \mathbf{W}^{(N)}\mathbf{R}^{(N)}] \in \mathbb{C}^{K_1 \times H}$, where $H = \sum_{i=1}^{N} H_i$
10:     Factorize $\mathbf{V}^{\dagger} = \mathbf{Q}\mathbf{R}$, where $\mathbf{Q}$ is (semi-)unitary and $\mathbf{R}$ is upper triangular
11:     **let** $\boldsymbol{\ell}'$ be a sum layer computing $\mathbf{Q}^{\dagger}[\boldsymbol{\ell}_1' \cdots \boldsymbol{\ell}_N']$
12:     **return** $(\boldsymbol{\ell}', \mathbf{R}^{\dagger})$
13: **if** $\boldsymbol{\ell}$ is a Kronecker product layer with inputs $\boldsymbol{\ell}_1, \boldsymbol{\ell}_2$ **then**
14:     $(\boldsymbol{\ell}_1', \mathbf{R}_1) \leftarrow$ UNITARIZE$(\boldsymbol{\ell}_1)$, $\mathbf{R}_1 \in \mathbb{C}^{K_1 \times K_2}$
15:     $(\boldsymbol{\ell}_2', \mathbf{R}_2) \leftarrow$ UNITARIZE$(\boldsymbol{\ell}_2)$, $\mathbf{R}_2 \in \mathbb{C}^{K_3 \times K_4}$
16:     **let** $\boldsymbol{\ell}'$ be a layer computing $\boldsymbol{\ell}_1' \otimes \boldsymbol{\ell}_2'$
17:     **return** $(\boldsymbol{\ell}', \mathbf{R}_1 \otimes \mathbf{R}_2)$                         $\triangleright$ $\otimes$: Kronecker matrix product
18: **if** $\boldsymbol{\ell}$ is an Hadamard product layer with inputs $\boldsymbol{\ell}_1, \boldsymbol{\ell}_2$ **then**
19:     $(\boldsymbol{\ell}_1', \mathbf{R}_1) \leftarrow$ UNITARIZE$(\boldsymbol{\ell}_1)$, $\mathbf{R}_1 \in \mathbb{C}^{K_1 \times K_2}$
20:     $(\boldsymbol{\ell}_2', \mathbf{R}_2) \leftarrow$ UNITARIZE$(\boldsymbol{\ell}_2)$, $\mathbf{R}_2 \in \mathbb{C}^{K_1 \times K_3}$
21:     **let** $\boldsymbol{\ell}'$ be a layer computing $\boldsymbol{\ell}_1' \otimes \boldsymbol{\ell}_2'$
22:     **return** $(\boldsymbol{\ell}', \mathbf{R}_1 \bullet \mathbf{R}_2)$                    $\triangleright$ $\bullet$: Face-splitting matrix product (see Def. A.8)

---

formally, Alg. A.4 is used to retrieve a tensorized circuit $c'$ from $c$ such that $c'(\mathbf{X}) = \beta c(\mathbf{X})$ for a non-negative constant $\beta$, where the weight matrices of sum layers in $c'$ are (semi-)unitary. To do so, we take inspiration from the unitarization (or canonization) algorithm in tree-shaped tensor networks (TTNs) (Shi et al., 2006; Orús, 2013; Cheng et al., 2019; Krämer, 2020). That is, the idea of Alg. A.4 is to recursively apply QR decompositions to make the weight matrices of sum layers (semi-)unitary, while still preserving the function computed by the circuit up to a non-negative multiplicative constant $\beta$. However, differently from the canonization algorithm in (TTNs), our Alg. A.4 generalizes to hierarchical tensor factorizations when represented by circuits (Loconte et al., 2025a) (see §§3 and 4).

**Assumptions.** In the proof we are going to assume that each sum layer receives inputs from product layers, and that each product layer receives inputs from either two input layers or two sum layers. These assumptions are without loss of generality, as they can be enforced in polynomial time without changing the function computed by $c$ and with at most a polynomial increase in circuit size. For instance, if a product layer receives inputs from another product layer, then we can "interleave" these product layers by introducing a sum layer whose parameter matrix is an identity matrix. Now, given $\boldsymbol{\ell}$ the output layer of a tensorized circuit, we will show by structural induction that Alg. A.4 returns a pair $(\boldsymbol{\ell}', \mathbf{R})$, where $\boldsymbol{\ell}'$ is the output layer of the tensorized circuit $c'$, and $\mathbf{R}$ is a matrix such that $\boldsymbol{\ell}$ equivalently computes the matrix-vector product $\mathbf{R}\boldsymbol{\ell}'$. In particular, $\mathbf{R}$ will have as many rows as the number of units in $\boldsymbol{\ell}$ (i.e., its layer width) and as many columns as the number of units in $\boldsymbol{\ell}'$. Moreover, we will have that the sum layers in the sub-circuit rooted in $\boldsymbol{\ell}'$ have (semi-)unitary matrices as parameters. Therefore, when Alg. A.4 is applied to the output layer of $c$, then $\mathbf{R}$ is a $1 \times 1$ matrix containing the value of $\beta$ as stated above. We proceed by cases below.

**Case (i): input layer.** The base case is when $\boldsymbol{\ell}$ is an input layer in $c$. Assume that $\boldsymbol{\ell}$ computes the value of $K$ functions. Then, L1-3 in Alg. A.4 returns $\boldsymbol{\ell}$ unchanged, i.e., $\boldsymbol{\ell}' = \boldsymbol{\ell}$ and it sets $\mathbf{R}$ to be the $K \times K$ identity matrix $\mathbf{I}_K$. We also have that the circuit rooted in $\boldsymbol{\ell}'$ does not have sum layers and therefore it trivially satisfies the requirement that the weights of the sub-circuit rooted in $\boldsymbol{\ell}'$ must be (semi-)unitary.

**Case (ii): sum layer.** Let $\boldsymbol{\ell}$ be a sum layer receiving input from layers $\mathsf{in}(\boldsymbol{\ell}) = \{\boldsymbol{\ell}_i\}_{i=1}^{N}$ and computing the matrix-vector product $\boldsymbol{\ell} = \mathbf{W}[\boldsymbol{\ell}_1 \cdots \boldsymbol{\ell}_N]$, where $\mathbf{W} \in \mathbb{C}^{K_1 \times K_2}$ denote the parameter matrix of $\boldsymbol{\ell}$. By inductive hypothesis, for each $\boldsymbol{\ell}_i, i \in [N]$, Alg. A.4 returns the output layer $\boldsymbol{\ell}_i'$ of circuit whose weights are (semi-)unitary, and a matrix $\mathbf{R}_i \in \mathbb{C}^{J_i \times H_i}$ with $K_2 = \sum_{i=1}^{N} J_i$, where $H_i$ is the

number of units in the layer $\boldsymbol{\ell}_i'$. That is, we have that $\boldsymbol{\ell}_i$ equivalently computes $\mathbf{R}_i\boldsymbol{\ell}_i'$. We denote $\mathbf{W}$ as the block matrix $\mathbf{W} = [\mathbf{W}^{(1)} \cdots \mathbf{W}^{(N)}]$ where $\mathbf{W}^{(i)} \in \mathbb{C}^{K_1 \times J_i}$, and we can rewrite the function computed by $\boldsymbol{\ell}$ as follows

$$\boldsymbol{\ell} = \mathbf{W}[\boldsymbol{\ell}_1 \cdots \boldsymbol{\ell}_N] = \sum_{i=1}^{N} \mathbf{W}^{(i)}\boldsymbol{\ell}_i = \sum_{i=1}^{N} \mathbf{W}^{(i)}\mathbf{R}_i\boldsymbol{\ell}_i' = \mathbf{V}[\boldsymbol{\ell}_1' \cdots \boldsymbol{\ell}_N'],$$

where we set $\mathbf{V} = [\mathbf{V}^{(1)} \cdots \mathbf{V}^{(N)}] \in \mathbb{C}^{K_1 \times H}$, with $H = \sum_{i=1}^{N} H_i$, and such that each block is $\mathbf{V}^{(i)} = \mathbf{W}^{(i)}\mathbf{R}_i \in \mathbb{C}^{K_1 \times H_i}$. To retrieve a (semi-)unitary matrix, we perform the QR decomposition on $\mathbf{V}^\dagger$, thus retrieving $\mathbf{V}^\dagger = \mathbf{QR}$. Now, in the following we distinguish two cases based on whether $\mathbf{V}^\dagger$ is a wide or tall matrix.

$$\mathbf{V}^\dagger = \begin{cases} \mathbf{QR} & \text{where } \mathbf{Q} \in \mathbb{C}^{H \times H}, \mathbf{R} \in \mathbb{C}^{H \times K_1} \quad \mathbf{V}^\dagger \text{ is wide or square, i.e., } H \leq K_1 \\ \mathbf{QR} & \text{where } \mathbf{Q} \in \mathbb{C}^{H \times K_1}, \mathbf{R} \in \mathbb{C}^{K_1 \times K_1} \quad \mathbf{V}^\dagger \text{ is tall, i.e., } H > K_1 \end{cases}$$

Moreover, in the wide or square case we have that $\mathbf{Q}^\dagger\mathbf{Q} = \mathbf{I}_H$, while in the tall case we have that $\mathbf{Q}^\dagger\mathbf{Q} = \mathbf{I}_{K_1}$. In both cases, we have that $\mathbf{R}$ is upper triangular. Thus, we can rewrite the function computed by $\boldsymbol{\ell}$ as

$$\boldsymbol{\ell} = \mathbf{V}[\boldsymbol{\ell}_1' \cdots \boldsymbol{\ell}_N'] = \mathbf{R}^\dagger\mathbf{Q}^\dagger[\boldsymbol{\ell}_1' \cdots \boldsymbol{\ell}_N'] = \mathbf{R}^\dagger\boldsymbol{\ell}'$$

where $\boldsymbol{\ell}'$ is a sum layer parameterized by the (semi-)unitary matrix $\mathbf{Q}^\dagger$ and computing $\mathbf{Q}^\dagger[\boldsymbol{\ell}_1' \cdots \boldsymbol{\ell}_N']$. Therefore, L4-12 in Alg. A.4 returns $(\boldsymbol{\ell}', \mathbf{R}^\dagger)$, and we have that the sub-circuit rooted in $\boldsymbol{\ell}'$ has (semi-)unitary matrices as the weights of sum layers. Finally, we observe that the number of sum units in $\boldsymbol{\ell}'$—or equivalently the number of rows in $\mathbf{Q}^\dagger$—is $\min(H, K_1)$ whatever $\mathbf{V}^\dagger$ is a wide, square or tall matrix. Therefore, the number of units in $\boldsymbol{\ell}'$ is bounded by the number of units $K_1$ in $\boldsymbol{\ell}$. Similarly, the size of the matrix $\mathbf{R}^\dagger$ returned by Alg. A.4 is at most of size $K_1 \times K_1$. Instead, in the particular case of $\mathbf{V}^\dagger$ being tall, i.e., $H > K_1$, we notice that $\mathbf{Q}^\dagger \in \mathbb{C}^{K_1 \times H}$ can possibly be larger than $\mathbf{W} \in \mathbb{C}^{K_1 \times K_2}$, which would account for an increase in the circuit size. As we detail below, this increase in circuit size is still polynomial, as it can only occur in the case of Hadamard product layers and it is bounded to be at most quadratic w.r.t. the original circuit size.

**Case (iii): Hadamard product layer.** Let $\boldsymbol{\ell}$ be a Hadamard product layer computing $\boldsymbol{\ell} = \boldsymbol{\ell}_1 \odot \boldsymbol{\ell}_2$. By inductive hypothesis, let $\boldsymbol{\ell}_1'$ and $\boldsymbol{\ell}_2'$ be the output layers of tensorized circuits obtained by recursively applying Alg. A.4 on $\boldsymbol{\ell}_1$ and $\boldsymbol{\ell}_2$, respectively. Moreover, let $\mathbf{R}_1 \in \mathbb{C}^{K_1 \times K_2}$ and $\mathbf{R}_2 \in \mathbb{C}^{K_1 \times K_3}$ be the matrices obtained via Alg. A.4 w.r.t. $\boldsymbol{\ell}_1$ and $\boldsymbol{\ell}_2$. That is, we have that $\boldsymbol{\ell}_1$ (resp. $\boldsymbol{\ell}_2$) equivalently computes $\mathbf{R}_1\boldsymbol{\ell}_1'$ (resp. $\mathbf{R}_2\boldsymbol{\ell}_2'$). For this reason, we can rewrite the function computed by $\boldsymbol{\ell}$ as

$$\boldsymbol{\ell} = (\mathbf{R}_1\boldsymbol{\ell}_1') \odot (\mathbf{R}_2\boldsymbol{\ell}_2') = (\mathbf{R}_1 \bullet \mathbf{R}_2)(\boldsymbol{\ell}_1' \otimes \boldsymbol{\ell}_2'),$$

where we used the Hadamard mixed-product property, and $\bullet$ denotes the face-splitting matrix product.

**Definition A.8** (Face-splitting matrix product). Let $\mathbf{A} \in \mathbb{C}^{m \times k}$ and $\mathbf{B} \in \mathbb{C}^{m \times r}$ be matrices. The face-splitting product $\mathbf{A} \bullet \mathbf{B}$ is defined as the matrix $\mathbf{C} \in \mathbb{C}^{m \times kr}$,

$$\mathbf{C} = \begin{bmatrix} \mathbf{a}_1 \otimes \mathbf{b}_1 \\ \vdots \\ \mathbf{a}_m \otimes \mathbf{b}_m \end{bmatrix} \quad \text{where} \quad \mathbf{A} = \begin{bmatrix} \mathbf{a}_1 \\ \vdots \\ \mathbf{a}_m \end{bmatrix} \quad \mathbf{B} = \begin{bmatrix} \mathbf{b}_1 \\ \vdots \\ \mathbf{b}_m \end{bmatrix},$$

and $\{\mathbf{a}_i\}_{i=1}^{m}, \{\mathbf{b}_i\}_{i=1}^{m}$ are row vectors.

L18-22 in Alg. A.4 constructs a Kronecker layer $\boldsymbol{\ell}'$ in $c'$ computing $\boldsymbol{\ell}' = \boldsymbol{\ell}_1' \otimes \boldsymbol{\ell}_2'$, i.e., $\boldsymbol{\ell}$ equivalently computes $(\mathbf{R}_1 \bullet \mathbf{R}_2)\boldsymbol{\ell}'$. Thus, L22 returns returns both $\boldsymbol{\ell}'$ and the matrix $\mathbf{R}_1 \bullet \mathbf{R}_2$. By inductive hypothesis, we have that the circuits rooted in $\boldsymbol{\ell}_1'$ and $\boldsymbol{\ell}_2'$ have sum layers with (semi-)unitary weights, thus also the circuit rooted in $\boldsymbol{\ell}'$ must have. Furthermore, we observe that an Hadamard product layer is replaced by a Kronecker ones, resulting in a quadratic increase in circuit size. To avoid the size of the layers in $c'$ and the matrices $\mathbf{R}$ being returned by Alg. A.4 to grow exponentially in the particular case of subsequent Hadamard product layers in $c$, the assumptions made at the beginning of this proof become here useful. As stated above, we can efficiently "interleave" consecutive Hadamard layers in $c$ by using sum layers having identity matrices as parameters. By doing so and as observed in **Case (ii)** above for sum layers, the size of the matrices $\mathbf{R}$ returned by Alg. A.4 and the number of units in

each sum layer being built in $c'$ remain bounded. Therefore, this effectively bounds the size of the Kronecker layers being built in $c'$ from Hadamard layers in $c$.

**Case (iv): Kronecker product layer.** Let $\boldsymbol{\ell}$ be a Kronecker product layer computing $\boldsymbol{\ell} = \boldsymbol{\ell}_1 \otimes \boldsymbol{\ell}_2$. By inductive hypothesis, let $\boldsymbol{\ell}'_1$ and $\boldsymbol{\ell}'_2$ be the output layers of tensorized circuits obtained by recursively applying Alg. A.4 on $\boldsymbol{\ell}_1$ and $\boldsymbol{\ell}_2$, respectively. Moreover, let $\mathbf{R}_1 \in \mathbb{C}^{K_1 \times K_2}$ and $\mathbf{R}_2 \in \mathbb{C}^{K_3 \times K_4}$ be the matrices obtained via Alg. A.4 w.r.t. $\boldsymbol{\ell}_1$ and $\boldsymbol{\ell}_2$. That is, we have that $\boldsymbol{\ell}_1$ (resp. $\boldsymbol{\ell}_2$) equivalently computes $\mathbf{R}_1\boldsymbol{\ell}'_1$ (resp. $\mathbf{R}_2\boldsymbol{\ell}'_2$). For this reason, we can rewrite the function computed by $\boldsymbol{\ell}$ as

$$\boldsymbol{\ell} = (\mathbf{R}_1\boldsymbol{\ell}'_1) \otimes (\mathbf{R}_2\boldsymbol{\ell}'_2) = (\mathbf{R}_1 \otimes \mathbf{R}_2)(\boldsymbol{\ell}'_1 \otimes \boldsymbol{\ell}'_2),$$

where we use the Kronecker mixed-product property. That is, we retrieve a Kronecker layer $\boldsymbol{\ell}'$ in $c'$ computing $\boldsymbol{\ell}' = \boldsymbol{\ell}'_1 \otimes \boldsymbol{\ell}'_2$, i.e., $\boldsymbol{\ell}$ equivalently computes $(\mathbf{R}_1 \otimes \mathbf{R}_2)\boldsymbol{\ell}'$. Thus, L13-17 in Alg. A.4 returns both $\boldsymbol{\ell}'$ and the matrix $\mathbf{R}_1 \otimes \mathbf{R}_2$. By inductive hypothesis, we have that the circuits rooted in $\boldsymbol{\ell}'_1$ and $\boldsymbol{\ell}'_2$ have sum layers with (semi-)unitary weights, thus also the circuit rooted in $\boldsymbol{\ell}'$ must have.

**Case (v): output layer.** We consider the case of $\boldsymbol{\ell}$ being the output layer in $c$, thus resulting in the last step of our Alg. A.4. Without loss of generality, we consider $\boldsymbol{\ell}$ being a sum layer. Then from our **Case (ii)** above, we have that $\mathbf{R} \in \mathbb{C}^{1 \times 1}$ is obtained by the QR decomposition of a column vector $\mathbf{V}^\dagger \in \mathbb{C}^{K \times 1}$, thus corresponding to the scalar $r_{11}$ such that $||r_{11}\mathbf{V}^\dagger||_2 = 1$, i.e., $r_{11} = ||\mathbf{V}^\dagger||_2^{-1} = \left(\sum_{i=1}^K |v_{i1}|^2\right)^{-\frac{1}{2}}$. Therefore, the non-negative scalar $\beta$ mentioned in the theorem must be exactly $\beta = r_{11}$.

**A note on the value of $\beta$ and on unitarity.** Assume that $c$ satisfies (U1) and (U2) of unitarity. Since Alg. A.4 does not change the input layers and the dependencies of the sum layer inputs to the input layers, we have that $c'$ also satisfies (U1) and (U2). Therefore, $c'$ is unitary because it satisfies conditions (U1) to (U3), and thus from Thm. 2 we have that the modulus squaring of $c'$ is an already normalized distribution, i.e., $p(\mathbf{X}) = |c'(\mathbf{X})|^2 = \beta^2|c(\mathbf{X})|^2$. This means that, under the assumptions of (U1) and (U2), the value of $\beta$ is exactly $\beta = Z^{-\frac{1}{2}}$ with $Z$ being the partition function of the modulus squaring of $c$, i.e., $Z = \int_{\mathsf{dom}(\mathbf{X})} |c(\mathbf{x})|^2 \, \mathrm{d}\mathbf{x}$. Finally, Thm. A.4 can be seen as the dual of another result about monotonic PCs shown by Peharz et al. (2015): they show an algorithm that updates the positive weights of a smooth and decomposable PC such that the distribution it encodes is already normalized, while Alg. A.4 updates the complex weights of a circuit such that its modulus squaring is an already-normalized distribution.

**A note on the computational complexity.** We now analyze the computational complexity of Alg. A.4. We observe that the complexity mainly depends on the complexity of performing QR decompositions and computing Kronecker (or face-splitting) products of matrices. In particular, we need to perform as many QR decompositions as the number of sum layers in $c$, each requiring time $\mathcal{O}(K_1^2 H)$ in the case of a wide matrix $\mathbf{V} \in \mathbb{C}^{K_1 \times H}$ and $\mathcal{O}(K_1 H^2)$ in the tall matrix case. Now, let $K_{\max}$ denote the maximum number of units in a layer in $c$. By the way the matrix $\mathbf{V}$ is computed (see **Case (ii)** above) and since the Hadamard layer is the only case accounting to a quadratic increase in layer width (i.e., transforming Hadamard into Kronecker and leveraging the face-splitting product), we have that $K_1 \le K_{\max}$ and $H \le K_{\max}^2$. As such, the complexity of performing the QR factorizations will be $\mathcal{O}(LK_{\max}^4)$, where $L$ is the number of layers in $c$. Similarly, the complexity of computing Kronecker and face-splitting products as in **Cases (iii-iv)** above is $\mathcal{O}(K_{\max}^4)$. Overall, the complexity of our Alg. A.4 is $\mathcal{O}(LK_{\max}^4)$. $\qquad\square$

## B  TREE TENSOR NETWORKS AS STRUCTURED-DECOMPOSABLE CIRCUITS

In this section, we show how the complete contraction of a tree-shaped tensor network (TTN) (Shi et al., 2006) can be encoded by a particular class of structured-decomposable tensorized circuits (Def. 8), where product layers are Kronecker products. By a very similar argument, one can see how also other TN structures, such as MPS (Schollwoeck, 2010) and tensor rings (Zhao et al., 2016), can be encoded by structured-decomposable circuits. Although a result representing the hierarchical Tucker tensor factorization (Grasedyck, 2010) as a circuit was already formally shown in Loconte et al. (2025a),[1] in App. B.1 we connect this construction to a canonical form of TTNs ensuring normalization of the distribution modeled via modulus squaring.

---

[1]Hierarchical Tucker tensor is essentially a TTN having a binary tree structure in Penrose graphical notation.

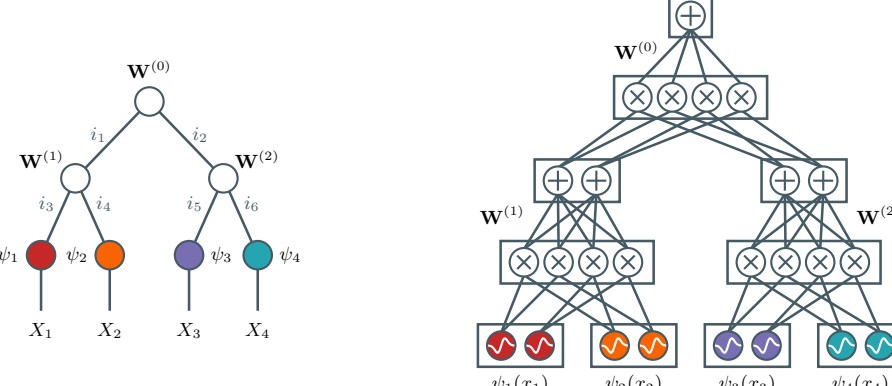

Figure B.1: **Tree tensor networks (TTNs) represented as tensorized circuits.** The contraction of a TTN over variables $\mathbf{X} = \{X_1, \ldots, X_4\}$ (left, in Penrose graphical notation) starting from the factors at the bottom towards the root node can be encoded by a tensorized circuit having Kronecker product layers, and whose sum layers are parameterized by the non-leaf tensors, i.e., $\mathbf{W}^{(0)}, \mathbf{W}^{(1)}, \mathbf{W}^{(2)}$ (right). Due to the densely connected structure, the circuit is *not* basis decomposable (Def. 6), as the products that are input to the output sum depend on the same input functions (in colors). Similarly to the circuit corresponding to the MPS factorization shown in Fig. 1, this circuit is structured-decomposable because the products encode a single hierarchical partitioning of the variables.

**From TTNs to tensorized circuits.** When compared to matrix-product states (MPS) TNs (Pérez-García et al., 2007), TTNs come with the advantage of better capturing longer variables sequence correlations by using a hierarchical tree-like structure (Murg et al., 2010; Seitz et al., 2022). Formally, let $\mathbf{X} = \{X_j\}_{j=1}^d$ be a set of variables, and for each $X_j \in \mathbf{X}$ let $\Psi_j = \{\psi_j^k \colon \mathrm{dom}(X_j) \to \mathbb{C}\}_{k=1}^R$ be a set of *factors* for the variable $X_j$, where $R$ is the factorization rank of the TTN. For simplicity, here we consider the case of binary TTNs, i.e., whose structure in Penrose graphical notation is a binary tree, but the following discussion can be translated to other TTNs as well. A rank-$R$ binary TTN factorization defines the following decomposition of $\psi(\mathbf{X})$.

$$\psi(\mathbf{x}) = \sum_{i_1=1}^R \sum_{i_2=1}^R \cdots \sum_{i_{2N}=1}^R w_{i_1 i_2}^{(0)} \left( \prod_{n=1}^{N-1} w_{i_n i_{2n+1} i_{2(n+1)}}^{(n)} \right) \left( \prod_{j=1}^d \psi_j^{i_{d-2+j}}(x_j) \right), \qquad (9)$$

where $\mathbf{W}^{(0)} \in \mathbb{C}^{R \times R}$ and for all $n \in [N-1] \colon \mathbf{W}^{(n)} \in \mathbb{C}^{R \times R \times R}$, with $N$ being the total number of inner tensors in the TTNs, i.e., $N = d - 1$ in this binary tree case. For example, Fig. B.1 (left) illustrates a TTN over $d = 4$ variables, which encodes the following factorization of $\psi(\mathbf{X})$.

$$\psi(x_1, x_2, x_3, x_4) = \sum_{i_1=1}^R \sum_{i_2=1}^R \cdots \sum_{i_6=1}^R w_{i_1 i_2}^{(0)} w_{i_1 i_3 i_4}^{(1)} w_{i_2 i_5 i_6}^{(2)} \psi_1^{i_3}(x_1) \psi_2^{i_4}(x_2) \psi_3^{i_5}(x_3) \psi_4^{i_6}(x_4) \quad (10)$$

The complete contraction of a TTN following a bottom-up topological ordering can be encoded by a tensorized circuit. In order to give an intuition of this, we focus on the example in Eq. (10). We reorder summations and multiplications in Eq. (10) as in the equation below, which corresponds to a contraction ordering that starts from the factor leaves and proceeds towards the root tensor of the TTN (i.e., the matrix $\mathbf{W}^{(0)}$).

$$\psi(x_1, x_2, x_3, x_4) =$$

$$\sum_{i_1=1}^R \sum_{i_2=1}^R w_{i_1 i_2}^{(0)} \left( \sum_{i_3=1}^R \sum_{i_4=1}^R w_{i_1 i_3 i_4}^{(1)} \psi_1^{i_3}(x_1) \psi_2^{i_4}(x_2) \right) \left( \sum_{i_5=1}^R \sum_{i_6=1}^R w_{i_2 i_5 i_6}^{(2)} \psi_3^{i_5}(x_3) \psi_4^{i_6}(x_4) \right)$$

$$(11)$$

In other words, we pushed the outer summations as inside as possible in the TTN factorization formula. By doing so, we recover three groups of sums and products that contract the indices $\{i_1, i_2\}$,

$\{i_3, i_4\}$ and $\{i_5, i_6\}$, respectively in red, green and blue colors in Eq. (11). In order to build a tensorized circuit $c$ encoding this contraction, i.e., $c(\mathbf{X}) = \psi(\mathbf{X})$, we construct one input layer $\boldsymbol{\ell}_j^{\text{in}}$ for each variable $X_j \in \mathbf{X}$ computing the corresponding factors in $\Psi_j$ as a $R$-dimensional vector. That is is, for all $X_j \in \mathbf{X}$ we have that $\boldsymbol{\ell}_j^{\text{in}}$ computes $\boldsymbol{\ell}_j^{\text{in}}(x_j) = [\psi_j^1(x_j) \cdots \psi_j^R(x_j)]^\top$. We then observe from Eq. (11) that the composition of groups of sums and products can be encoded by a hierarchical composition of sum layers and Kronecker product layers. Formally, let $\hat{\mathbf{W}}^{(0)}$ denote the reshaping of $\mathbf{W}^{(0)}$ as a $1 \times R^2$ matrix, and similarly let $\hat{\mathbf{W}}^{(1)}$ and $\hat{\mathbf{W}}^{(2)}$ respectively denote the reshaping of $\mathbf{W}^{(1)}$ and $\mathbf{W}^{(2)}$ as $R \times R^2$ matrices. Then, we can rewrite Eq. (11) as

$$\psi(x_1, x_2, x_3, x_4) = \hat{\mathbf{W}}^{(0)} \left[ \hat{\mathbf{W}}^{(1)} \left( \boldsymbol{\ell}_1^{\text{in}}(x_1) \otimes \boldsymbol{\ell}_2^{\text{in}}(x_2) \right) \right] \otimes \left[ \hat{\mathbf{W}}^{(2)} \left( \boldsymbol{\ell}_3^{\text{in}}(x_3) \otimes \boldsymbol{\ell}_4^{\text{in}}(x_4) \right) \right].$$
(12)

The above can be equivalently encoded by a tensorized circuit $c$, as we illustrate in Fig. B.1 (right). That is, the Kronecker products are computed by Kronecker layers in $c$, and the matrix-vector multiplications are computed by sum layers respectively parameterized by the matrices $\hat{\mathbf{W}}^{(0)}$, $\hat{\mathbf{W}}^{(1)}$, $\hat{\mathbf{W}}^{(2)}$. We also observe that the corresponding circuit $c$ is structured-decomposable, as we can interpret its structure as encoding a single hierarchical partitioning of the set of variables $\mathbf{X}$ it is defined on, i.e., $\mathbf{X}$ is partitioned into $\{X_1, X_2\}$ and $\{X_3, X_4\}$ by the Kronecker product layers, and in turn these are split towards the univariate input layers respectively over $\{X_1\}, \{X_2\}$ and $\{X_3\}, \{X_4\}$.

Next, we connect our orthogonality conditions defined over circuits with a popular TTN canonical form—sometimes called upper-canonical form (Cheng et al., 2019)—which ensures that the corresponding Born machine encodes an already-normalized distribution. That is, we show that this canonical form in TTNs is a particular case of unitarity, i.e., the corresponding tensorized circuit satisfies the conditions (U1) to (U3) shown in §4. We then make some observations on how unitary tensorized circuits can represent a strictly larger set of hierarchical factorizations when compared to TTNs, which instead can only be structured-decomposable by construction (see §2 and App. B).

## B.1 Unitary Circuits Generalize Upper Canonical Tree Tensor Networks

**The upper-canonical form is a special case of unitarity.** The *upper canonical* form of a TTN consists of two assumptions on the factors and the inner tensors. That is, we require the factors over the same variable to be orthonormal, and that each inner tensor is an isometry w.r.t. the two indices pointing downwards. More formally, a TTN is upper canonical if it satisfies the following conditions.

$$\forall X_j \in \mathbf{X}, \forall k_1, k_2 \in [R]: \int_{\text{dom}(X_j)} \psi_j^{k_1}(x_j) \psi_j^{k_2}(x_j)^* \, dx_j = \delta_{k_1 k_2} \tag{13}$$

$$\sum_{i_2=1}^{R} w_{i_1, i_2}^{(0)} w_{j_1, i_2}^{(0)*} = \delta_{i_1 j_1}, \text{ and } \forall n \in [N-1]: \sum_{i_{2n+1}=1}^{R} \sum_{i_{2(n+1)}=1}^{R} w_{i_n i_{2n+1} i_{2(n+1)}}^{(n)} w_{j_n i_{2n+1} i_{2(n+1)}}^{(n)*} = \delta_{i_n j_n} \tag{14}$$

It is possible to show that these two conditions ensure that the partition function of the corresponding Born machine obtained by modulus squaring of $\psi$ is $Z = \int_{\text{dom}(\mathbf{X})} |\psi(\mathbf{x})|^2 \, d\mathbf{x} = 1$ (Cheng et al., 2019; Seitz et al., 2022). We interpret Eqs. (13) and (14) as conditions defined over the input layers and weight matrices of the tensorized circuit $c$ described by Eq. (12). That is, under upper canonicity of the TTNs, we recover that each input layer $\boldsymbol{\ell}_j^{\text{in}}$ over the variable $X_j \in \mathbf{X}$ satisfies $\int_{\text{dom}(X_j)} \boldsymbol{\ell}_j^{\text{in}}(x_j) \otimes \boldsymbol{\ell}_j^{\text{in}}(x_j)^* \, dx_j = \mathbf{I}_R$. For this reason, the tensorized circuit $c$ satisfies (U1) of unitarity. Furthermore, we have that $c$ satisfies (U2) of unitarity trivially, since by construction each sum layer receives input from exactly one other layer, i.e., a Kronecker product. Finally, we can equivalently rewrite Eq. (14) as $\hat{\mathbf{W}}^{(0)} \hat{\mathbf{W}}^{(0)\dagger} = \mathbf{I}_R$, $\hat{\mathbf{W}}^{(1)} \hat{\mathbf{W}}^{(1)\dagger} = \mathbf{I}_R$, and $\hat{\mathbf{W}}^{(2)} \hat{\mathbf{W}}^{(2)\dagger} = \mathbf{I}_R$. In other words, each sum layer in $c$ is parameterized by a (semi-)unitary matrix, thus it satisfies (U3) of unitarity. Therefore, we conclude that the tensorized circuit $c$ encoding the same upper canonical TTN is unitarity.

**Going beyond TTNs with unitarity.** As also stressed in §§3 and 4, tensorized circuits can represent a strictly larger set of hierarchical factorizations when compared to TTNs. This is because TTNs are

a particular instance of tensorized circuits that are structured-decomposable. Instead, non-structured-decomposable tensorized circuits can encode multiple hierarchical partitionings of variables, e.g., see the circuit in Fig. 4. Despite this crucial difference w.r.t. TTNs, the modulus squaring of a non-structured-decomposable tensorized circuit can still encode a normalized distribution via unitarity (as for Thm. 2), and it supports the tractable computation of marginals (as for Thm. 3). In particular, theoretical results in circuit complexity have shown that structured-decomposable circuits can be exponentially less expressive than non-structured-decomposable ones (Pipatsrisawat and Darwiche, 2008; 2010; de Colnet and Mengel, 2021). For this reason, our contributions motivate future work aimed at developing novel TN structures different from TTNs that can possibly be more expressive, yet they support tractable marginalization and sampling via unitarity.

## C  RELATED WORK

**The relationship between circuits and TNs.** To the best of our knowledge, Ko et al. (2020) was the first work linking ideas from both TNs and from particular circuits known as sum-product networks (SPNs) (Poon and Domingos, 2011), by showing an approach to approximate sparse SPNs into non-negative MPS TNs (Glasser et al., 2019). Representing popular factorization methods such as CP (Carroll and Chang, 1970; Harshman, 1970), hierarchical Tucker (Grasedyck, 2010) and MPS TNs as circuits was later highlighted in Loconte et al. (2023; 2025b;a). In particular, casting TN contractions into a composition of sums and products is analogous to performing variable elimination in a graphical model (Koller and Friedman, 2009; Glasser et al., 2018), whose implementation can also be encoded by a circuit (Darwiche and Provan, 1996; Darwiche, 2003; 2009). The property-driven framework of circuits provides sufficient and necessary conditions to compose them in operations and enable the computation of quantities in closed-form, such as expectations and information-theoretic measures (Vergari et al., 2021; Wang et al., 2024). Recently, determinism has been generalized as a property between two circuits in Wang et al. (2024) to bring complexity simplifications for exact causal inference and weighted model counting (Chavira and Darwiche, 2008). Similarly, we believe one can extend our orthogonality (§3) as a property between two circuits, thus possibly simplifying the computation of compositional operations while being possibly less restrictive than determinism. Note that these properties and operations can be translated to TNs as well, as for their close relationship with circuits (§2). Furthermore, in some cases one can efficiently restructure the hierarchical variables decomposition implicitly encoded by a structured-decomposable circuit (and thus TNs) (Zhang et al., 2025b)—also called vtree (Pipatsrisawat and Darwiche, 2008; Kisa et al., 2014)—thus enabling the efficient renormalization of the product of certain non-compatible circuits.

**Canonical forms of TNs** exploit parameterizations in terms of (semi-)unitary matrices to unlock many practical advantages (Schollwoeck, 2010). Among these, canonical forms provide simplifications for the computation of certain physical quantities (Orús, 2013), as well as the computation of marginal and conditional probabilities (Bonnevie and Schmidt, 2021) by ensuring the modeled distribution is normalized. By connecting with circuit determinism, we provide novel conditions defined in the circuit language to unlock similar advantages, namely ensuring squared PCs encode already-normalized distributions (§4) and to enable fast marginalization (§5 and App. A.1). Moreover, TNs expressed in canonical forms come with an enhanced numerical stability, support optimization methods aimed at avoiding vanishing and exploding gradients (Sun et al., 2020), and are amenable to advanced Riemannian optimization techniques (Hauru et al., 2020; Luchnikov et al., 2021). These practical advantages can be translated to circuits as well. Furthermore, popular TNs such as MPS and TTNs can be efficiently turned into a particular canonical form by iteratively performing either SVD or QR decompositions (Shi et al., 2006; Orús, 2013; Cheng et al., 2019; Krämer, 2020). Our algorithm to make the parameters of a circuit (semi-)unitary (Thm. A.4) takes inspiration from procedures to make a TN canonical, but generalizes to tensorized circuits.

**Possible choices of orthonormal functions.** Depending on whether a variable is discrete or continuous, we have different ways to encode it with orthonormal functions. For a variable $X$ with domain $\text{dom}(X) = [v]$, any function $f(X)$ can be expressed as $\sum_{k=1}^{v} f(k)\delta_{xk}$, i.e., $f$ can be written in terms of $v$ Kronecker deltas $\{\delta_{xk}\}_{k=1}^{v}$ that are orthonormal. That is, $\sum_{x \in [v]} \delta_{xk}\delta_{xk'} = \delta_{kk'}$ for $k, k' \in [v]$. For a continuous variable $X$, many function families can be expressed in terms of orthonormal basis. E.g., periodic functions can be represented by Fourier series (Jackson, 1941) and, under certain continuity conditions, functions can be approximated arbitrarily well by finite Fourier partial sums (Jackson, 1930). Furthermore, certain families of functions can also be described in terms of orthogonal polyno-

mials (Abramowitz et al., 1965), e.g., Hermite functions generalize Gaussians and form an orthonormal basis of square-integrable functions over all $\mathbb{R}$ (Roman and Rota, 1978). More in general, any set of linearly independent functions can be described as a linear projection of a set of orthonormal basis functions (see Meiburg et al. (2025, §A)). In practice, different choices of orthonormal functions and polynomials have been used in signal processing (Pinheiro and Vidakovic, 1997), score-based variational inference (Cai et al., 2024) and also in TNs modeling density functions (Meiburg et al., 2025). Recently, orthonormal functions have been used in circuits to better scale polynomial chaos expansion (Wiener, 1938) for uncertainty quantification analysis to high dimensions (Exenberger et al., 2025).

**Learning (semi-)unitary matrices.** Many works in the deep learning community have investigated the challenging problem of learning on the manifold of (semi-)unitary matrices, also called the Stiefel manifold (Absil et al., 2007). That is, there are many ways of parameterizing unitary matrices, with different advantages regarding efficiency, numerical stability and generality (Arjovsky et al., 2015; Huang et al., 2017; Bansal et al., 2018; Lezcano-Casado and Martínez-Rubio, 2019), which could be employed for learning the parameters of squared PCs. More recently, Hauru et al. (2020); Luchnikov et al. (2021) proposed optimizing the parameters of MPS TNs and quantum gates using Riemmanian optimization approaches (Kochurov et al., 2020).

**Many models supporting tractable marginalization** have been recently introduced. These include squared neural families with real or complex parameters that square the 2-norm of the output of a single-hidden-layer neural network (Tsuchida et al., 2023; 2024; 2025). In addition to squared PCs and mixtures thereof (Loconte et al., 2024; 2025b), other models are based on squared circuit representations. These are PSD circuits (Sladek et al., 2023) inspired from PSD kernel methods (Marteau-Ferey et al., 2020; Rudi and Ciliberto, 2021), and Inception PCs generalizing structured-decomposable monotonic and squared PCs (Wang and Van den Broeck, 2025). Recently, Zellinger et al. (2026) developed expectation estimators using squared PCs in the context of variational inference and importance sampling. Furthermore, Zuidberg Dos Martires (2025) has recently unified these circuit families under a single formalism—*positive unital circuits* (PUnCs)—based on concepts from quantum information theory (Nielsen and Chuang, 2010). The realization of the squaring of a structured-decomposable circuit as yet another decomposable circuit is subsumed by PUnCs. However, differently from PUnCs where the layer activations are $K \times K$ PSD matrices, a unitary squared PC admits a more memory efficient representation *by means of a circuit that does not necessarily require being squared*, i.e., whose layer activations are $K$-dimensional vectors instead. While PUnCs has been proposed also as a way to construct non-structured-decomposable non-monotonic PCs, we find that ensuring either orthogonality (§3) or unitarity (§§4 and 5) is sufficient for it in non-structured-decomposable squared PCs instead.

**About expressiveness**. The satisfaction of either orthogonality or unitarity allows us to build squared PCs that are *not* structured-decomposable, yet they still enable the tractable computation of marginals (§§3 and 5). Since popular TN structures such as MPS and TTNs are encoded by structured-decomposable circuit by construction (App. B), our contribution motivates future works aimed at understanding how non-structured-decomposable squared PCs are related to structured-decomposable ones in terms of expressive efficiency. We believe that answering to these questions might require techniques that are different to the ones used to prove separations between circuits and squared PCs that are structured-decomposable (de Colnet and Mengel, 2021; Loconte et al., 2024; 2025b). Furthermore, as shown by Agarwal and Bläser (2024) and Oliver Broadrick (2024), other instances of non-monotonic PCs that are *not* squared include determinantal point processes (Kulesza and Taskar, 2012; Zhang et al., 2020) and probabilistic generating circuits (Zhang et al., 2021; Harviainen et al., 2023). Understanding the relationship in terms of expressiveness also w.r.t. these other non-monotonic PCs and PUnCs (Zuidberg Dos Martires, 2025) is an interesting direction.

# D  EXPERIMENTAL DETAILS

In this section, we describe all the necessary details to reproduce the results from §6.

**Computational resources.** To run all experiments, we use a cluster of 8 NVIDIA L40S GPUs and 8 NVIDIA RTX A6000 GPUs managed via Slurm. Every experiment uses a single GPU. For the benchmark experiments (see App. D.3), we make sure all experiments run in isolation in one of the NVIDIA RTX A6000, to properly compare their time and memory requirements.

**Implementation and sanity checks.** Our implementations of squared unitary PCs are based on the `cirkit` library (The APRIL Lab, 2025), and extend a previous code base for squared PCs with complex parameters by Loconte et al. (2025b). Since the proposed unitary parameterization ensure squared PCs encode already-normalized distributions, we do not materialize their square as another decomposable circuit in order to compute the partition function. This allows us to efficiently train squared PCs by maximum-likelihood even in those cases where materializing the squared PC would be too expensive memory-wise, e.g., when using Kronecker product layers. As we do not compute the partition function explicitly, *how do we make sure that the squared unitary PCs in our software implementation encode a normalized distribution?* Besides the theory presented in this manuscript, we employed several units tests to check that the distribution modeled by squared unitary PCs integrates to one via numerical integration on randomly-initialized circuits with Kronecker product layers, as well as in the case of non-structured-decomposable circuits having different sizes. As a result, we have empirically corroborated that our implementation of squared unitary PCs indeed model normalized distributions up to unavoidable numerical errors due to floating point precision.

**Squared PCs families.** In the following we denote as $\pm^2_{\mathbb{C}}$ the class of squared PCs with complex parameters, while we use $\perp^2_{\mathbb{C}}$ to denote the class of squared unitary PCs.

## D.1  CONTINUOUS INPUT FEATURES

In this section, we perform some preliminary experiments assessing the expressiveness of orthogonal input functions in the case where we have continuous variables, as discussed in App. C and §4.

**Fourier input functions.** To this end, for each input function in the circuit we use one single term of a Fourier series with equal periodicity across input functions. That is, if we have $2K + 1$ input functions—we assume an odd number of them—then each input function $f_k$ is of the form $f_k(x) \propto \exp(2\pi i \frac{k}{P}(x+b))$, where $i \in \mathbb{C}$ is the imaginary unit, $k \in \{-K, -K+1, \ldots, K\}$, $P$ is the same for every $k$ and larger than the size of $\mathsf{dom}(X)$, and $b \in \mathbb{R}$ is a learnable bias term. As a result, all input functions are orthogonal between then, and we make them orthonormal by normalizing them, such that they integrate out to one. We set $P = 6$ for the spinner dataset, and $P = 12$ for the spiral dataset.

**Experimental setting.** We take two synthetic datasets from the official code released by Loconte et al. (2025b), and train different circuit architectures to perform distribution estimation. To this end, in each iteration we sample a new batch of size 1024 from the synthetic generator function, and add noise from

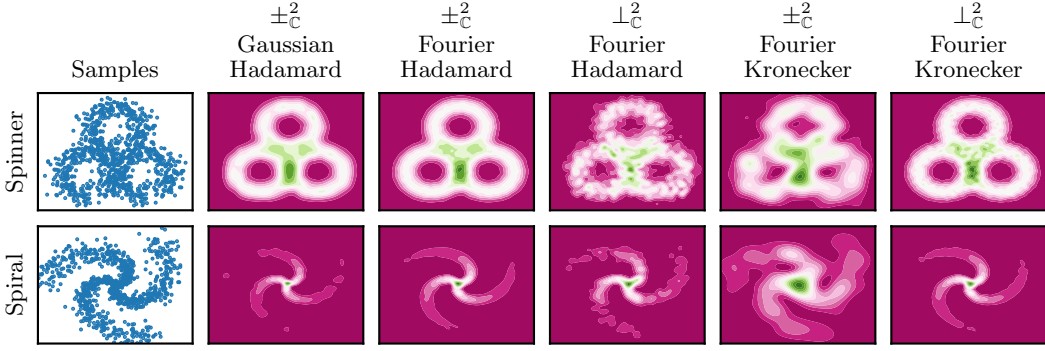

Figure D.1: Densities estimated by different combinations of circuit classes, product layers, and input layers (one per column), fitted with samples from two different synthetic datasets (one per row).

a centered Gaussian with a standard deviation of $0.1$ to avoid overfitting. We train all models for a thousand iterations and apply cosine learning rate annealing to avoid instabilities at the end of training.

**Circuits.** We take a baseline a squared PC with Hadamard product layers and Gaussian input functions, and then test the Fourier input functions in all combinations of usual and unitary squared PCs with Hadamard or Kronecker product layers, i.e., the configurations in $\{\pm_{\mathbb{C}}^{2}, \perp_{\mathbb{C}}^{2}\} \times \{\text{Hadamard}, \text{Kronecker}\}$. For regular squared PCs we use Adam (Kingma and Ba, 2015) with learning rate 0.001, and our LandingPC (see App. E) for squared unitary PCs with learning rate 0.01 and $\lambda = 0.1$. For every combination we use 21 input and sum units, except for $\pm_{\mathbb{C}}^{2}$ with Hadamard layers for which we keep them at 7, since otherwise the model takes too long to run.

**Results.** We show the estimated densities for each model combination and dataset in Fig. D.1. Despite the simplicity of the synthetic datasets at hand, we find a couple of interesting insights. First, we see that the Fourier layers do outperform the fitting (at least, qualitatively) of the identical same circuit with Gaussian input functions. Therefore, validating the expressivity claims regarding the input functions made at the end of §4. Second, we find that certain combinations work better than others. Specifically, $\pm_{\mathbb{C}}^{2}$ seem to work better when combined with Hadamard product layers, while $\perp_{\mathbb{C}}^{2}$ do particularly well with Kronecker layers. This is consistent with the fact that a multivariate Fourier series definition considers all possible product combinations of univariate complex exponentials (Smith and Smith, 1995), i.e., similarly to a Kronecker product. In addition, we later validate again in App. D.2 that Hadamard layers do not seem to work well with $\perp_{\mathbb{C}}^{2}$.

## D.2 Image Distribution Estimation

Here we describe the details for the experiments on image distribution estimation, as well as present some additional results that complement the findings from the main text.

**Datasets.** For the distribution estimation experiments with image data, we employ the MNIST (LeCun et al., 2010) and FashionMNIST (Xiao et al., 2017) datasets composed of, respectively, digits and clothing black-and-white pictures of size $28 \times 28 \, \text{px}$, yielding a total of $784$ input features. We treat each of these inputs as Categorical inputs with 256 classes (one for each of the grayscale intensity values). We randomly reserve $5\%$ of the training dataset split for validation.

**Building structured-decomposable circuit architectures.** One way of easily construct smooth and decomposable (Def. 2) circuit architectures is by parameterizing via product and sum units a hierarchical partitioning of the variables scope. This hierarchical variables partitioning—known as *region graph* (Dennis and Ventura, 2012)—recursively splits a set of variables $\mathbf{X}$ into disjoint sets, which provides a "skeleton" for the circuit architecture. In other words, a region graph tells us how the product units will split their scope towards their inputs, thus guaranteeing the satisfaction of decomposability. A region graph whose structure is constrained to be a tree is analogous to *mode cluster trees* as in hierarchical factorization methods (Grasedyck, 2010), which also guarantees the corresponding circuit is structured-decomposable (Pipatsrisawat and Darwiche, 2008). Now, following Peharz et al. (2020b;a); Loconte et al. (2025a) we build tensorized circuits by (i) instantiating a region graph and (ii) parameterizing each variables partitioning node in the region graph by adding a product layer followed by a sum layer. This guarantees that the resulting tensorized circuit is smooth ad decomposable. To build structured-decomposable circuits over image pixel variables, we consider a tree-shaped region graph called *quad-tree*, which is obtained by recursively splitting the image into four even and aligned patches (Mari et al., 2023). Our baseline and unitary squared PCs ($\pm_{\mathbb{C}}^{2}$ and $\perp_{\mathbb{C}}^{2}$, respectively) are based on this region graph.

**Building non-structured-decomposable circuit architectures.** In addition to structured-decomposable squared unitary PCs, we experiment with non-structured-decomposable ones, i.e., squared PCs that satisfy our conditions (U1) to (U4) and whose structure encode multiple variable partitionings (unlike TTNs, e.g., see Fig. 4). Differently from the quad-tree region graph, which only results in structured-decomposable circuits, we devise a new region graph by considering multiple ways to recursively split an image patch into smaller patches. Formally, given a set of image pixel variables $\mathbf{X}$, we partition them into two distinct ways by splitting the image either horizontally or vertically, resulting in partitions $(\mathbf{X}_{\text{above}}, \mathbf{X}_{\text{below}})$ and $(\mathbf{X}_{\text{left}}, \mathbf{X}_{\text{right}})$, respectively. We do the same recursively for each obtained image patch $\mathbf{X}_{\text{above}}, \mathbf{X}_{\text{below}}, \mathbf{X}_{\text{left}}, \mathbf{X}_{\text{right}}$, until either the patch height or width is too small w.r.t. a certain threshold (we choose our minimum patch width and height to be 8), or the

patch is composed of a single pixel. This approach of constructing a region graph for images by considering multiple ways of splitting the same patch recursively is similar to other ones in the circuit literature (Poon and Domingos, 2011; Peharz et al., 2020a; Mari et al., 2023). However, these approaches construct one input layer for each pixel variable, whose outputs are then shared by different parts of the non-structured-decomposable circuit architecture. Instead, in order to ensure orthogonality between input layers over the same variable (i.e., to satisfy our conditions (U1) and (U2) and (U4)) we do not share the embedding input layers and parameterize them such that they are pairwise orthonormal. Note that, since the goal of the work is showing that one can train squared non-structured-decomposable squared unitary PCs supporting tractable marginalization, we did not focus on the construction of the region graph. We believe that our results for non-structured-decomposable squared unitary PCs (Fig. D.2) could be further improved by exploring different ways of constructing their architecture.

**Hyperparameters.** In our experiments whose results are shown in Fig. 5b and Fig. D.2, we vary the number of computational units in each input and sum layer by ensuring that all models have a comparable number of trainable parameters. That is, for circuits build from the quad-tree region graph, we consider $\{16, 32, 64, 128, 256, 512\}$ units in the case of squared PCs with Hadamard layers and $\{4, 6, 8, 10, 12, 14, 16\}$ units in the case of Kronecker layers. For the non-structured-decomposable squared PCs with Kronecker layers, we also consider $\{18, 20, 22\}$ units per layer. For the baseline squared PCs, we tune the models where we consider Adam with a learning rate of $0.01$ (which were the best hyperparameters found by Loconte et al. (2025b)), as well as SGD with learning rates $0.01$ and $0.001$. For squared unitary PCs, we consider LandingPC* with learning rate of $0.05$ and LandingSGD* with a momentum of $0.9$ and learning rates $0.01$ and $0.001$, where we fix $\lambda = 0.1$ always. Here, an asterisk denotes LandingSGD with one or both modifications described in App. E. Finally, for unitary circuits we also consider LandingSGD as described by Ablin et al. (2024) and the same hyperparameter as described before for LandingSGD*.

**Additional settings.** To provide a broader view on the design choices made in our experimental section, we expand in Fig. D.2 the plot from Fig. 5b with more combinations of product layers and optimizers. The first observation is that the performances of baseline squared PCs heavily relies on the optimizer: despite testing various learning rates, we could not obtain satisfactory results using SGD as the optimizer. Second, we observe that indeed squared unitary PCs do not play well with Hadamard product layers: for every optimizer and circuit size we tried, their performance is significantly worse that the best models. Finally, we see the importance of the adjustments made to the LandingSGD algorithm (Ablin and Peyré, 2022) described in App. E: while the original algorithm (LandingSGD) does not perform well, by projecting back to the Stiefel manifold each time a matrix goes too far from it (LandingSGD*) we significantly improve their performance, yet they struggle with the larger circuits. Then, by replacing the Euclidean gradient in the algorithm (LandingPC*) we strictly improve the performance of the trained unitary circuits in every setting we tested.

Table D.1: Distribution estimation performances of a squared PC and a unitarity squared PC on the MNIST dataset (LeCun et al., 2010) as we increase the number of layer units. Performance shows mean and standard deviation across three random initializations.

| Circuit class | Product layer | Optimizer | # units | # params ($\times 10^6$) | Test performance (bpd) |
|---|---|---|---|---|---|
| $\pm^2_{\mathbb{C}}$ | Hadamard | Adam | 16 | 6.5577 | $1.3071 \pm 0.0105$ |
| | | | 32 | 13.3858 | $1.2676 \pm 0.0068$ |
| | | | 64 | 27.8529 | $1.2518 \pm 0.0033$ |
| | | | 128 | 60.0312 | $1.2337 \pm 0.0011$ |
| | | | 256 | 137.3640 | $1.2147 \pm 0.0015$ |
| | | | 512 | 343.9340 | $1.1991 \pm 0.0004$ |
| $\perp^2_{\mathbb{C}}$ | Kronecker | LandingPC (see App. E) | 4 | 2.1353 | $1.3112 \pm 0.0001$ |
| | | | 6 | 6.4260 | $1.2567 \pm 0.0003$ |
| | | | 8 | 20.1339 | $1.2328 \pm 0.0005$ |
| | | | 10 | 55.6461 | $1.2201 \pm 0.0005$ |
| | | | 12 | 133.2764 | $1.2064 \pm 0.0005$ |
| | | | 14 | 283.2467 | $1.1998 \pm 0.0015$ |
| | | | 16 | 547.6680 | $1.1923 \pm 0.0007$ |

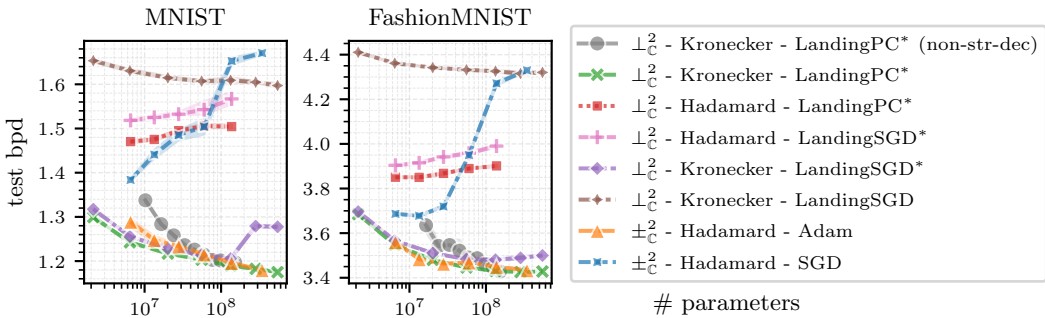

Figure D.2: Image distribution estimation experiment from Fig. 5b with additional settings. Specifically, we show normal and unitary squared PCs using Hadamard or Kronecker product layers, and using different optimizers. An asterisk denotes LandingSGD with one or both modifications described in App. E, referring to the latter as LandingPC. The plot is computed over three random initializations.

Table D.2: Distribution estimation performances of a squared PC and a unitarity squared PC on the FashionMNIST dataset (Xiao et al., 2017) as we increase the number of layer units. Performance shows mean and standard deviation across three random initializations.

| Circuit class | Product layer | Optimizer | # units | # params $(\times 10^6)$ | Test performance (bpd) |
|---|---|---|---|---|---|
| $\pm_{\mathbb{C}}^2$ | Hadamard | Adam | 16 | 6.5577 | $3.5451 \pm 0.0020$ |
| | | | 32 | 13.3858 | $3.4689 \pm 0.0019$ |
| | | | 64 | 27.8529 | $3.4479 \pm 0.0008$ |
| | | | 128 | 60.0312 | $3.4545 \pm 0.0044$ |
| | | | 256 | 137.3640 | $3.4349 \pm 0.0024$ |
| | | | 512 | 343.9340 | $3.4199 \pm 0.0012$ |
| $\perp_{\mathbb{C}}^2$ | Kronecker | LandingPC (see App. E) | 4 | 2.1353 | $3.6700 \pm 0.0008$ |
| | | | 6 | 6.4260 | $3.5325 \pm 0.0005$ |
| | | | 8 | 20.1339 | $3.4651 \pm 0.0008$ |
| | | | 10 | 55.6461 | $3.4306 \pm 0.0016$ |
| | | | 12 | 133.2764 | $3.4148 \pm 0.0007$ |
| | | | 14 | 283.2467 | $3.4105 \pm 0.0009$ |
| | | | 16 | 547.6680 | $3.4128 \pm 0.0028$ |

## D.3 BENCHMARKING SQUARED PCS

In this section, we briefly describe the experimental details regarding the benchmark results plotted in Fig. 5a, as well as provide the quantitative results of said experiment, see Tab. D.3.

**Experimental setting.** We keep the experimental setting as close as possible to that from App. D.2, meaning that we use the same circuit architectures as there. To increase the number of parameters, we increase the number of units in input and sum layers, as we report in Tab. D.3. To provide reliable timings and peak GPU memory measurement, we simulate a single optimization step (or training iteration) that minimizes the negative log-likelihood computed over one batch of data points. That is, we measure time and peak GPU memory required to evaluate the input and inner layers, as well as to perform the backpropagation step and parameters update using a particular optimizer (SGD, Adam and LandingPC (App. E)). Finally, we average the results over 50 training iterations and perform 10 initial burn-in iterations to discard initial artifacts and overheads.

**Results.** In Tab. D.3 we report time and peak GPU memory measurements illustrated in Fig. 5a in tabular format, for both squared PCs ($\pm_{\mathbb{C}}^2$) and squared unitary PCs ($\perp_{\mathbb{C}}^2$). As discussed in §6, the unitary parameterization in squared PCs permits us to not materialize the squared PC as a decomposable circuit in order to compute the partition function (as it is fixed to 1) required by the

negative log-likelihood loss. As such, the unitary parameterization together with the LandingPC optimizer (App. E) brings computationally cheaper parameter updates, when compared to baseline squared PCs learned using either SGD or Adam as optimizer.

Table D.3: Time and memory consumption of different combinations of squared PCs and optimizers for a single training iteration. We find that squared unitary PCs are faster and use less memory than their counterparts, even if they employ Kronecker product layers.

| Circuit class | Product layer | Optimizer | # units | # params ($\times 10^6$) | GPU Mem. (GiB) | Time (ms/iter) |
|---|---|---|---|---|---|---|
| $\pm_{\mathbb{C}}^2$ | Hadamard | SGD | 8 | 0.0873 | 0.0889 | 0.0366 |
| | | | 16 | 0.3492 | 0.1647 | 0.0374 |
| | | | 32 | 1.3968 | 0.3348 | 0.0371 |
| | | | 64 | 5.5870 | 0.7497 | 0.0337 |
| | | | 128 | 22.3479 | 1.8779 | 0.0443 |
| | | | 256 | 89.3914 | 5.3277 | 0.1285 |
| | | | 512 | 357.5649 | 17.0012 | 0.5226 |
| $\pm_{\mathbb{C}}^2$ | Hadamard | Adam | 8 | 0.0873 | 0.0893 | 0.0366 |
| | | | 16 | 0.0349 | 0.1660 | 0.0374 |
| | | | 32 | 1.3968 | 0.3400 | 0.0375 |
| | | | 64 | 5.5870 | 0.7705 | 0.0375 |
| | | | 128 | 22.3479 | 1.9611 | 0.0459 |
| | | | 256 | 89.3914 | 5.6607 | 0.1337 |
| | | | 512 | 357.5649 | 18.3332 | 0.5426 |
| $\perp_{\mathbb{C}}^2$ | Hadamard | LandingPC (see App. E) | 8 | 0.0873 | 0.0866 | 0.0170 |
| | | | 16 | 0.3492 | 0.1555 | 0.0171 |
| | | | 32 | 1.3968 | 0.2984 | 0.0173 |
| | | | 64 | 5.5870 | 0.6040 | 0.0199 |
| | | | 128 | 22.3479 | 1.2953 | 0.0303 |
| | | | 256 | 89.3914 | 2.9979 | 0.0737 |
| | | | 512 | 357.5649 | 11.0130 | 0.2666 |
| $\perp_{\mathbb{C}}^2$ | Kronecker | LandingPC (see App. E) | 8 | 11.2108 | 0.8895 | 0.0364 |
| | | | 10 | 34.1124 | 1.9109 | 0.0508 |
| | | | 12 | 84.7711 | 3.7685 | 0.0848 |
| | | | 14 | 183.0993 | 6.9384 | 0.1695 |
| | | | 16 | 356.8435 | 12.0623 | 0.2923 |

## E    THE FAMILY OF LANDING ALGORITHMS

Here, we briefly describe the family of Landing optimization algorithms used to learn semi-unitary matrices, as well as the modifications we performed to train squared unitary PCs. Refer to the original works to see a full description of the LandingSGD algorithm, its variants, as well as their theoretical properties (Ablin and Peyré, 2022; Ablin et al., 2024).

Say that we want to optimize one of the matrices $\mathbf{W} \in \mathbb{R}^{n \times p}$ with $n > p$ of a circuit, constraining $\mathbf{W}$ to lie in the Stiefel manifold, i.e. such that $\mathbf{W}^\top \mathbf{W} = \mathbf{I}_p$. The LandingSGD algorithm (Ablin and Peyré, 2022) will then produce a sequence of iterates as follows:

$$\mathbf{W}_{t+1} \coloneqq \mathbf{W}_t - \eta \Lambda(\mathbf{W}_t) \tag{15}$$

where $\Lambda$ is the landing field defined as

$$\Lambda(\mathbf{W}) \coloneqq \mathrm{grad} f(\mathbf{W}) + \lambda \mathbf{W}(\mathbf{W}^\top \mathbf{W} - \mathbf{I}_p) \tag{16}$$

and where $\mathrm{grad} f(\mathbf{W}) = \mathrm{skew}(\nabla f(\mathbf{W})\mathbf{W}^\top)\mathbf{W}$ is the relative gradient (i.e. gradient in the tangent space of the non-singular matrix manifold with respect to multiplicative noise, rather than additive) of the loss function $f$ we are trying to optimize. The second term of the last equation can also be

---

**Algorithm E.1** The original LandingSGD algorithm (Ablin and Peyré, 2022).

---

**Input:** The matrix $\mathbf{W}$, its gradient $\nabla f(\mathbf{W})$, a momentum buffer $\mathbf{A}$ (initiated as $\nabla f(\mathbf{W})$), and the iteration $t$.
**Hyper-parameters:** Learning rate $\eta$, momentum $\gamma$, weight decay $\psi$, dampening $\upsilon$, attraction strength $\lambda$, safe step $\epsilon$, stabilization steps $T$.

1: **let** $\mathbf{g} = \nabla f(\mathbf{W}) + \psi \mathbf{W}$          ▷ Weight decay
2: **let** $\mathrm{grad}\, f(\mathbf{W}) = \mathrm{skew}(\nabla f(\mathbf{W})\mathbf{W}^\top)\mathbf{W}$          ▷ Relative gradient (in the manifold)
3: **if** $\gamma > 0$ **then**
4:      **let** $\mathbf{A} = \gamma \mathbf{A} + (1 - \upsilon)\mathbf{g}$          ▷ Momentum
5:      **let** $\mathbf{g} = \mathbf{A}$
6:      **if** use Nesterov momentum **then**
7:          **let** $\mathbf{g} = \nabla f(\mathbf{W}) + \gamma \mathbf{A}$
8:      **let** $\nabla \mathcal{N}(\mathbf{W}) = \lambda \mathbf{W}(\mathbf{W}^\top \mathbf{W} - \mathbf{I}_p)$          ▷ Normal direction (towards the manifold)
9:      **if** $\epsilon > 0$ **then**          ▷ Compute safe step size
10:          **let** $d = ||\mathbf{W}^\top \mathbf{W} - \mathbf{I}_p||_F$
11:          **let** $r = ||\mathbf{g} + \nabla \mathcal{N}(\mathbf{W})||_F$
12:          **let** $\eta^* = (-\lambda d(d-1) + \sqrt{\lambda^2 d^2 (d-1)^2 + r^2 \max(0, \epsilon - d)})/(r^2 + 1e^{-8})$
13:          **let** $\eta = \min(\eta^*, \eta)$
14:      **let** $\mathbf{W} = \mathbf{W} - \eta \mathbf{g}$
15:      **if** $t \bmod T = 0$ **then**
16:          **let** $\mathbf{W} = (\mathbf{W}^\top \mathbf{W})^{-\frac{1}{2}} \mathbf{W}$          ▷ Project back to the manifold
17:          **let** $\mathbf{A} = \mathbf{A} - \mathbf{W}\mathbf{A}^\top \mathbf{W}$          ▷ Project to the tangent space of $\mathbf{W}$

---

seen as the gradient of the distance of the matrix $\mathbf{W}$ to the manifold, $\nabla \mathcal{N}(\mathbf{W})$, where $\mathcal{N}(\mathbf{W}) = \frac{1}{4}||\mathbf{W}^\top \mathbf{W} - \mathbf{I}_p||^2$. In turns out that the two terms of the sum above are actually *orthogonal*, and thus the landing algorithm can be understood as the combination of an $f$-informed force (the relative gradient) and an attractive force which pulls the iterates towards the Stiefel manifold.

From the base algorithm introduced by Ablin and Peyré (2022), and described in Alg. E.1, Ablin et al. (2024) generalize it and introduce its stochastic version, LandingSGD, as well as another variant for variance reduction. In order to make the algorithm work on our setting, we took our own spin and modified LandingSGD. Namely, we introduced two main changes: (i) we replace the Euclidean gradient $\nabla f(\mathbf{W})$ to be the one given by the result of combining VectorAdam (Ling et al., 2022) and RAdam (Liu et al., 2019); and (ii) project the gradient back to the manifold if we find their distance to exceed the same threshold $\epsilon$ as the one given to the LandingSGD algorithm. To distinguish it from the original algorithm, in this manuscript we refer to the final algorithm after the aforementioned adjustments as LandingPC. The change of gradient does not break the theoretical guarantees of landing algorithms since the generalized analysis by Ablin et al. (2024) works as long as $\mathrm{grad}\, f(\mathbf{W})$ is skew-symmetric, which is still the case if we replace $\nabla f(\mathbf{W})$.

