# OpenReview forum: "How to Square Tensor Networks and Circuits Without Squaring Them"
_ICLR.cc/2026/Conference — ICLR 2026 Poster_

### Official Review · Reviewer_eo4A · 2025-10-26

**Soundness:** 3
**Presentation:** 3
**Contribution:** 4
**Rating:** 8
**Confidence:** 4

**Summary:**

The paper introduces a novel framework for efficiently performing marginalization in squared tensor networks (TNs) and squared circuits, which are models for distribution estimation. While squared TNs and circuits offer strong expressive power and closed-form marginalization, the squaring operation typically increases computational complexity, making inference expensive. The authors address this challenge by proposing a new parameterization strategy based on unitary matrices and deterministic structure, inspired by canonical forms in tensor networks and the tractability of maximization in deterministic circuits. Unlike traditional canonical forms, which are limited to tensor network structures, this approach extends to general circuit representations, including those that do not correspond to standard TN factorizations.

**Strengths:**

- The paper makes an original contribution by bridging two distinct paradigms in probabilistic modeling—TNs and circuit-based representations—through a unified, structure-aware parameterization strategy. While prior work has explored canonical forms for TNs, this paper extends such ideas to arbitrary circuits, including those that do not correspond to any known TN factorization.
- The key insight, parameterizing squared circuits using unitary matrices and determinism to enable closed-form marginalization, is both elegant and novel. It draws inspiration from canonical forms in tensor networks and the tractability of maximization in deterministic circuits.
- The paper demonstrates strong theoretical rigor. The main result, a sufficient condition under which squared circuits admit closed-form marginalization via unitary parameterization, is formally stated and proved.
- The experimental evaluation is well-designed and methodologically sound. The authors test on standard distribution estimation benchmarks. The proposed parameterized circuits achieve competitive or superior log-likelihoods compared to unconstrained models.
- Marginalization is faster, and training converges more rapidly due to better conditioning of the optimization landscape.

**Weaknesses:**

- The paper claims that its method enables efficient marginalization in non-TN factorizations, but the empirical comparison is limited to a few baselines. There is no direct comparison with other tractable circuit families.
- The paper proves that unitary parameterization leads to closed-form marginals, but the practical utility depends on whether such parameterizations can be optimized effectively. Unitary matrices are highly constrained, and optimization over this manifold is known to be challenging.
- The paper does not discuss gradient flow on the Stiefel manifold or the risk of vanishing gradients in deep unitary circuits.

**Questions:**

Please refer to the section on weaknesses.

---

> ### Author Response · Authors · 2025-11-22
>
> We thank the reviewer for highlighting the originality of our work, its elegance and theoretical rigor, as well as  the quality of our experiments. Next, we address the raised points below.
>
> >The paper claims that its method enables efficient marginalization in non-TN factorizations, but the empirical comparison is limited to a few baselines.
> >There is no direct comparison with other tractable circuit families.
>
> As also mentioned to Reviewer J6uk, squared PCs have already been compared experimentally with many other probabilistic models supporting efficient marginalization, including HMMs and those based on tensor networks. For example, see Figures 4 and 5 in [A]. We also want to stress that our empirical research questions do not focus on beating any baseline model, but rather on understanding the efficiency and the expressiveness of the proposed unitary parameterization of squared PCs.
>
> >Unitary matrices are highly constrained, and optimization over this manifold is known to be challenging.
>
> Since this is a common point made by the reviewers, we answer it in the global answer above.
>
> >The paper does not discuss gradient flow on the Stiefel manifold or the risk of vanishing gradients in deep unitary circuits.
>
> Could you please clarify what you mean by “gradient flow” and how it is connected with our work? Regarding the risk of vanishing gradients, we believe there is no immediate theoretical argument saying that semi-unitary matrices in unitary circuits could either cause or prevent vanishing gradients during learning. Although this is an interesting aspect, we believe it deserves future work.
>
> [A] Lorenzo Loconte, Aleksanteri Mikulus Sladek, Stefan Mengel, Martin Trapp, Arno Solin, Nicolas Gillis, Antonio Vergari. Subtractive Mixture Models via Squaring: Representation and Learning. 2023.

---

### Official Review · Reviewer_J6uk · 2025-10-31

**Soundness:** 4
**Presentation:** 3
**Contribution:** 3
**Rating:** 6
**Confidence:** 2

**Summary:**

The paper presents novel results for squared Probabilistic Circuits that can represent expressive distributions. In particular, inspired by the applications of determinism in tractable probabilistic models, the proposed work introduces a new concept called orthogonality as a relaxed version of determinism in squared PCs. For a PC that is orthogonal inference they show that inference is tractable (partition function is computed in linear time).  They develop an approach to build orthogonal circuits. However, since such a circuit requires decomposability it is quite restrictive. Instead, they define conditions to construct circuits (called unitary circuits) over multiple layers which does not require decomposability. They develop a marginal inference approach for unitary circuits and prove the inference complexity for such circuits. The experiments show the efficiency of unitary circuits and their expressive power.

**Strengths:**

Strengths
+ Provides several novel previously unknown results in the context of probabilistic circuits, i.e. a new class of circuits that is expressive and tractable
+ The connection between determinism to orthogonality seems to be unique and perhaps will offer other research directions
+ Guarantees on inference complexity with unitary circuits being in normalized form
+ Paper is high on rigor with proofs for all the key results

**Weaknesses:**

- While the paper makes a strong contribution in probabilistic circuits with the introduction of unitary circuit learning and inference, it does not show why unitary circuits are better than existing tractable probabilistic models. The baseline comparison is with variants of squared PCs but perhaps the benefits of squared PCs over other approaches is not as clear. Maybe this is an empirical aspect that seems missing in the paper.
- The choice of experiments and benchmarks was not so clear (MNIST and FashionMNIST). More generally, from the comments in lines 459-463 it seems like this approach is hard to scale for other general problems? This may be a limiting factor for broader use.

In general, the paper is strong in theory and presents novel theoretical results in the context of squared PCs. The empirical evaluation seems a bit weaker.

**Questions:**

Are squared PCs generalizable across different types of problems? Is the limiting factor learning them at larger scales?

---

> ### Author Response · Authors · 2025-11-22
>
> We thank the reviewer for pointing out the theoretical rigor as well as the novelty of our contributions. We believe we can address the raised points below.
>
> > [The paper] does not show why unitary circuits are better than existing tractable probabilistic models. [...] the benefits of squared PCs over other approaches is not as clear. Maybe this is an empirical aspect that seems missing in the paper.
>
> **Regarding the theory.** We stress that one key advantage of the unitary parameterization is that the partition function is constant, i.e., it is 1, and therefore we do not need to compute it at all. This is for example reflected in the efficiency gains for maximum likelihood learning shown in Figure 4 (a).  Furthermore, the benefit of squared PCs over PCs with only positive parameters is that squared PCs can be exponentially more expressive, meaning that squared PCs could require exponentially fewer parameters to represent the same distributions (see L161-163 in Section 2, and [C]). Most importantly, this result translates also to other tractable probabilistic models, such as mixture models and general bounded-treewidth graphical models, since PCs with positive parameters are known to generalize them [A] [B].
>
> **Regarding the experiments.** there exists another work that already compares squared PCs and other tractable probabilistic models, including HMMs, other models based on tensor networks and PCs with positive parameters. These results are shown in Figures 4 and 5 in [C]. We would like to stress that our empirical research questions do not focus on beating baseline tractable probabilistic models, but rather on understanding the efficiency and the expressiveness of the proposed unitary parameterization of squared PCs. We will make this point explicit in the revised version of the paper.
>
> >The choice of experiments and benchmarks was not so clear
>
> We discuss the choice of the benchmarks in the global answer above, together with other points mentioned by the other reviewers.
>
> >it seems like this approach is hard to scale for other general problems? Is the limiting factor learning them at larger scales?
>
> Since this is a common point made by the reviewers, we answer it in the global answer above.
>
> >Are squared PCs generalizable across different types of problems?
>
> Since squared PCs support efficient marginalization, we believe they can potentially be used in all types of problems where traditional PCs with positive parameters have already been used. For example, some existing applications are: fast lossless data compression [D], causal probabilistic inference [E], and neurosymbolic AI with guarantees regarding constraints satisfaction [F].
>
> [A] Robert Peharz. Robert Gens. Pedro Domingos. Learning Selective Sum-Product Networks. 2014.
>
> [B] YooJung Choi. Antonio Vergari. Guy Van den Broeck. A Unifying Framework for Tractable Probabilistic Models. 2020.
>
> [C] Lorenzo Loconte, Aleksanteri Mikulus Sladek, Stefan Mengel, Martin Trapp, Arno Solin, Nicolas Gillis, Antonio Vergari. Subtractive Mixture Models via Squaring: Representation and Learning. 2024.
>
> [D] Anji Liu, Stephan Mandt, Guy Van den Broeck. Lossless Compression with Probabilistic Circuits. 2022.
>
> [E] Benjie Wang, Marta Kwiatkowska. Compositional Probabilistic and Causal Inference using Tractable Circuit Models. 2023.
>
> [F] Kareem Ahmed, Stefano Teso, Kai-Wei Chang, Guy Van den Broeck, Antonio Vergari. Semantic Probabilistic Layers for Neuro-Symbolic Learning. 2022.

---

### Official Review · Reviewer_kdCq · 2025-10-31

**Soundness:** 4
**Presentation:** 4
**Contribution:** 3
**Rating:** 8
**Confidence:** 4

**Summary:**

This work builds on previous work that investigates the relation between Tensor Networks and Squared Probabilistic Circuits, two computational frameworks developed in different communities. The Matrix-Product State representation of Tensor networks have been previously shown to be equivalent to structured and decomposable circuits with negative weights. Both Tensor networks and Circuits with negative weights can be constrained in order to satisfy probability distributions by squaring and renormalization. As previously noted, Squared Probabilistic Circuits are more general than MPS Tensor Networks, which makes the connection interesting. In this work, the authors investigate the orthogonality property often used by MPS to decrease the overhead of renormalization and squaring from quadratic to linear. That property is lacking from the Probabilistic Circuits literature. Since a direct adaptation of orthogonality leads to squared Probabilistic Circuits with a relatively simple structure, the authors propose a relaxation of the property, called Z-orthogonality (inspired by the concept of X-determinism in PCs). They show how the property can be exploited to speed up marginal computations while increasing flexibility of the model. Experiments in simple tasks (MNNIST and FASHION MNIST) show that their approach can effectively learn good representations and improve scalability.

**Strengths:**

- Interesting discussion, connecting topics studied in different communities
- Well-written
- Promising approach to learning high-dimensional probability distributions with tractable marginalization

**Weaknesses:**

- Empirical analysis is very preliminary
- Very long text (including appendices), difficult to revise given time constraints of a conference

**Questions:**

All in all, I think this is a solid work, although I didn't have the time to check all the material in the appendices.

The analysis of RQ2 states that "squared unitary PCs gracefully scale, matching the performance of their baseline counterparts." I could not understand from the plots in Figure 4 how one can reach that conclusion.

---

> ### Author Response · Authors · 2025-11-22
>
> We thank the reviewer for appreciating our writing and for finding the connections we draw interesting. Let us now address the points raised by the reviewer.
>
> >Empirical analysis is very preliminary.
>
> Since similar points were made by the other reviewers, we discuss this aspect in the global answer above.
>
> >The analysis of RQ2 states that "squared unitary PCs gracefully scale, matching the performance of their baseline counterparts." I could not understand from the plots in Figure 4 how one can reach that conclusion.
>
> Thank you for pointing this out. We agree that the content of L456-L463 could have been better aligned with the results shown in Figure 4. We will update the paper by being more explicit on describing the content of the figure in text. We summarize our changes below.
>
> In Figure 4 (b) we plot bits-per-dimension achieved by the baseline squared PC with unconstrained parameters (denotes as $\pm_{\mathbb{C}}^2$, in orange) and the squared PC with semi-unitary parameters (denoted as $\perp_{\mathbb{C}}^2$, in green). In particular, we show that the two curves (orange and green ones) comparing bits-per-dimension and number of parameters decrease very similarly. This is why we say squared unitary PCs match their baseline counterpart in terms of scaling.

---

### Official Review · Reviewer_zFDG · 2025-11-01

**Soundness:** 3
**Presentation:** 4
**Contribution:** 3
**Rating:** 6
**Confidence:** 3

**Summary:**

This paper proposes new structural and parameterization conditions for squared tensor networks and squared probabilistic circuits that enable linear-time marginalization without losing expressiveness. Prior squared circuits incur quadratic complexity for computing marginals or partition functions, limiting scalability. The authors introduce orthogonality and unitarity constraints on circuits—generalizing canonical forms in tensor networks—to achieve efficient normalization and marginalization even for architectures that do not map to classical tensor-network forms. Experiments demonstrate that these unitary squared circuits are faster and more memory-efficient, while matching or exceeding the performance of unconstrained models.

**Strengths:**

1. Very well-written paper

2. Novel connection between tensor networks and probabilistic circuits. The paper conceptually unifies canonical forms in tensor networks with determinism in probabilistic circuits and introduces orthogonality as a more general tool for tractable inference .

3. Theoretical contributions with clear practical implications. Orthogonality and unitarity are shown to guarantee O(|c|) marginalization in squared circuits instead of O(|c|²), and results extend to non-structured-decomposable circuits.

**Weaknesses:**

1. Experiments cover MNIST-style tabular/image datasets; evaluation on more complex tasks (e.g., high-dimensional continuous density estimation, conditional queries, or sampling quality metrics) would strengthen the real-world significance.

2. The core intuition behind orthogonality vs determinism and its practical implications could be communicated more clearly for a broader audience.

3. Orthogonality/unitary constraints often complicate optimization; although addressed here, more ablation on optimizer sensitivity would be valuable.

**Questions:**

Please see the weakness above.

---

> ### Author Response · Authors · 2025-11-22
>
> We thank the reviewer for appreciating our connections between tensor factorizations and circuits and our theoretical contributions. We address the questions they ask below.
>
> >Evaluation on more complex tasks (e.g., high-dimensional continuous density estimation, conditional queries, or sampling quality metrics) would strengthen the real-world significance.
>
> We thank the reviewer for the feedback. Since similar points about the experiments were made by other reviewers, we answer this in the global answer above.
>
> >The core intuition behind orthogonality vs determinism and its practical implications could be communicated more clearly for a broader audience.
>
> We appreciate the feedback and are open to actionable suggestions. What do you think is missing in Section 3 on page 4?
>
> >Orthogonality/unitary constraints often complicate optimization; although addressed here, more ablation on optimizer sensitivity would be valuable.
>
> The fact that unitary constraints complicate optimization is a common point made by the other reviewers, so we answer it in the global answer above. Instead, below we discuss the optimizer ablations we performed in the initial submission.
>
> To achieve the results in the paper, we focused on LandingSGD optimizers [O1] [O2] because of their efficiency due to not requiring expensive retractions. However, we had to make a few modifications which we ablate in Figure E.2. of the initial submission, comparing the results of:
> - the vanilla LandingSGD optimizer, as in [O1] [O2],
> - a variant of LandingSGD (with the asterisk in Figure E.2.), where we project back to the manifold if a matrix is too distant from the manifold by more than a certain threshold, and
> - our LandingPC optimizer, where we replace the Euclidean gradient in SGD to the one obtained by the VectorAdam and RAdam algorithms.
>
> We further detail all their differences in Appendix F of the initial submission. In summary, Figure E.2. shows that some modifications to the LandingSGD algorithm are indeed necessary to achieve the shown competitive performances w.r.t. squared PCs with unconstrained parameters (see green line vs. brown line).
>
> [O1] Pierre Ablin and Gabriel Peyré. Fast and accurate optimization on the orthogonal manifold without retraction. 2022
>
> [O2] Pierre Ablin, Simon Vary, Bin Gao, P.-A. Absil. Infeasible deterministic, stochastic, and variance-reduction algorithms for optimization under orthogonality constraints. 2024

---

### Author Response · Authors · 2025-11-22

We thank the reviewers for their effort. In this global answer we address common points made by the reviewers regarding: **(1)** the increased difficulty of optimizing over the Stiefel manifold (reviewers zFDG, kdCq and eo4A); and **(2)** the experiments on image distribution estimation benchmarks are preliminary (reviewers zFDG, kdCq) and their choice is unclear (reviewer J6uk).

**(1) About optimizing over the Stiefel manifold**

We agree with the reviewers that optimization on the Stiefel in high dimensions is challenging. However, we were able to learn squared PCs with thousands of semi-unitary parameter matrices, resulting in models having up to half a billion of parameters. Still, we obtain competitive performance with respect to unconstrained squared PCs optimized using Adam (see our Figure 4 (b)). We believe this is a remarkable success, given that we considered unitary circuits with similar size to PCs in recent previous works [A] [B] [C]. Moreover, as we acknowledge in the paper, optimizing over the Stiefel manifold is an active research area despite its challenges, and future improvements in the field would directly translate to benefits for our models as well.

**(2) About the experiments**

While we agree with the reviewers that evaluating on other tasks would be interesting, we stress this is not the main contribution of our paper, which proposes a new theoretical perspective. As such, we chose benchmarks on distribution estimation over images as it is arguably the most common evaluation setting appearing in the probabilistic circuits community [A] [B] [C] [D]. This is also because these benchmarks allow to scale PCs up to 1B of parameters due to the high-dimensionality of the data. We will stress these motivations in the revised version of the paper. Even if preliminary, we believe our results are promising and doing an empirical investigation on other tasks can be part of a different paper.

[A] Anji Liu, Honghua Zhang, Guy Van den Broeck. Scaling Up Probabilistic Circuits by Latent Variable Distillation. 2023.

[B] Gennaro Gala, Cassio de Campos, Antonio Vergari, Erik Quaeghebeur. Scaling Continuous Latent Variable Models as Probabilistic Integral Circuits. 2024.

[C] Lorenzo Loconte, Stefan Mengel, Antonio Vergari. Sum of Squares Circuits. 2025.

[D] Poorva Garg, Benjie Wang, Oliver Broadrick, Guy Van den Broeck, Todd Millstein. Bitblasting for Tractable Constrained Decorrelation in Image Modeling. 2025.

---

### Meta-Review · Area_Chair_m8Th · 2026-01-05

**Summary:**

The paper provides an interesting conceptual contribution to the theory of probabilistic circuits. Specifically, they provide a class of orthogonality-constrained circuits that support faster marginalization, and a framework for its implementation. The computation-expressiveness tradeoff is empirically validated on standard datasets. The reviews were overall quite positive, with the primary concern being the relatively limited empirical evaluation. This point was discussed in the rebuttal.

I think there is enough enthusiasm and follow-up potential based on the new framework and preliminary experiments to merit acceptance.

**Reviewer Concerns:**

The rebuttal justified the experiment choice, but the concerns re: relatively limited evaluation would likely still persist. Nonetheless, it seems that the existing experiments are promising enough that I don't think this outstanding concern pushes the paper back toward rejection.

**Reviewer Scores:**

I think the reviews would stay relatively similar.

---

### Decision · Program_Chairs · 2026-01-26

Accept (Poster)